# A metanalysis of microcosm experiments shows that dimethyl sulfide (DMS) production in polar waters is insensitive to ocean acidification

Frances E. Hopkins[1], Philip D. Nightingale[1], John A. Stephens[1], C. Mark Moore[2], Sophie

Richier[2], Gemma L. Cripps[2], Stephen D. Archer[3]

[1]Plymouth Marine Laboratory, Plymouth, PL1 3DH, U.K.

[2]Ocean and Earth Science, National Oceanography Centre, University of Southampton, Southampton, U.K.

[3]Bigelow Laboratory for Ocean Sciences, Maine, U.S.A.

*Correspondence to*: Frances E. Hopkins (fhop@pml.ac.uk)

**Abstract.** Emissions of dimethylsulfide (DMS) from the polar oceans play a key role in

atmospheric processes and climate. Therefore, it is important to increase our understanding of

how DMS production in these regions may respond to climate change. The polar oceans are

particularly vulnerable to ocean acidification (OA). However, our understanding of the polar

DMS response is limited to two studies conducted in Arctic waters, where in both cases DMS

concentrations decreased with increasing acidity. Here, we report on our findings from seven

summertime shipboard microcosm experiments undertaken in a variety of locations in the

Arctic Ocean and Southern Ocean. These experiments reveal no significant effects of short

term OA on the net production of DMS by planktonic communities. This is in contrast to

similar experiments from temperate NW European shelf waters where surface ocean

communities responded to OA with significant increases in dissolved DMS concentrations. A

meta-analysis of the findings from both temperate and polar waters ($n = 18$ experiments)

reveals clear regional differences in the DMS response to OA.  Based on our findings, we

hypothesise that the differences in DMS response between temperate and polar waters reflect

the natural variability in carbonate chemistry to which the respective communities of each

region may already be adapted. If so, future temperate oceans could be more sensitive to OA

resulting in an increase in DMS emissions to the atmosphere, whilst perhaps surprisingly
DMS emissions from the polar oceans may remain relatively unchanged. By demonstrating
that DMS emissions from geographically distinct regions may vary in their response to OA,
our results may facilitate a better understanding of Earth's future climate. Our study suggests
that the way in which processes that generate DMS respond to OA may be regionally distinct
and this should be taken into account in predicting future DMS emissions and their influence
on Earth's climate.
**1 Introduction**
The trace gas dimethylsulfide (DMS) is a key ingredient in a cocktail of gases that exchange
between the ocean and atmosphere. Dissolved DMS is produced via the enzymatic
breakdown of dimethylsulfoniopropionate (DMSP), a secondary algal metabolite implicated
in a number of cellular roles, including the regulation of carbon and sulfur metabolism via an
overflow mechanism (Stefels, 2000) and protection against oxidative stress (Sunda et al.,
2002). Oceanic DMS emissions amount to 17 - 34 Tg S $y^{-1}$, representing 80 - 90% of all
marine biogenic S emissions, and up to 50% of global biogenic emissions (Lana et al., 2011).
DMS and its oxidation products play vital roles in atmospheric chemistry and climate
processes. These processes include aerosol formation pathways that influence the
concentration of cloud condensation nuclei (CCN) with implications for Earth's albedo and
climate (Charlson et al., 1987; Korhonen et al., 2008a), and the atmospheric oxidation
pathways of other key climate gases, including isoprene, ammonia and organohalogens (Chen
and Jang, 2012; von Glasow and Crutzen, 2004; Johnson and Bell, 2008). Thus, our ability to
predict the climate into the future requires an understanding of how marine DMS production
may respond to global change (Carpenter et al., 2012; Woodhouse et al., 2013; Menzo et al.,

50     2018).

The biologically-rich ice-edge regions and open seas of the Arctic are a strong source of DMS to the Arctic atmosphere (Levasseur, 2013). A seasonal cycle in CCN numbers can be related to seasonality in the Arctic DMS flux (Chang et al., 2011). Indeed, observations confirm that DMS oxidation products promote the growth of particles to produce aerosols that may influence cloud processes and atmospheric albedo (Bigg and Leck, 2001; Rempillo et al., 2011; Korhonen et al., 2008b; Chang et al., 2011). Arctic new particle formation events and peaks in aerosol optical depth (AOD) occur during summertime clean air periods (when levels of anthropogenic black carbon diminish), and have been linked to chlorophyll *a* maxima in surface waters and the presence of aerosols formed from DMS oxidation products such as methanesulfonate (MSA). The atmospheric oxidation products of DMS - $SO_2$ and $H_2SO_4$ - contribute to both the growth of existing particles and new particle formation (NPF) in the Arctic atmosphere (Leaitch et al., 2013; Gabric et al., 2014; Sharma et al., 2012). Thus, the ongoing and projected rapid loss of seasonal Arctic sea ice may influence the Arctic radiation budget via changes to both the DMS flux and the associated formation and growth of cloud-influencing particles (Sharma et al., 2012). The influence that OA will have on the production and flux of DMS, and how this may further influence the Arctic radiative balance, is poorly understood and requires further experimental and modelling efforts.

During its short but highly productive summer season, the Southern Ocean is a hotspot of DMS flux to the atmosphere, influenced by the prevalence of intense blooms of DMSP-rich *Phaeocystis antarctica* (Schoemann et al., 2005) and the presence of persistent  high winds particularly in regions north of the sub-Antarctic front (Jarníková and Tortell, 2016). Around 3.4 Tg of sulfur is released from the Southern Ocean to the atmosphere between December and February, a flux that represents ~15 % of global annual emissions of DMS (Jarníková and Tortell, 2016). Elevated CCN numbers are seen in the most biologically active regions of the Southern Ocean, with a significant contribution from DMS-driven secondary aerosol

formation processes (McCoy et al., 2015; Korhonen et al., 2008a). DMS-derived aerosols
from this region are estimated to contribute 6 to 10 W m$^{-2}$ to reflected short wavelength
radiation, similar to the influence of anthropogenic aerosols in the polluted Northern
Hemisphere (McCoy et al., 2015). Given this important influence of polar DMS emissions on
atmospheric processes and climate, it is vital we increase our understanding of the influence
of future ocean acidification on DMS production.
The polar oceans are characterised by high dissolved inorganic carbon ($C_T$) concentrations
and a low carbonate system buffering capacity, mainly due to the increased solubility of $CO_2$
in cold waters (Sabine et al., 2004; Orr et al., 2005). This makes these regions particularly
susceptible to the impacts of ocean acidification (OA). For example, extensive carbonate
mineral undersaturation is expected to occur in Arctic waters within the next 20 – 80 years
(McNeil and Matear, 2008; Steinacher et al., 2009). OA has already led to a 0.1 unit decrease
in global surface ocean pH, with a further fall of ~0.4 units expected by the end of the century
(Orr et al., 2005). The greatest declines in pH are likely in the Arctic Ocean with a predicted
fall of 0.45 units by 2100 (Steinacher et al., 2009), with a fall of ~0.3 units predicted for the
Southern Ocean (McNeil and Matear, 2008; Hauri et al., 2016). OA is occurring at a rate not
seen on Earth for 300 Ma, and so the potential effects on marine organisms, communities and
ecosystems could be wide-ranging and severe (Raven et al., 2005; Hönisch et al., 2012).
Despite the imminent threat to polar ecosystems and the importance of DMS emissions to
atmospheric processes, our knowledge of the response of polar DMS production to OA is
limited to a single mesocosm experiment performed in a coastal fjord in Svalbard (Riebesell
et al., 2013a; Archer et al., 2013) and one shipboard microcosm experiment with seawater
collected from Baffin Bay (Hussherr et al., 2017). Both studies reported significant
reductions in DMS concentrations with increasing levels of $p$CO$_2$ during seasonal
phytoplankton blooms. Hussherr et al. (2017) also saw reductions in total DMSP whilst
Archer et al. (2013) observed a significant increase in this compound, driven by $CO_2$-induced
increases in growth and abundance of dinoflagellates. However, these two single studies
provide limited information on the wider response of the open Arctic or Southern Oceans.
Mesocosm experiments have been a critical tool for assessing OA effects on surface ocean
communities (Engel et al., 2005; Engel et al., 2008; Schulz et al., 2008; Hopkins et al., 2010;
Schulz et al., 2013; Webb et al., 2015; Kim et al., 2006; Kim et al., 2010; Crawfurd et al.,
2016; Webb et al., 2016). The response of DMS to OA has been examined several times,
predominantly at the same site in Norwegian coastal waters (Vogt et al., 2008; Hopkins et al.,
2010; Webb et al., 2015; Avgoustidi et al., 2012), twice in Korean coastal waters (Kim et al.,
2010; Park et al., 2014), and a single study in the coastal Arctic waters of Svalbard (Archer et
al., 2013). Mesocosm enclosures, ranging in volume from ~11,000 – 50,000 L, allow the
response of surface ocean communities to a range of $CO_2$ treatments to be monitored under
near-natural light and temperature conditions over time scales (weeks - months). This is
sufficient time to allow a 'winners vs loser' dynamic to develop, whereby the succession of
the phytoplankton community is altered due to the differing sensitivities of different
taxonomic groups to changes in carbonate chemistry (Bach et al., 2017). The response of
DMS cycling to elevated $CO_2$ is generally driven by changes to the microbial community
structure (Brussaard et al., 2013; Archer et al., 2013; Hopkins et al., 2010; Engel et al., 2008).
The pseudo-natural conditions of mesocosm experiments offer the benefit of the inclusion of
community dynamics of three or more trophic levels, providing the opportunity to investigate
the influence of ecosystem dynamics on biogeochemical processes under experimental
conditions (Riebesell et al., 2013b). Furthermore, physical processes such as particle export
(Bach et al., 2016), which would be excluded by smaller scale experiments, can be
considered within the holistic mesocosm framework, and make the results relevant for use
within Earth system models (Six et al. 2013). However, the size, construction and associated
costs of mesocosms has limited their deployment to coastal/sheltered waters, resulting in
minimal geographical coverage, and leaving large gaps in our understanding of the response
of open ocean phytoplankton communities to OA.
Here, we adopt an alternative but complementary approach to explore the effects of OA on
the cycling of DMS with the use of short-term shipboard microcosm experiments. We build
on the previous temperate NW European shelf studies of Hopkins & Archer (2014) by
presenting data from four previously unpublished experiments from the NW European shelf
cruise, and by extending our experimental approach to the Arctic and Southern Oceans.
Vessel-based research enables multiple short term (days) near-identical incubations to be
performed over extensive spatial scales, that encompass natural gradients in carbonate
chemistry, temperature and nutrients (Richier et al., 2014; Richier et al., 2018). This allows
an assessment to be made of how a range of surface ocean communities, adapted to a variety
of environmental conditions, respond to the same driver. The focus is then on the effect of
short-term $CO_2$ exposure on physiological processes, as well as the extent of the variability in
acclimation between communities. The capacity of organisms to acclimate to changing
environmental conditions contributes to the resilience of key ecosystem functions, such as
DMS production. Therefore, do spatially-diverse communities respond differently to short
term OA, and can this be explained by the range of environmental conditions to which each is
presumably already adapted? The rapid $CO_2$ changes implemented in this study, and during
mesocosm studies, are far from representative of the predicted rate of change to seawater
chemistry over the coming decades, and the potential to induce a 'shock' response to the
sudden alteration of carbonate chemistry should be considered, particularly when working at
the smaller microcosm scale. Nevertheless, our approach can provide insight into the
physiological response and level of sensitivity to future OA of a variety of surface ocean
communities adapted to different in situ carbonate chemistry environments (Stillman and
Paganini, 2015), alongside the implications this may have for DMS production.
Communities of the NW European shelf consistently responded to acute OA with significant
increases in net DMS production, likely a result of an increase in stress-induced algal
processes (Hopkins and Archer, 2014). Do polar phytoplankton communities, which are
potentially adapted to contrasting biogeochemical environments, respond in the same way?
By expanding our approach to encompass both polar oceans, we can assess regional contrasts
in response. To this end, we combine our findings for temperate waters with those for the
polar oceans into a meta-analysis to advance our understanding of the regional variability and
drivers in the DMS response to OA.
**2 Material and Methods**
**2.1 Sampling stations**
This study presents new data from two sets of field experiments carried out as a part of the
UK Ocean Acidification Research Programme (UKOA) aboard the RRS James Clark Ross in
the sub-Arctic and Arctic in June-July 2012 (JR271) and in the Southern Ocean in January-
February 2013 (JR274). Data are combined with the results from an earlier study on board the
RRS Discovery (D366) described in Hopkins & Archer (2014) performed in the temperate
waters of the NW European shelf. Additionally, four previously unpublished experiments
from D366 are also included (E02b, E04b, E05b, E06) as well as two temperate experiments
from JR271 (NS and IB) (see Table 1). In total, 18 incubations were performed; 11 in
temperate and sub-Arctic waters of the NW European shelf and North Atlantic, 3 in Arctic
waters and 4 in the Southern Ocean. Figure 1 shows the cruise tracks, surface concentrations
of DMS and total DMSP (DMSPt) at CTD sampling stations as well as the locations of
sampling for shipboard microcosms (See Table 1 for further details).

## 2.2 Shipboard microcosm experiments

The general design and implementation of the experimental microcosms for JR271 and JR274 was essentially the same as for D366 and described in Richier et al. (2014), (2018) and Hopkins & Archer (2014), but with the additional adoption of trace metal clean sampling and incubation techniques in the low trace metal open ocean waters (see Richier et al. (2018)). At each station, pre-dawn vertical profiles of temperature, salinity, oxygen, fluorescence, turbidity and irradiance were used to choose and characterise the depth of experimental water collection. Subsequently, water was collected within the mixed layer from three successive separate casts of a trace-metal clean titanium CTD rosette comprising twenty-four 10 L Niskin bottles. Depth profiles of auxiliary measurements are shown in Figure 2. Each cast was used to fill one of a triplicated set of experimental bottles (locations and sample depths, Table 1). Bottles were sampled within a class-100 filtered air environment within a trace metal clean container to avoid contamination during the set up. The water was directly transferred into acid-cleaned 4.5 L polycarbonate bottles using acid-cleaned silicon tubing, with no screening or filtration.

The carbonate chemistry within the experimental bottles was manipulated by addition of equimolar HCl and $NaHCO_3^-$ (1 mol $L^{-1}$) to achieve a range of $CO_2$ treatments: Mid $CO_2$ (Target: 550 µatm), High $CO_2$ (Target: 750 µatm), High+ $CO_2$ (Target: 1000 µatm) and High++ $CO_2$ (Target: 2000 µatm) (Gattuso et al., 2010). Three treatment levels were used during the sub-Arctic/Arctic microcosms (Mid, High, High+). For Southern Ocean experiments, two experiments (*Drake Passage* and *Weddell Sea*) considered one $CO_2$ treatments (High). Three $CO_2$ treatments (High, High+, High++) were tested in the last two experiments (*South Georgia* and *South Sandwich*). Full details of the carbonate chemistry manipulations can be found in Richier et al. (2014) and Richier et al. (2018). Broadly, achieved $pCO_2$ levels were well-matched to target values at the start of the experiments (0 h),

although differences in $p\mathrm{CO}_2$ between target and initial values were greater in the higher
$p\mathrm{CO}_2$ treatments, due to lowered carbonate system buffer capacity at higher $p\mathrm{CO}_2$. For all 18
experiments, actual $p\mathrm{CO}_2$ values at 0 h were on average around $89\% \pm 12\%$ ($\pm$ 1 SD) of
target values. The attained $p\mathrm{CO}_2$ values, and $p\mathrm{CO}_2$ at each experimental time point, are
presented in Figures 3 and 4. After first ensuring the absence of bubbles or headspace, the
bottles were sealed with high density polyethylene (HDPE) lids with silicone/
polytetrafluoroethylene (PTFE) septa and placed in the incubation container. Bottles were
incubated inside a custom-designed temperature- and light-controlled shipping container, set
to match ($\pm$<1°C) the *in situ* water temperature at the time of water collection (shown in
Table 1) (see Richier et al. 2018). A constant light level (100 $\mu\mathrm{E}\ \mathrm{m}^{-2}\ \mathrm{s}^{-1}$) was provided by
daylight simulating LED panels (Powerpax, UK). The light period within the microcosms
was representative of *in situ* conditions. For the sub-Arctic/Arctic Ocean stations,
experimental bottles were subjected to continuous light representative of the 24 h daylight of
the Arctic summer. For Southern Ocean and all temperate water stations, an 18:6 light: dark
cycle was used. Each bottle belonged to a set of triplicates, and sacrificial sampling of bottles
was performed at two time points (see Table 1 for exact times). Use of three sets of triplicates
for each time point allowed for the sample requirements of the entire scientific party (3 x 3
bottles, x 2 time points (see Table 1 for specific times for each experiment), x 4 $\mathrm{CO}_2$
treatments = 72 bottles in total). Experiments were run for between 4 and 7 days (96 h – 168
h) (15 out of 18 experiments), with initial sampling proceeded by two further time points. For
three temperate experiments (E02b, E04b, E05b see Table 1 and Table 2) shorter two day
incubations were performed, with a single sampling point at the end. E06 was run for 96 h
(Table 1 and 2). Incubation times were extended for Southern Ocean stations *Weddell Sea*,
*South Georgia* and *South Sandwich* (see Table 1) as minimal $\mathrm{CO}_2$ response, attributed to
slower microbial metabolism at low water temperatures, was observed for Arctic stations and

the first Southern Ocean station *Drake Passage*. The differential growth/metabolic rates

between temperate and polar waters justify the comparison of response of shorter duration

temperate experiments and longer duration polar experiments. The magnitude of response

was not related to incubation times, and expected differences in net growth rates (2- to 3-fold

higher in temperate compared to polar waters (Eppley, 1972)) did not account for the

differences in response magnitude despite the increased incubation time in polar waters (see

Richier et al. (2018) for detailed discussion). Samples for carbonate chemistry measurements

were taken first, followed by sampling for DMS, DMSP and related parameters.

**2.3 Standing stocks of DMS and DMSP**

Methods for the determination of seawater concentrations of DMS and DMSP are identical to

those described in Hopkins & Archer (2014) and will therefore be described in brief here.

Seawater DMS concentrations were determined by cryogenic purge and trap, with gas

chromatography and pulsed flame photometric detection (GC-PFPD) (Archer et al., 2013).

DMSP concentrations were measured as DMS following alkaline hydrolysis. Samples for

total DMSP concentrations from temperate waters were fixed by addition of 35 µl of 50 %

$H_2SO_4$ to 7 mL of seawater (Kiene and Slezak, 2006), and analysed following hydrolysis

within 2 months of collection (Archer et al., 2013). Samples of DMSP that were collected in

polar waters were hydrolysed within 1 h of sample collection and analysed 6 – 12 h later. The

$H_2SO_4$ fixation method was not used for samples from polar waters given the likely

occurrence of *Phaeocystis sp*. which can result in the overestimation of DMSP concentrations

(del Valle et al., 2009). Similarly, concentrations of DMSPp were determined at each time

point by gravity filtering 7 ml of sample onto a 25 mm GF/F filter and preserving the filter in

7 ml of 35 mM $H_2SO_4$ in MQ-water (temperate samples) or immediately hydrolysing (polar

samples) and analysing by GC-PFPD. DMS calibrations were performed using alkaline cold-

hydrolysis (1 M NaOH) of DMSP sequentially diluted three times in MilliQ water to give
working standards in the range $0.03 - 3.3$ ng S mL$^{-1}$. Five point calibrations were performed
every $2 - 4$ days throughout the cruise.
**2.4 *De novo* DMSP synthesis**
*De novo* DMSP synthesis and gross production rates were determined for all microcosm
experiments, except *Barents Sea* and *South Sandwich*, at each experimental time point, using
methods based on the approach of Stefels et al. (2009) and described in detail in Archer et al.
(2013) and Hopkins and Archer (2014). Triplicate rate measurements were determined for
each $CO_2$ level. For each rate measurement three x 500 mL polycarbonate bottles were filled
by gently siphoning water from each replicate microcosm bottle. Trace amounts of
NaH$^{13}$CO$_3$, equivalent to ~6 % of *in situ* dissolved inorganic carbon ($C_T$), were added to each
500 mL bottle. The bottles were incubated in the microcosm incubation container with
temperature and light levels as described earlier. Samples were taken at 0 h, then at two
further time points over a 6 - 9 h period. At each time point, 250 mL was gravity filtered in
the dark through a 47 mm GF/F filter, the filter gently folded and placed in a 20 mL serum
vial with 10 mL of Milli-Q and one NaOH pellet, and the vial was crimp-sealed. Samples
were stored at -20°C until analysis by proton transfer reaction-mass spectrometer (PTR-MS)
(Stefels et al. 2009).
The specific growth rate of DMSP (μDMSP) was calculated assuming exponential growth
from:

$$\mu_t(\Delta t^{-1}) = \alpha_k \text{ x AVG}\left[\ln\left(\frac{^{64}\text{MP}_{eq} - {}^{64}\text{MP}_{t-1}}{^{64}\text{MP}_{eq} - {}^{64}\text{MP}_t}\right), \ln\left(\frac{^{64}\text{MP}_{eq} - {}^{64}\text{MP}_t}{^{64}\text{MP}_{eq} - {}^{64}\text{MP}_{t+1}}\right)\right]$$

268                                                                                                                    1

269 (Stefels et al. 2009) where $^{64}MP_t$, $^{64}MP_{t-1}$, $^{64}MP_{t+1}$ are the proportion of 1 x $^{13}C$ labelled

270 DMSP relative to total DMSP at time t, at the preceding time point (t-1) and at the subsequent

271 time point (t+1), respectively. Values of $^{64}MP$ were calculated from the protonated masses of

272 DMS as: mass 64/(mass63 + mass64 + mass65), determined by PTR-MS. $^{64}MP_{eq}$ is the

273 theoretical equilibrium proportion of 1 x $^{13}C$ based on a binomial distribution and the

274 proportion of tracer addition. An isotope fractionation factor $\alpha_k$ of 1.06 is included, based on

275 laboratory culture experiments using *Emiliania huxleyi* (Stefels et al. 2009). In vivo DMSP

276 gross production rates during the incubations (nmol $L^{-1}$ $h^{-1}$) were calculated from µDMSP

277 and the initial particulate DMSP (DMSPp) concentration of the incubations (Hopkins &

278 Archer 2014, Stefels et al. 2009). These rates provide important information on how the

279 physiological status of DMSP-producing cells may be affected by OA within the bioassays.

280 **2.5 Seawater carbonate chemistry analysis**

281 The techniques and methods used to determine both the *in situ* and experimental carbonate

282 chemistry parameters, and to manipulate seawater carbonate chemistry within the

283 microcosms, are described in Richier et al. (2014) and will be only given in brief here.

284 Experimental $T_0$ measurements were taken directly from CTD bottles, and immediately

285 measured for total alkalinity ($A_T$) (Apollo SciTech AS-Alk2 Alkalinity Titrator) and

286 dissolved inorganic carbon ($C_T$) (Apollo SciTech $C_T$ analyser (AS-C3) with LICOR 7000).

287 The CO2SYS programme (version 1.05) (Lewis and Wallace, 1998) was used to calculate the

288 remaining carbonate chemistry parameters including $pCO_2$.

289 Measurements of $T_A$ and $C_T$ were made from each bottle at each experimental time point and

290 again used to calculate the corresponding values for $pCO_2$ and $pH_T$. The carbonate chemistry

291 data for each sampling time point for each experiment are summarised in Supplementary

292 Table S1, S2 and S3 (Experimental starting conditions are given in Table 1).

## 2.6 Chlorophyll a (Chl *a*) determinations

Concentrations of Chl *a* were determined as described in Richier et al. (2014). Briefly, 100 mL aliquots of seawater from the incubation bottles were filtered through either 25 mm GF/F (Whatman, 0.7 μm pore size) or polycarbonate filters (Whatman, 10 μm pore size) to yield total and >10 μm size fractions, with the <10 μm fraction calculated by difference. Filters were extracted in 6 mL HPLC-grade acetone (90%) overnight in a dark refrigerator. Fluorescence was measured using a Turner Designs Trilogy fluorometer, which was regularly calibrated with dilutions of pure Chl *a* (Sigma, UK) in acetone (90%).

## 2.8 Community composition

Small phytoplankton community composition was assessed by flow cytometry. For details of methodology, see Richier et al. (2014).

## 2.9 Data handling and statistical analyses

Permutational analysis of variance (PERMANOVA) was used to analyse the difference in response of DMS and DMSP concentrations to OA, both between and within the two polar cruises in this study. Both dependant variables were analysed separately using a nested factorial design with three factors; (i) Cruise Location: Arctic and Southern Ocean, (ii) Experiment location nested within Cruise location (see Table 1 for station IDs) and (iii) $CO_2$ level: 385, 550, 750, 1000 and 2000 μatm. Main effects and pairwise comparisons of the different factors were analysed through unrestricted permutations of raw data. If a low number of permutations were generated then the *p*-value was obtained through random sampling of the asymptotic permutation distribution, using Monte Carlo tests.

One-way analysis of variance was used to identify differences in ratio of >10 μm Chl *a* to total Chl *a* ($chl_{>10um} : chl_{tot}$, see Discussion). Initially, tests of normality were applied ($p < 0.05$

= not normal), and if data failed to fit the assumptions of the test, linearity transformations of
the data were performed (logarithmic or square root), and the ANOVA proceeded from this
point. The results of ANOVA are given as follows: $F$ = ratio of mean squares, $df$ = degrees of
freedom, $p$ = level of confidence. For those data still failing to display normality following
transformation, a rank-based Kruskal-Wallis test was applied ($H$ = test statistic, $df$ = degrees
of freedom, $p$ = level of confidence).

## 3 Results

### 3.1 Sampling stations

At temperate sampling stations, sea surface temperatures ranged from 10.7°C for *Iceland*
*Basin*, to 15.3°C for *Bay of Biscay*, with surface salinity in the range 34.1 – 35.2, with the
exception of station E05b which had a relatively low salinity of 30.5 (Figure 2 and Table 1).
Seawater temperatures at the polar microcosm sampling stations ranged from -1.5°C at sea-
ice influenced stations (*Greenland Ice-edge* and *Weddell Sea*) up to 6.5°C for *Barents Sea*
(Fig. 2 A). Salinity values at all the Southern Ocean stations were <34, whilst they were ~35
at all the Arctic stations with the exception of *Greenland Ice-edge* which had the lowest
salinity of 32.5 (Fig. 2 B). Phototrophic nanoflagellate abundances were variable, with >3 x
$10^4$ cells mL$^{-1}$ at *Greenland Gyre*, 1.5 x $10^4$ cells mL$^{-1}$ at *Barents Sea* and <3 x $10^3$ cells mL$^{-1}$
for all other stations (Fig. 2 D). Total bacterial abundances ranged from 3 x $10^5$ cells mL$^{-1}$ at
*Greenland Ice-edge* up to 3 x $10^6$ cells mL$^{-1}$ at *Barents Sea* (Fig. 2 E).
Chl *a* concentrations in temperate waters ranged from 0.3 µg L$^{-1}$ for two North Sea stations
(*E05* and *North Sea*) up to 3.5 µg L$^{-1}$ for *Irish Sea* (Figure 2 and Table 1). Chl a was also
variable in polar waters, exceeding 4 µg L$^{-1}$ at *South Sandwich* and 2 µg L$^{-1}$ at *Greenland Ice-*
*edge*, whilst the remaining stations ranged from 0.2 µg L$^{-1}$ (*Weddell Sea*) to 1.5 µg L$^{-1}$

(*Barents Sea*) (Figure 2). The high Chl *a* concentrations at *South Sandwich* correspond to low in-water irradiance levels at this station (Fig. 2 C).

In temperate waters, maximum DMS concentrations were generally seen in near surface measurements, ranging from 1.0 nmol L$^{-1}$ for *E04* to 21.1 nmol L$^{-1}$ for *E06*, with rapidly decreasing concentrations with depth (Figure 2 G). As an exception to this, DMS concentrations at *South Sandwich* showed a sub-surface maximum of 15 nM at 32 m, coincident with a subsurface Chl *a* maximum of 5.4 µg L$^{-1}$. DMSP generally ranged from 12 – 20 nmol L$^{-1}$, except *Barents Sea* where surface concentrations exceeded 60 nmol L$^{-1}$ (Figure 2 H). DMSP tended to peak in the near surface waters, ranging from 12.0 nmol L$^{-1}$ for E04 to 72.5 nmol L$^{-1}$ for *E06*, although in some cases a subsurface maximum in overall DMSP concentrations was seen, as observed for *E05b* (89.8 nmol L$^{-1}$ 20 m), and again coincident with a subsurface Chl a peak of >2 µg L$^{-1}$ (Figure 2 F and H). Surface DMS concentrations in polar waters were generally lower than temperate waters, ranging from 1 – 3 nmol L$^{-1}$, with the exception of *South Sandwich* where concentrations of ~12 nmol L$^{-1}$ were observed (Figure 2 G), and resulted in high DMS:DMSP of 0.6 – 0.9 in the surface layer (Figure 2 I). DMS:DMSP did not exceed 0.5 at any other sampling stations.

**3.2 Response of DMS and DMSP to OA**

The temporal trend in DMS concentrations showed a similar pattern for the three Arctic Ocean experiments. Initial concentrations of 1 – 2 nmol L$^{-1}$ remained relatively constant over the first 48 h and then showed small increases of 1 - 4 nmol L$^{-1}$ over the remainder of the incubation period (Figure 3). Increased variability between triplicate incubations became apparent in all three Arctic experiments by 96 h, but no significant effects of elevated $CO_2$ on DMS concentrations were observed. Initial DMSP concentrations were more variable, from 6 nmol L$^{-1}$ at *Greenland Ice-edge* to 12 nmol L$^{-1}$ at *Barents Sea*, and either decreased slightly

(net loss 1 – 2 nmol L$^{-1}$ GG), or increased slightly (net increase ~4 nmol L$^{-1}$ *Greenland Ice-*
*edge*, ~3 nmol L$^{-1}$ *Barents Sea*) (Figure 5 A – C). DMSP concentrations were found to
decrease significantly in response to elevated $CO_2$ after 48 h for *Barents Sea* (Fig. 5 C, $t =$
2.05, $p = 0.025$), whist no significant differences were seen after 96 h. No other significant
responses in DMSP were identified.
The range of initial DMS concentrations was greater at Southern Ocean sampling stations
compared to the Arctic, from 1 nmol L$^{-1}$ at *Drake Passage* up to 13 nmol L$^{-1}$ at *South*
*Sandwich* (Figure 4). DMS concentrations showed little change over the course of 96 – 168 h
incubations and no effect of elevated $CO_2$, with the exception of *South Sandwich* (Fig. 4 D).
Here, concentrations decreased sharply after 96 h by between 3 and 11 nmol L$^{-1}$.
Concentrations at 96 h were $CO_2$-treatment dependent, with significant decreases in DMS
concentration occurring with increasing levels of $CO_2$ (PERMANOVA, $t = 2.61$, $p = 0.028$).
Significant differences ceased to be detectable by the end of the incubations (168 h). Initial
DMSP concentrations were higher at the Southern Ocean stations than for Arctic stations,
ranging from 13 nmol L$^{-1}$ for *Weddell Sea* to 40 nmol L$^{-1}$ for *South Sandwich* (Figure 5 D –
G). Net increases in DMSP occurred throughout, except at South Georgia, and were on the
order of between <10 nmol L$^{-1}$ - >30 nmol L$^{-1}$ over the course of the incubations.
Concentrations were not generally $p$CO$_2$-treatment dependent with the exception of the final
time point at *South Georgia* (144 h) when a significantly lower DMSP with increasing $CO_2$
was observed (PERMANOVA, $t = -5.685$, $p<0.001$).
Results from the previously unpublished experiments from temperate waters are in strong
agreement with the five experiments presented in Hopkins and Archer (2014), with
consistently decreased DMS concentrations and enhanced DMSP under elevated $CO_2$. The
data is presented in the Supplementary Information, Table S4 and Figure S2, and included in
the meta-analysis in section 4.1 of this paper.

**3.3 Response of de novo DMSP synthesis and production to OA**

Rates of *de novo* DMSP synthesis (μDMSP) at initial time points ranged from 0.13 $d^{-1}$ (*Weddell Sea*, Fig. 6 G) to 0.23 $d^{-1}$ (*Greenland Ice-edge*, Fig. 6 C), whilst DMSP production ranged from 0.4 nmol $L^{-1}$ $d^{-1}$ (*Greenland Gyre*, Fig. 6 B) to 2.27 nmol $L^{-1}$ $d^{-1}$ (*Drake Passage*, Fig. 6 F). Maximum rates of μDMSP of 0.37 -0.38 $d^{-1}$ were observed at *Greenland Ice-edge* after 48 h of incubation in all $CO_2$ treatments (Fig. 6 C). The highest rates of DMSP production were observed at *South Georgia* after 96 h of incubation, and ranged from 4.1 – 6.9 nmol $L^{-1}$ $d^{-1}$ across $CO_2$ treatments (Fig. 6 J). Rates of DMSP synthesis and production were generally lower than those measured in temperate waters (Hopkins and Archer, 2014) (Initial rates: μDMSP 0.33 – 0.96 $d^{-1}$, 7.1 – 37.3 nmol $L^{-1}$ $d^{-1}$), but were comparable to measurements made during an Arctic mesocosm experiment (Archer et al., 2013) (0.1 – 0.25 $d^{-1}$, 3 – 5 nmol $L^{-1}$ $d^{-1}$ in non-bloom conditions). The lower rates in cold polar waters likely reflect slower metabolic processes and are reflected by standing stock DMSP concentrations which were also lower than in temperate waters (5 – 40 nmol $L^{-1}$ polar, 8 – 60 nmol $L^{-1}$ temperate (Hopkins and Archer, 2014)). No consistent effect of high $CO_2$ were observed for either DMSP synthesis or production in polar waters, similar to findings for DMSP standing stocks. However, some notable but contrasting differences between $CO_2$ treatments were observed. There was a 36% and 37% increase in μDMSP and DMSP production respectively at 750 μatm for the *Drake Passage* after 96 h (Figure 6 E, F), and a 38% and 44% decrease in both at 750 μatm after 144 h for *Weddell Sea* (Figure 5 G, H). For *Drake Passage*, the difference between treatments at 96 h coincided with significantly higher nitrate concentrations in the High $CO_2$ treatment (Nitrate/nitrite at 96 h: Ambient = $18.9 \pm 0.2$ μmol $L^{-1}$, +$CO_2$ = $20.2 \pm 0.1$ μmol $L^{-1}$, ANOVA $F = 62.619$, $df = 1$, $p = 0.001$). However, it is uncertain whether the difference in nutrient availability between treatments (approximately 5 %) would be significant enough to strongly influence the rate of DMSP production.

The differences in DMSP production rates did not correspond to any other measured
parameter. It is possible that changes in phytoplankton community composition may have led
to differences in DMSP production rates for *Drake Passage* and *Weddell Sea*, but no
quantification of large cells (diatoms, dinoflagellates) was undertaken for these experiments.
**4 Discussion**
**4.1 Regional differences in the response of DMS(P) to OA**
We combine our findings from the polar oceans with those from temperate waters into a
meta-analysis in order to assess the regional variability and drivers in the DMS(P) response to
OA. Figures 7 and 8 provide an overview of the results discussed so far in this current study,
together with the results from Hopkins & Archer (2014) as well as the results from 4
previously unpublished microcosm experiments from the NW European shelf cruise and a
further 2 temperate water microcosm experiments from the Arctic cruise (*North Sea* and
*Iceland Basin*, Table 1). This gives a total of 18 microcosm experiments, each with between 1
and 3 high $CO_2$ treatments.
Hopkins & Archer (2014) reported consistent and significant increases in DMS concentration
in response to elevated $CO_2$ that were accompanied by significant decreases in DMSPt
concentrations. Bacterially-mediated DMS processes appeared to be insensitive to OA, with
no detectable effects on dark rates of DMS consumption and gross production, and no
consistent response seen in bacterial abundance (Hopkins and Archer, 2014). In general,
there were large short-term decreases in Chl *a* concentrations and phototrophic nanoflagellate
abundance in response to elevated $CO_2$ in these experiments (Richier et al., 2014).
The relative treatment effects ($[x]_{highCO2}/[x]_{ambientCO2}$) for DMS and DMSP (Figure 7), DMSP
synthesis and production (Figure 8), and Chl *a* and phototrophic nanoflagellate abundance
(Figure 9)  are plotted against the Revelle Factor of the sampled waters. The Revelle Factor
($R$), calculated here with CO2Sys using measurements of carbonate chemistry parameters ($R$
$= (\Delta pCO_2/\Delta TCO_2)/(pCO_2/TCO_2)$, Lewis and Wallace, 1998), describes how the partial
pressure of $CO_2$ in seawater ($PCO_2$) changes for a given change in DIC (Sabine et al., 2004;
Revelle and Suess, 1957). Its magnitude varies latitudinally, with lower values (9 – 12) from
the tropics to temperate waters, and the highest values in cold high latitude waters (13 – 15).
Thus polar waters can be considered poorly buffered with respect to changes in DIC.
Therefore, biologically-driven seasonal changes in seawater $pCO_2$ would result in larger
changes in pH than would be experienced in temperate waters (Egleston et al., 2010).
Furthermore, the seasonal sea ice cycle strongly influences carbonate chemistry, such that sea
ice regions exhibit wide fluctuations in carbonate chemistry (Revelle and Suess, 1957; Sabine
et al., 2004). Sampling stations with a $R$ above ~12 represent the seven polar stations (right of
red dashed line Fig. 7, 8, 9).  The surface waters of the polar oceans have naturally higher
levels of DIC and a reduced buffering capacity, driven by higher $CO_2$ solubility in colder
waters (Sabine et al., 2004). Thus, the relationship between experimental response and $R$ is a
simple way of demonstrating the differences in response to OA between temperate and polar
waters and provides some insight into how the $CO_2$ sensitivity of different surface ocean
communities may relate to the *in situ* carbonate chemistry. The effect of elevated $CO_2$ on
DMS concentrations at polar stations, relative to ambient controls, was minimal at both
sampling points, and is in strong contrast to the results from experiments performed in waters
with lower values of $R$ on the NW European shelf. In contrast, at temperate stations, DMSP
concentrations displayed a clear negative treatment effect, whilst at polar stations a positive
effect was evident under high $CO_2$ and particularly at the first time point (48 – 96 h) (Fig. 7 C
and D). *De novo* DMSP synthesis and DMSP production rates show a less consistent
response in either environment (Fig. 8 A and B), although  a significant suppression of
DMSP production rates in temperate waters compared to polar waters was seen (Fig. 8 B,
Kruskal-Wallis One Way ANOVA $H = 8.711$, $df = 1$, $p = 0.003$). A similar but not significant
response was seen for *de novo* DMSP synthesis (Fig. 8A).
Our data imply that DMSP concentrations in temperate waters were downregulated in
response to OA, attributed to the adverse effects of rapid OA on the growth of DMSP
producers which led to reductions in the abundance of these types of phytoplankton (Richier
et al. 2014, Hopkins and Archer 2014). By comparison, a more muted, but generally positive,
DMSP response was seen in polar waters at the first time point, whilst these treatment effects
were more or less undetectable by the second time point. There is some evidence that the
enhanced DMSP concentrations in polar waters were accompanied by increased DMSP
production rates (Figure 8), although data is not available for all experiments. However, these
changes may reflect a short term 'shock' physiological protective response to the
experimental OA, similar to that seen in response to other short term stressors such as high
irradiance that result in an increase in DMSP concentrations (Sunda et al., 2002;Galindo et
al., 2016). The lack of treatment effect in DMSP concentrations by the second time point may
be indicative that the community had, to some extent, acclimated to the change, allowing
DMSP production/concentrations to return to baseline levels. This may reflect a higher
degree of tolerance to rapid changes in carbonate chemistry amongst polar communities -
species which are already adapted to highly variable irradiance/carbonate chemistry regimes
(Thomas and Dieckmann, 2002; Rysgaard et al., 2012; Thoisen et al., 2015). Further
experiments with polar communities would help to unravel the potential importance of such
mechanisms and whether they facilitated the ability of polar phytoplankton communities to
resist the high $CO_2$ treatments.
The responses to OA observed for DMS and DMSP production are likely to be reflected in
the dynamics of the DMSP-producing phytoplankton. In an assessment across all
experiments, Richier et al. (2018) showed that the magnitude of biological responses to short
term $CO_2$ changes reflected the buffer capacity of the sampled waters. A consistent
suppression of net growth rates in small phytoplankton (<10 μm) and total Chl $a$
concentrations was observed under high $CO_2$ within experiments performed in temperate
waters with higher buffer capacity.
Generally, less significant relationships were found between the phytoplankton response and
the other wide range of physical, chemical or biological variables that were examined
(Richier et al. 2018).
In correspondence with the analyses carried out by Richier et al (2018), at 48 – 96 h (see
Table 1), a statistically significant difference in response was seen between temperate and
polar waters for Chl $a$ (Kruskal-Wallis One Way ANOVA $H = 20.577$, $df = 1$, $p<0.001$). In
general, at polar stations phytoplankton showed minimal response to elevated $CO_2$, in
contrast to a strong negative response in temperate waters (Fig. 9A). By the second time point
(96 – 144 h, see Table 1), no significant difference in response of Chl $a$ between temperate
and polar waters was apparent (Fig. 9B). As shown in Richier et al. (2014), phototrophic
nanoflagellates responded to high $CO_2$ with large decreases in abundance in temperate waters
and increases in abundance in polar waters (Fig. 9 C and D), with some exceptions: *North*
*Sea* and *South Sandwich* gave the opposite response. The responses had lessened by the
second time point (96 – 168 h, see Table 1).
In contrast, bacterial abundance did not show the same regional differences in response to
high $CO_2$ (see Hopkins and Archer (2014) for temperate waters, and Figure S1,
supplementary information, for polar waters). Bacterial abundance in temperate waters gave
variable and inconsistent responses to high $CO_2$. For all Arctic stations, as well as Southern
Ocean stations *Drake Passage* and *Weddell Sea*, no response to high $CO_2$ was observed. For
*South Georgia* and *South Sandwich*, bacterial abundance increased at 1000 and 2000 μatm,
with significant increases for *South Georgia* after 144 h of incubation (ANOVA $F = 137.936$,
$p<0.001$). Additionally, at Arctic stations *Greenland Gyre* and *Greenland Ice-edge*, no
overall effect of increased $CO_2$ on rates of DOC release, total carbon fixation or POC : DOC
was observed (Poulton et al. 2016).
Overall, the observed differences in the regional response of DMSP and DMS to carbonate
chemistry manipulation could not be attributed to any other measured factor that varied
systematically between temperate and polar waters. These include ambient nutrient
concentrations, which varied considerably but where direct manipulation had no influence on
the response, and initial community structure, which was not a significant predictor of the
phytoplankton response (Richier et al. 2018).
**4.2 Influence of community cell-size composition on DMS response**
It has been proposed that variability in the concentrations of carbonate species (e.g. $p$CO$_2$,
$HCO_3^-$, $CO_3^{2-}$) experienced by phytoplankton is related to cell size, such that smaller-celled
taxa (<10 μm) with a reduced diffusive boundary layer are naturally exposed to relatively less
variability compared to larger cells (Flynn et al., 2012). Thus, short-term and rapid changes in
carbonate chemistry, such as the kind imposed during our microcosm experiments, may have
a disproportionate effect on the physiology and growth of smaller celled species. Larger cells
may be better able to cope with variability as normal cellular metabolism results in significant
cell surface changes in carbonate chemistry parameters (Richier et al., 2014). Indeed, the
marked response in DMS concentrations to short term OA in temperate waters has been
attributed to this enhanced sensitivity of small phytoplankton (Hopkins and Archer, 2014).
Was the lack of DMS response to OA in polar waters therefore a result of the target
communities being dominated by larger-celled, less carbonate-sensitive species?
Size-fractionated Chl *a* measurements give an indication of the relative contribution of large
and small phytoplankton cells to the community. For experiments in temperate waters, the
mean ratio of >10 μm Chl *a* to total Chl *a* (hereafter *>10 μm : total*) of $0.32 \pm 0.08$ was lower
than the ratio for polar stations of $0.54 \pm 0.13$ (Table 2). Although the difference was not
statistically significant, this might imply a tendency towards communities dominated by
larger cells in the polar oceans, which may partially explain the apparent lack of DMS
response to elevated $CO_2$. However, this is not a consistent explanation for the observed
responses. For example, the Arctic *Barents Sea* station had the lowest observed *>10 μm :*
*total* of $0.04 \pm 0.01$, suggesting a community comprised almost entirely of <10 μm cells; yet
the response to short term OA differed to the response seen in temperate waters. No
significant $CO_2$ effects on DMS or DMSP concentrations or production rates were observed
at this station, whilst total Chl *a* significantly increased under the highest $CO_2$ treatments
after 96 h (PERMANOVA $F = 33.239$, $p<0.001$). Thus, our cell size theory does not hold for
all polar waters, suggesting that regardless of the dominant cell size, polar communities are
more resilient to OA. In the following section, we explore the causes of this apparent
insensitivity to OA in terms of the environmental conditions to which the communities have
presumably adapted.
**4.3 Adaptation to a variable carbonate chemistry environment**
Given that DMS production by polar phytoplankton communities appeared to be insensitive
to experimental OA compared to significant sensitivity in temperate communities, we
hypothesise that polar communities are adapted to greater natural variability in carbonate
chemistry over spatial and seasonal scales. This greater variability is partly the result of the
lower buffering capacity (Revelle Factor) of polar waters compared to lower latitude waters,
and partly due to specific processes that occur in the polar regions that strongly alter DIC
concentrations (e.g. sea ice formation and melt, enhanced $CO_2$ dissolution into cold polar
waters, upwelling of $CO_2$ rich water). Therefore, polar plankton communities are not only
subject to geophysical processes that strongly alter in situ carbonate chemistry on both spatial
and seasonal scales, but such changes are accompanied by larger pH changes than would
occur in more strongly buffered temperate waters. Therefore, polar surface ocean
communities are perhaps more likely to experience fluctuations between high pH and low pH
over relatively smaller time/space scales (Tynan et al., 2016). Thus below, we discuss our
findings in the context of the spatial pH variability we observed for each cruise track, and
explore some of the processes that drive this variability in polar waters. Information on the
pH variability at each sampling station is not available, so we cannot be certain of the exact
carbonate chemistry variability to which each of the sampled communities may have been
exposed and adapted. However, we can consider the overall variability in carbonate
chemistry over the spatial scales of the cruise tracks to demonstrate the characteristics of each
study area.
The polar waters sampled during our study were characterised by pronounced gradients in
carbonate chemistry over relatively small spatial scales.  In underway samples taken along
each cruise track (Arctic Ocean 3500 nm, Southern Ocean 4000 nm), pH varied by 0.45 units
(8.00 – 8.45) in the Arctic, and 0.40 units (8.30 - 7.90) in the Southern Ocean (Tynan et al.
2016). In some cases this range in variability was seen over relatively small distances: Figure
4 in Tynan et al. (2016) shows that pH fluctuated from 8.45 and 8.0 over a distance of 50 –
100 miles in the sea-ice influenced Fram Strait. By comparison, pH varied by a total of 0.2
units (8.22 - 8.02) in underway samples from the NW European shelf sea cruise (Rerolle et
al. 2014).  The observed horizontal gradients in polar waters were driven by different
physical and biogeochemical processes in each ocean. In the Arctic Ocean, this variability in
carbonate chemistry was partly driven by physical processes that controlled water mass
composition, temperate and salinity, particularly in areas such as the Fram Strait and
Greenland Sea. Along the ice-edge and into the Barents Sea, biological processes exerted a
strong control, as abundant iron resulted in high chlorophyll concentrations, low DIC and
elevated pH. By contrast, variations in temperature and salinity had only a small influence on
carbonate chemistry in the Southern Ocean in areas with iron limitation, and larger changes
were driven by a combination of calcification, advection and upwelling. Where iron was
replete, e.g. near South Georgia, biological DIC drawdown had a large impact on carbonate
chemistry (Tynan et al. 2016). A further set of processes was in play in sea ice influenced
regions. At the Arctic ice edge, abundant iron drove strong bloom development along the ice
edge, whilst sea ice retreat in the Southern Ocean was not always accompanied by iron
release (Tynan et al. 2016).
For comparison with Arctic stations, Hagens and Middelburg (2016) report a seasonal pH
variability of up to 0.25 units from a single site in the open ocean surface waters in the
Iceland Sea, whilst Kapsenberg et al. (2015) report an annual variability of 0.3 – 0.4 units in
the McMurdo Sound, Antarctica. This implies that both open ocean and sea ice-influenced
polar waters experience large variations in carbonate chemistry over seasonal cycles. By
contrast, monthly averaged surface $p\text{CO}_2$ data collected from station L4 in the Western
English Channel over the period 2007 – 2011 provides an example of typical carbonate
chemistry dynamics in NW European shelf sea waters. Over this period, pH had an annual
range of 0.15 units (8.05 – 8.20), accompanied by a range in $p\text{CO}_2$ of 302 – 412 µatm (Kitidis
et al., 2012).
The sea ice environment in particular is characterised by strong spatial and seasonal
variability in carbonate chemistry. Sea ice  is inhabited by a specialised microbial community
with a complex set of metabolic and physiological adaptations allowing these organisms to
withstand wide fluctuations in pH up to as high as 9.9 in brine channels to as low as 7.5 in the
under-ice water (Thomas and Dieckmann, 2002; Rysgaard et al., 2012; Thoisen et al., 2015).
The open waters associated with the ice edge also experience strong gradients in pH and
other carbonate chemistry parameters. This can be attributed to two processes: 1. The strong
seasonal drawdown of DIC due to rapid biological uptake by phytoplankton blooms at the
productive ice edge which drives up pH. On the Arctic cruise, increases of up to 0.33 pH
units were attributed to such processes in this region (Tynan et al., 2016). The effect was less
dramatic in the Fe-limited and less productive Weddell Sea with gradients in pH ranging
from 8.20 – 8.10 (Tynan et al., 2016). 2. The drawdown of DIC is countered by the release
and accumulation of respired DIC under sea ice due to the degradation of organic matter.
However, this accumulation occurs in subsurface/bottom waters, which are isolated from the
productive surface mixed layer by strong physical stratification and hence, of less relevance
to the current study.
The influence of sea ice on carbonate chemistry combined with the strong biological
drawdown of DIC in polar waters may have influenced the ability of some of the
communities we sampled during our study to withstand the short term changes to carbonate
chemistry they experienced within the bioassays. Two of our sampling stations were 'sea-ice
influenced': *Greenland Ice Edge* and *Weddell Sea*. Both were in a state of sea ice retreat as
our sampling occurred in the summer months. Sampling for the *Greenland Ice Edge* station
was performed in open, deep water, near to an area of thick sea ice, with low fluorescence but
reasonable numbers of diatoms (Leakey, 2012). Similarly, the *Weddell Sea* station was
located near the edge of thick pack ice but in an area of open water that allowed sampling to
occur without hindrance by brash ice (Tarling, 2013). At both stations we saw little or no
response in DMS or DMSP to experimental acidification, which may imply that the *in situ*
communities were more or less adapted to fluctuations in pH. Our experimental OA resulted
in pH decreases of between 0.4 and 0.7 units. However, it is unclear whether the communities
we sampled were able to withstand the artificial pH perturbation because they were adapted
to living in sea ice, or whether they had adapted to cope with other fluctuations in carbonate
chemistry that occur in polar waters.
In summary, this demonstrates the high variability in carbonate chemistry, including pH,
which polar communities may experience relative to their temperate counterparts, and which
is partly driven by the lowered buffer capacity of polar waters to changes in DIC, relative to
the more well-buffered temperate waters. This may have resulted in polar communities that
have adapted to and are more resilient to experimentally-induced OA. Of course, it is
important to recognise that this data represent only a snapshot (4 – 6 weeks) of a year, and
thus does not contain information on the range in variability over daily and seasonal cycles,
timescales which might be considered most important in terms of the carbonate system
variability experienced by the cells and how this drives $CO_2$ sensitivity (Flynn et al. 2012;
Richier et al. 2018). Nevertheless, this inherent carbonate chemistry variability experienced
by organisms living in polar waters may equip them with the resilience to cope with both
experimental and future OA.
Adaptation to such natural variability may induce the ability to resist abrupt changes within
the polar biological community (Kapsenberg et al., 2015). This is manifested here as
negligible impacts on rates of *de novo* DMSP synthesis and net DMS production in the
microbial communities of the polar open oceans to short term changes in carbonate
chemistry. A number of previous studies in polar waters have reported similar findings.
Phytoplankton communities were able to tolerate a $pCO_2$ range of 84 – 643 µatm in ~12 d
minicosm experiments (650 L) in Antarctic coastal waters, with no effects on
nanophytoplankton abundance, and enhanced abundance of picophytoplankton and
prokaryotes (Davidson et al., 2016; Thomson et al., 2016). In experiments under the Arctic
ice, microbial communities demonstrated the capacity to respond either by selection or
physiological plasticity to elevated $CO_2$ during short term experiments (Monier et al., 2014).
Subarctic phytoplankton populations demonstrated a high level of resilience to OA in short
term experiments, suggesting a high level of physiological plasticity that was attributed to the
prevailing strong gradients in $pCO_2$ levels experienced in the sample region (Hoppe et al.,
2017). Furthermore, a more recent study describing ten $CO_2$ manipulation experiments in
Arctic waters found that primary production was largely insensitive to OA over a large range
of light and temperature levels (Hoppe et al., 2018). This supports our hypothesis that,
relative to temperate communities, polar microbial communities may have a high capacity to
compensate for environmental variability (Hoppe et al., 2018), and are thus already adapted
to, and are able to tolerate, large variations in carbonate chemistry. Thus by performing
multiple, replicated experiments over a broad geographic range, the findings of this study
imply that the DMS response may be both a reflection of: (i) the level of sensitivity of the
community to changes in the mean state of carbonate chemistry, and (ii) the regional
variability in carbonate chemistry experienced by different communities. This highlights the
limitations associated with simple extrapolation of results from a small number of
geographically-limited experiments e.g. Six et al. (2013). Such an approach lacks a
mechanistic understanding that would allow a model to capture the regional variability in
response that is apparent from the microcosms experiments presented here.
**4.4 Comparison to an Arctic mesocosm experiment**
Experimental data clearly provide useful information on the potential future DMS response to
OA, but these data become most powerful when incorporated in Earth System Models (ESM)
to facilitate predictions of future climate. To date, two modelling studies have used ESM to
assess the potential climate feedback resulting from the DMS sensitivity to OA (Six et al.,
2013;Schwinger et al., 2017), and both have used results from mesocosm experiments.
However, the DMS responses to OA within our short term microcosm experiments contrast
with the results of most previous mesocosm experiments, and of particular relevance to this
study, an earlier Arctic mesocosm experiment (Archer et al., 2013). Whilst no response in
DMS concentrations to OA was generally seen in the polar microcosm experiments discussed
here, a significant decrease in DMS with increasing levels of $CO_2$ in the earlier mesocosm
study was seen. Therefore, it is useful to consider how the differences in experimental design,
and other factors, between microcosms and mesocosms may result in contrasting DMS
responses to OA.
The short duration of the microcosm experiments (4 – 7 d) allows the physiological
(phenotypic) capacity of the community to changes in carbonate chemistry to be assessed. In
other words, how well is the community adapted to variable carbonate chemistry and how
does this influence its ability to acclimate to change? Although the mesocosm experiment
considered a longer time period (4 weeks), the first few days can be compared to the
microcosms. No differences in DMS or DMSP concentrations were detected for the first
week of the mesocosm experiment, implying a certain level of insensitivity of DMS
production to the rapid changes in carbonate chemistry. In fact, when taking all previous
mesocosm experiments into consideration, differences in DMS concentrations have
consistently been undetectable during the first 5 – 10 days, implying there is a limited short-
term physiological response by the in situ communities (Hopkins et al., 2010; Avgoustidi et
al., 2012; Vogt et al., 2008; Kim et al., 2010; Park et al., 2014). This is in contrast to the
strong response in the temperate microcosms from the NW European shelf (Hopkins and
Archer, 2014). However, all earlier mesocosm experiments have been performed in coastal
waters, which like polar waters, can experience a large natural range in carbonate chemistry.
In the case of coastal waters this is driven to a large extent by the influence of riverine
discharge and biological activity (Fassbender et al., 2016). Thus coastal communities may
also possess a higher level of adaptation to variable carbonate chemistry compared to the
open ocean communities of the temperate microcosms (Fassbender et al., 2016).
The later stages of mesocosm experiments address a different set of hypotheses, and are less
comparable to the microcosms reported here. With time, an increase in number of generations
leads to community structure changes and taxonomic shifts, driven by selection on the
standing genetic variation in response to the altered conditions. Moreover, the coastal Arctic
mesocosms were enriched with nutrients after 10 days, affording relief from nutrient
limitation and allowing differences between $p$CO$_2$ treatments to be exposed, including a
strong DMS(P) response.(Archer et al., 2013; Schulz et al., 2013). During this period of
increased growth and productivity, CO$_2$ increases drove changes which reflected both the
physiological and genetic potential within the community, and resulted in taxonomic shifts.
The resultant population structure was changed, with an increase in abundance of
dinoflagellates, particularly *Heterocapsa rotundata*. Increases in DMSP concentrations and
DMSP synthesis rates were attributed to the population shift towards dinoflagellates. The
drivers of the reduced DMS concentrations were less clear, but may have been linked to
reduced DMSP-lyase capacity within the dominant phytoplankton, a reduction in bacterial
DMSP lysis, or an increase in bacterial DMS consumption rates (Archer et al., 2013). Again,
this is comparable to all other mesocosm experiments, wherein changes to DMS
concentrations can be associated with CO$_2$-driven shifts in community structure (Hopkins et
al., 2010; Avgoustidi et al., 2012; Vogt et al., 2008; Kim et al., 2010; Park et al., 2014; Webb
et al., 2015). However, given the lack of further experiments of a similar location, design and
duration to the Arctic mesocosm, it is unclear how representative the mesocosm result is of
the general community-driven response to OA in high latitude waters.
We did not generally see any broad-scale CO$_2$-effects on community structure in polar
waters. This can be demonstrated by a lack of significant differences in the mean ratio of >10

μm Chl *a* to total Chl *a* (*>10 μm : total*) between $CO_2$ treatments, implying there were no broad changes in community composition (Table 2). *South Sandwich* was an exception to this, where large and significant increases in the mean ratio of *>10 μm : total* were observed at 750 μatm and 2000 μatm $CO_2$ relative to ambient $CO_2$ (ANOVA, $F = 207.144$, $p<0.001$, *df = 3*), demonstrating that even at the short timescale of the microcosm experiments it is possible for some changes to community composition to occur. Interestingly, this was also the only polar station that exhibited any significant effects on DMS after 96 h of incubation (Figure 4 D). However, given the lack of similar response at 1000 μatm, it remains equivocal whether this was driven by a $CO_2$-effect or some other factor.

In contrast to our findings, a recent single 9 day microcosm experiment (Hussherr et al., 2017) performed in Baffin Bay (Canadian Arctic) saw a linear 80% decrease in DMS concentrations during spring bloom-like conditions. It should be noted that this response was seen over a range of *p*$CO_2$ from 500 - 3000 μatm, far beyond the levels used in the present study. Nevertheless, this implies that polar DMS production may be sensitive to OA at certain times of the year, such as during the highly productive spring bloom, but less sensitive during periods of low and stable productivity, such as the summer months sampled during this study. Furthermore, a number of other studies from both the Arctic e.g. (Coello-Camba et al., 2014; Holding et al., 2015; Thoisen et al., 2015) and the Southern Ocean e.g. (Trimborn et al., 2017; Tortell et al., 2008; Hoppe et al., 2013) suggest that polar phytoplankton communities can demonstrate sensitivity to OA, in contrast to our findings. This emphasises the need to gain a more detailed understanding of both the spatial and seasonal variability in the polar phytoplankton community and associated DMS response to changing ocean acidity.

**5 Conclusions**

We have shown that net DMS production by summertime polar open ocean microbial communities is insensitive to OA during multiple, highly replicated short term microcosm experiments. We provide evidence that, in contrast to temperate communities (Hopkins and Archer, 2014), the polar communities we sampled were relatively insensitive to variations in carbonate chemistry (Richier et al., 2018), manifested here as a minimal effect on net DMS production. Our findings contrast with two previous studies performed in Arctic waters (Archer et al. 2013; Hussherr et al. 2017) which showed significant decreases in DMS in response to OA. These discrepancies may be driven by differences in experimental design, variable sensitivity of microbial communities to changing carbonate chemistry between different areas, or by variability in the response to OA depending on the time of year, nutrient availability, and ambient levels of growth and productivity. This serves to highlight the complex spatial and temporal variability in DMS response to OA which warrants further investigation to improve model predictions.

Our results imply that the phytoplankton communities of the temperate microcosms initially responded to the rapid increase in $pCO_2$ via a stress-induced response, resulting in large and significant increases in DMS concentrations occurring over the shortest timescales (2 days), with a lessening of the treatment effect with an increase in incubation time (Hopkins and Archer 2014). The dominance of short response timescales in well-buffered temperate waters may also indicate rapid acclimation of the phytoplankton populations following the initial stress response, which forced the small-sized phytoplankton beyond their range of acclimative tolerance and lead to increased DMS (Richier et al. 2018, Hopkins and Archer 2014). This supports the hypothesis that populations from higher latitude, less well-buffered waters, already possess a certain degree of acclimative tolerance to variations in carbonate chemistry environment. Although initial community size structure was not a significant predictor of the response to high $CO_2$, it is possible that a combination of both community

composition and the natural range in variability in carbonate chemistry – as a function of
buffer capacity – may influence the DMS/P response to OA over a range of timescales
(Richier et al. 2018).
Our findings should be considered in the context of timescales of change (experimental vs
real world OA) and the potential of microbial communities to adapt to a gradually changing
environment. Microcosm experiments focus on the physiological response of microbial
communities to short term OA. Mesocosm experiments consider a timescale that allows the
response to be driven by community composition shifts, but are not long enough in duration
to incorporate an adaptive response. Neither approach is likely to accurately simulate the
response to the gradual changes in surface ocean pH that will occur over the next 50 – 100
years, nor the resulting changes in microbial community structure and distribution. However,
we hypothesise that the DMS response to OA should be considered not only in relation to
experimental perturbations to carbonate chemistry, but also in relation to the magnitude of
background variability in carbonate chemistry experienced by the DMS-producing organisms
and communities. Our findings suggest a strong link between the DMS response to OA and
background regional variability in the carbonate chemistry.
Models suggest the climate may be sensitive to changes in the spatial distribution of DMS
emissions over global scales (Woodhouse et al., 2013; Menzo et al., 2018). Such changes
could be driven by both physiological and adaptive responses to environmental change.
Accepting the limitations of experimental approaches, our findings suggest that net DMS
production from polar oceans may be resilient to OA in the context of its short term effects
on microbial communities. The oceans face a multitude of $CO_2$-driven changes in the coming
decades, including OA, warming, deoxygenation and loss of sea ice (Gattuso et al., 2015).
Our study addresses only one aspect of these future ocean stressors, but contributes to our
understanding of how DMS emissions from the polar oceans may alter, facilitating a better
understanding of Earth's future climate.
**Data availability**
All data has been deposited in and is accessible from the British Oceanographic Data Centre.
**Author contributions**
CMM, SR, FH, PDN and SDA designed the experiments. FH and JAS conducted the
measurements, FH and GLC analysed the data. FH prepared the paper with assistance and
contributions from all co-authors.
**Competing interests**
The authors declare that they have no conflict of interest.
**Financial support**
This work was funded under the UK Ocean Acidification thematic programme (UKOA) via
the UK Natural Environment Research Council (NERC) grants to PD Nightingale and SD
Archer (NE/H017259/1) and to T Tyrell, EP Achterberg and CM Moore (NE/H017348/1).
The UK Department for Environment, Food and Rural Affairs (Defra) and the UK
Department of Energy and Climate Change (DECC) also contributed to funding UKOA. The
National Science Foundation, United States, provided additional support to SD Archer ((NSF
OCE-1316133).
**Review statement**
This paper was edited by Gerhard Herdl and reviewed by three anonymous referees.
**Acknowledgements**
Our work and transit in the coastal waters of Greenland, Iceland and Svalbard was granted
thanks to permissions provided by the Danish, Icelandic and Norwegian diplomatic
authorities. We thank the captains and crew of the RRS Discovery (cruise D366) and RRS
James Clark Ross (cruises JR271 and JR274), and the technical staff of the National Marine
Facilities and the British Antarctic Survey. We are grateful to Mariana Ribas-Ribas and
Eithne Tynan for carbonate chemistry data, Elaine Mitchell and Clement Georges for flow
cytometry data, and Mariana Ribas-Ribas and Rob Thomas (BODC) for data management.

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

Table 1. Summary of the station locations and characteristic of the water sampled for the 18 microcosm experiments performed in temperate,
sub-polar and polar waters. All polar stations were sampled for JR271 and JR274, with the exception of NS and IB.

| Cruise | Station ID | Location | Sampling location | Sampling date | Sampling depth (m) | SST (°C) | Salinity | Nitrate (uM) | Total Chl $a$ ($\mu g\ L^{-1}$) | chl$_{>10\ \mu m}$ : chl$_{total}$ | $p$CO$_2$ ($\mu$atm) $T_0$ | pH (total) $T_0$ | Experimental timepoints $T_1$, $T_2$ (hours) | Reference |
|---|---|---|---|---|---|---|---|---|---|---|---|---|---|---|
| D366 | E01 | Mingulay Reef | 56°47.688N 7°24.300W | 8 June 2011 | 6 | 11.3 | 34.8 | 1.1 | 3.3 | no data | 334.9 | 8.1 | 48, 96 | *Hopkins & Archer (2* |
| | E02 | Irish Sea | 52°28.237N 5°54.052W | 14 June 2011 | 5 | 11.8 | 34.4 | 0.3 | 3.5 | 0.80 ± 0.03 | 329.3 | 8.1 | 48, 96 | *Hopkins & Archer (2* |
| | E02b | Bay of Biscay | 46°29.794N 7°12.355W | 19 June 2011 | 5 | 14.5 | 35.6 | 0.9 | 1.8 | no data | 340.3 | 8.1 | 48 | *This study* |
| | E03 | Bay of Biscay | 46°12.137N 7°13.253W | 21 June 2011 | 10 | 15.3 | 35.8 | 0.6 | 0.8 | 0.43 ± 0.03 | 323.9 | 8.1 | 48, 96 | *Hopkins & Archer (2* |
| | E04 | Southern North Sea | 52°59.661N 2°29.841E | 26 June 2011 | 5 | 14.6 | 34.1 | 0.9 | 1.3 | 0.19 ± 0.02 | 399.8 | 8.0 | 48, 96 | *Hopkins & Archer (2* |
| | E04b | Mid North Sea | 57°45.729N 4°35.434E | 29 June 2011 | 5 | 13.2 | 34.8 | No data | 0.5 | 0.14 ± 0.003 | 327.3 | 8.1 | 48 | *This study* |
| | E05 | Mid North Sea | 56°30.293N 3°39.506E | 2 July 2011 | 12 | 14.0 | 35.0 | 0.2 | 0.3 | 0.23 ± 0.01 | 360.2 | 8.1 | 48, 96 | *Hopkins & Archer (2* |
| | E05b | Atlantic Ocean | 59°40.721N 4°07.633E | 3 July 2011 | 4 | 13.4 | 30.7 | 0.3 | 0.7 | 0.12 ± 0.01 | 310.7 | 8.1 | 48 | *This study* |
| | E06 | Atlantic Ocean | 59°59.011N 2°30.896E | 3 July 2011 | 4 | 12.5 | 34.9 | 0.4 | 1.1 | 0.14 ± 0.01 | 287.1 | 8.2 | 48 | *This study* |
| JR271 | NS | Mid North Sea | 56°15.59N 2°37.59E | 3 June 2012 | 15 | 10.8 | 35.1 | 0.04 | 0.3 | 0.52 ± 0.05 | 300.5 | 8.2 | 48, 96 | *This study* |
| | IB | Iceland Basin | 60°35.39N 18°51.23W | 8 June 2012 | 7 | 10.7 | 35.2 | 5.0 | 1.8 | 0.27 ± 0.02 | 309.7 | 8.1 | 48, 96 | *This study* |
| | GG-AO | Greenland Gyre | 76°10.52 N 2°32.96 W | 13 June 2012 | 5 | 1.7 | 34.9 | 9.3 | 1.0 | 0.34 ± 0.001 | 289.3 | 8.2 | 48, 96 | *This study* |
| | GI-AO | Greenland ice edge | 78°21.15 N 3°39.85 W | 18 June 2012 | 5 | -1.6 | 32.6 | 4.2 | 2.7 | 0.78 ± 0.03 | 304.7 | 8.1 | 48, 96 | *This study* |
| | BS-AO | Barents Sea | 72°53.49 N 26°00.09 W | 24 June 2012 | 5 | 6.6 | 35.0 | 5.4 | 1.3 | 0.04 ± 0.01 | 304.3 | 8.1 | 48, 96 | *This study* |
| JR274 | DP-SO | Drake Passage | 58°22.00 S 56°15.12 W | 13 Jan 2013 | 8 | 1.9 | 33.2 | 22.0 | 2.4 | 1.00 ± 0.06 | 279.3 | 8.2 | 48, 96 | *This study* |
| | WS-SO | Weddell Sea | 60°58.55 S 48°05.19 W | 18 Jan 2013 | 6 | -1.4 | 33.6 | 24.9 | 0.6 | 0.67 ± 0.06 | 510.5 | 7.9 | 72, 144 | *This study* |
| | SG-SO | South Georgia | 52°41.36 S 36°37.28 W | 25 Jan 2013 | 5 | 2.2 | 33.9 | 24.1 | 0.7 | 0.35 ± 0.04 | 342.6 | 8.1 | 72, 144 | *This study* |
| | SS-SO | South Sandwich | 58°05.13 S 25°55.55 W | 1 Feb 2013 | 7 | 0.5 | 33.7 | 18.5 | 4.6 | 0.57 ± 0.02 | 272.6 | 8.2 | 96, 168 | *This study* |

Table 2. Mean (± SD) ratio of >10μm Chl $a$ to total Chl $a$ (chl$_{>10\mu m}$ :chl$_{total}$) for polar
microcosm sampling stations. * indicates significant difference from the response to ambient
$CO_2$. Exact CO2 treatments are down in Figure 3 and 4.

| Station | Time | Ambient | Mid $CO_2$ | High $CO_2$ | High+ $CO_2$ | High++ $CO_2$ |
|---|---|---|---|---|---|---|
| GG | | | | | | |
| | 48 h | 0.3 ± 0.1 | 0.3 ± 0.03 | 0.4 ± 0.2 | 0.3 ± 0.1 | N/A |
| | 96 h | 1.0 ± 0.02 | 0.9 ± 0.2 | 0.8 ± 0.1 | 0.7 ± 0.2 | |
| GI | | | | | | |
| | 48 h | 1.0 ± 0.1 | 1.0 ± 0.1 | 0.8 ± 0.1 | 1.0 ± 0.0 | N/A |
| | 96 h | 1.0 ± 0.1 | 1.1 ± 0.1 | 0.8 ± 0.1 | 0.8 ± 0.1 | |
| BS | | | | | | |
| | 48 h | 0.02 ± 0.01 | 0.04 ± 0.01 | 0.03 ± 0.01 | 0.02 ± 0.01 | N/A |
| | 96 h | 0.04 ± 0.01 | 0.05 ± 0.04 | 0.05 ± 0.04 | 0.04 ± 0.04 | |
| DP | | | | | | |
| | 48 h | 1.0 ± 0.3 | N/A | 1.0 ± 0.1 | N/A | N/A |
| | 96 h | 0.9 ± 0.1 | | 1.0 ± 0.1 | | |
| WS | | | | | | |
| | 72 h | 0.6 ± 0.1 | N/A | 0.7 ± 0.1 | N/A | N/A |
| | 144 h | 0.7 ± 0.1 | | 0.7 ± 0.1 | | |
| SG | | | | | | |
| | 72 h | 0.3 ± 0.02 | N/A | 0.4 ± 0.1 | 0.3 ± 0.1 | 0.4 ± 0.03 |
| | 144 h | 0.5 ± 0.1 | | 0.6 ± 0.04 | 0.5 ± 0.1 | 0.4 ± 0.03 |
| SS | | | | | | |
| | 96 h | 0.7 ± 0.04 | N/A | 1.5 ± 0.1* | 0.7 ± 0.02 | 1.6 ± 0.1* |
| | 168 h | 0.9 ± 0.2 | | 1.4 ± 0.02* | 0.8 ± 0.004 | 1.4 ± 0.2* |


Table 3. DMS and DMSPt response (mean ± SD, $n$ = 3) to high $CO_2$ treatments during
previously unpublished small-scale experiments from the NW European shelf cruise D366.
For details of sampling stations, see Table 1.

| | 0 h Ambient | 48 h Ambient | 48 h Mid $CO_2$ | 48 h High $CO_2$ | 96 h Ambient | 96 h Mid $CO_2$ | 96 h High $CO_2$ |
|---|---|---|---|---|---|---|---|
| **DMS (nM)** | | | | | | | |
| E02b | 2.4 ± 0.3 | 2.1 ± 0.6 | | 2.7 ± 0.6 | | | |
| E04b | | 6.4 ± 1.4 | | 14.7 ± 8.1 | | | |
| E05b | | 3.3 ± 0.1 | | 4.5 ± 0.6 | | | |
| E06 | 18.7 ± 0.5 | 18.1 | 24.2 | 25.2 | 18.1 | 24.2 | 25.3 |
| **DMSPt (nM)** | | | | | | | |
| E02b | | 49.5 ± 2.0 | | 26.4 ± 2.9 | | | |
| E04b | | 68.2 ± 10.3 | | 36.8 ± 7.5 | | | |
| E05b | | 48.7 ± 11.2 | | 37.4 ± 4.8 | | | |
| E06 | 76.7 ± 5.7 | 114.6 | 98.43 | 108.5 | 20.4 | 30.7 | 32.0 |




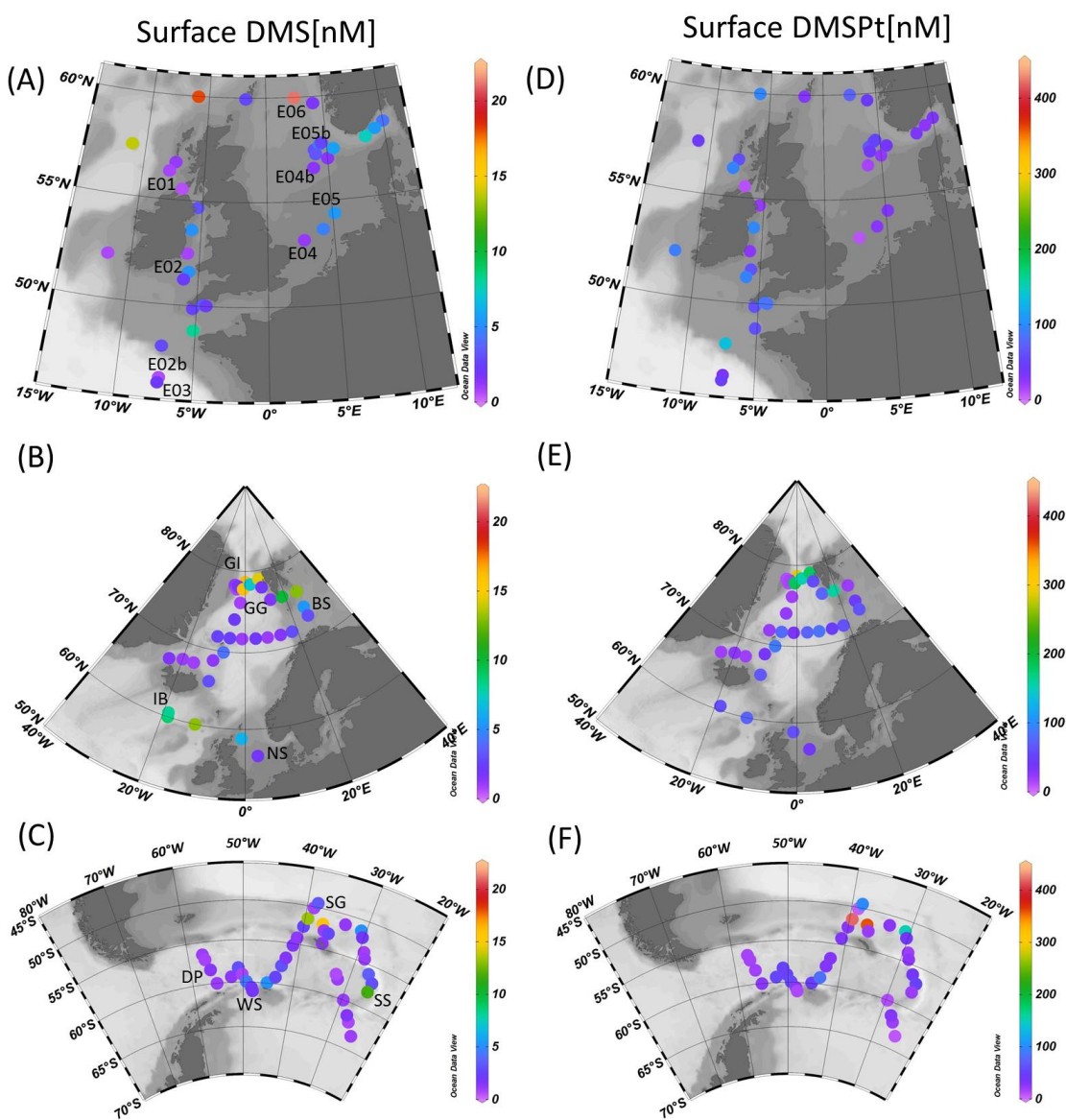


Figure 1. Surface (<5 m) concentrations (nM) of DMS (A-C) and total DMSP (D-F) for
cruises in the NW European shelf (D366) (A,D), the sub-Arctic and Arctic Ocean (JR271)
(B,E) and the Southern Ocean (JR274) (C,F). Locations of sampling stations for microcosm
experiments shown in letters/numbers. E01 – E05: see Hopkins & Archer 2014. NS = *North
Sea*, IB = *Iceland Basin*, GI = *Greenland Ice-edge*, GG = *Greenland Gyre*, BS = *Barents Sea*,
DP = *Drake Passage*, WS = *Weddell Sea*, SG = *South Georgia*, SS = *South Sandwich*.

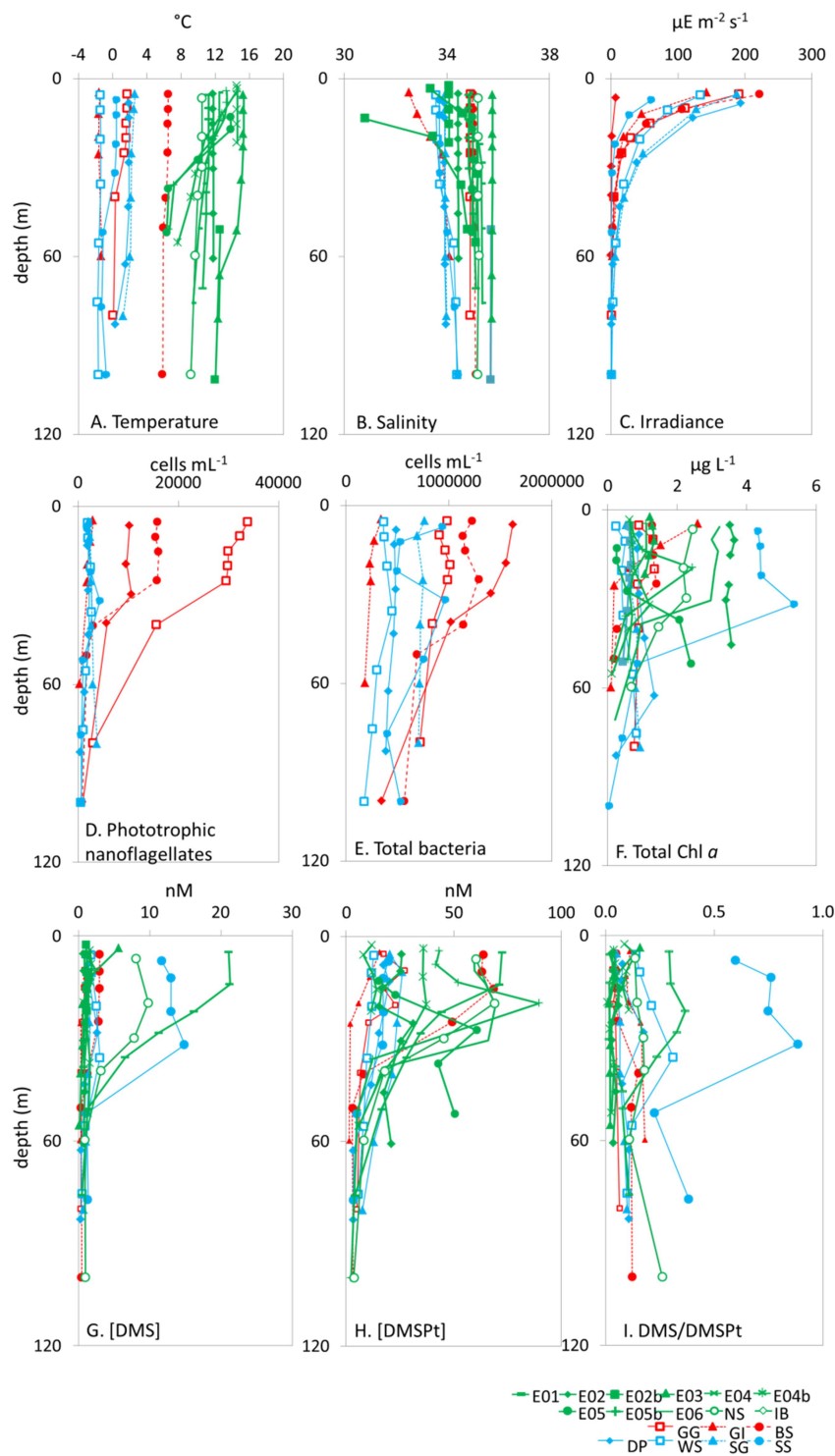

1217

Figure 2. Depth profiles down to 100 m depth for all 18 sampling stations showing A. Temperature (°C), B. Salinity, C. Irradiance ($\mu$E m$^{-2}$ s$^{-1}$), D. phototrophic nanoflagellate abundance (cells mL$^{-1}$), E. total bacteria abundance (cells mL$^{-1}$), F. total Chl a ($\mu$g L$^{-1}$), G. [DMS] (nM), H. total [DMSP] (nM) and I. DMS/DMSPt from CTD casts at sampling stations for microcosm experiments in temperate (green), Arctic (red) and Southern Ocean (blue) waters. See Table 1 for station details. Data for irrandiance, phototrophic nanoflagellates and total bacteria were not collected for temperate stations.

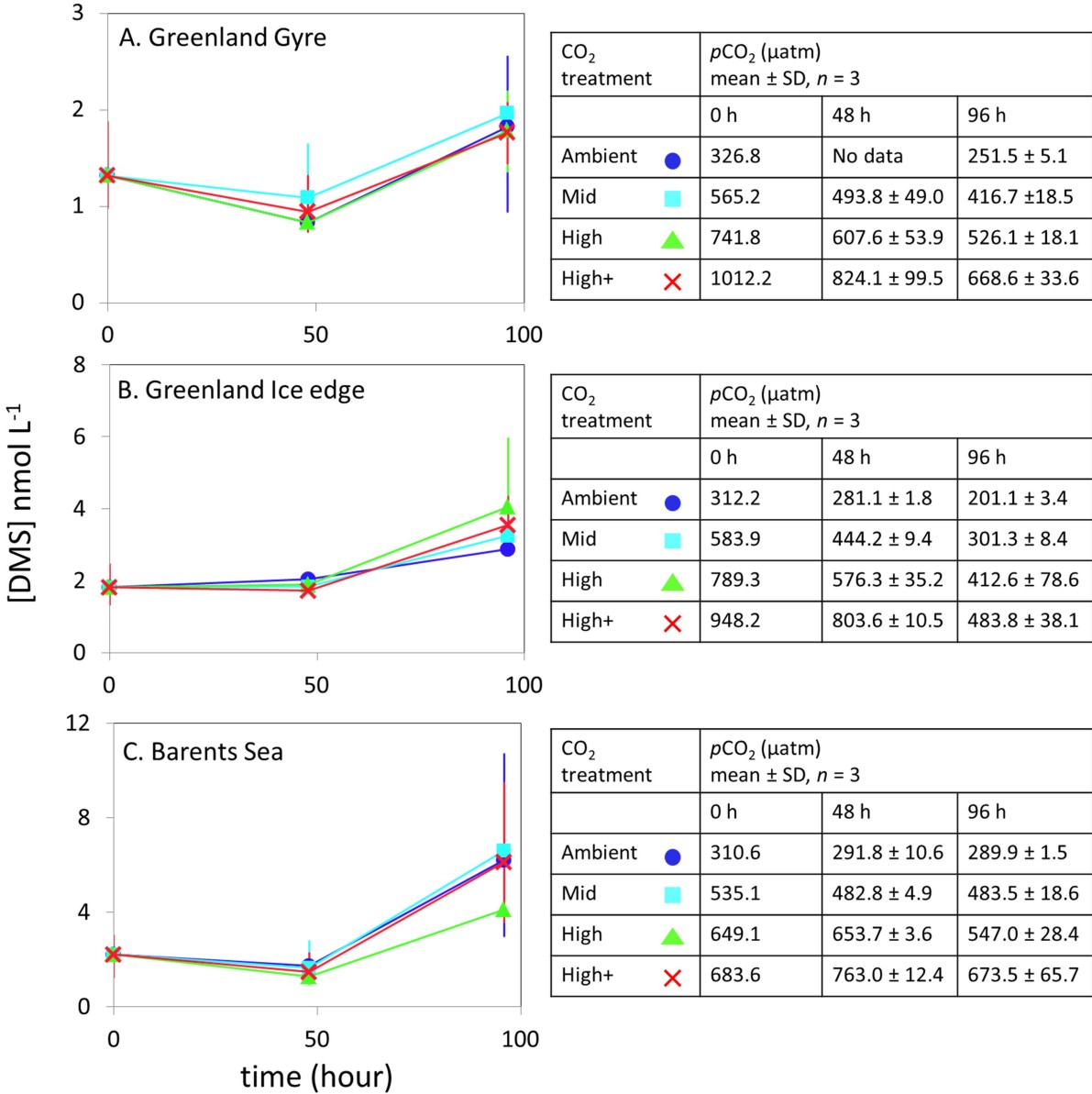

1226

Figure 3. DMS concentrations (nmol L$^{-1}$) during experimental microcosms performed in Arctic waters. Data shown is mean of triplicate incubations, and error bars show standard error on the mean. Tables show measurements of $p$CO$_2$ (µatm) for each treatment at each sampling time point. Initial measurements (0 h) were from a single sample, whilst measurements at 48 h and 96 h show mean ± SD of triplicate experimental bottles. Locations of water collection for microcosms shown in Figure 1 C – F.

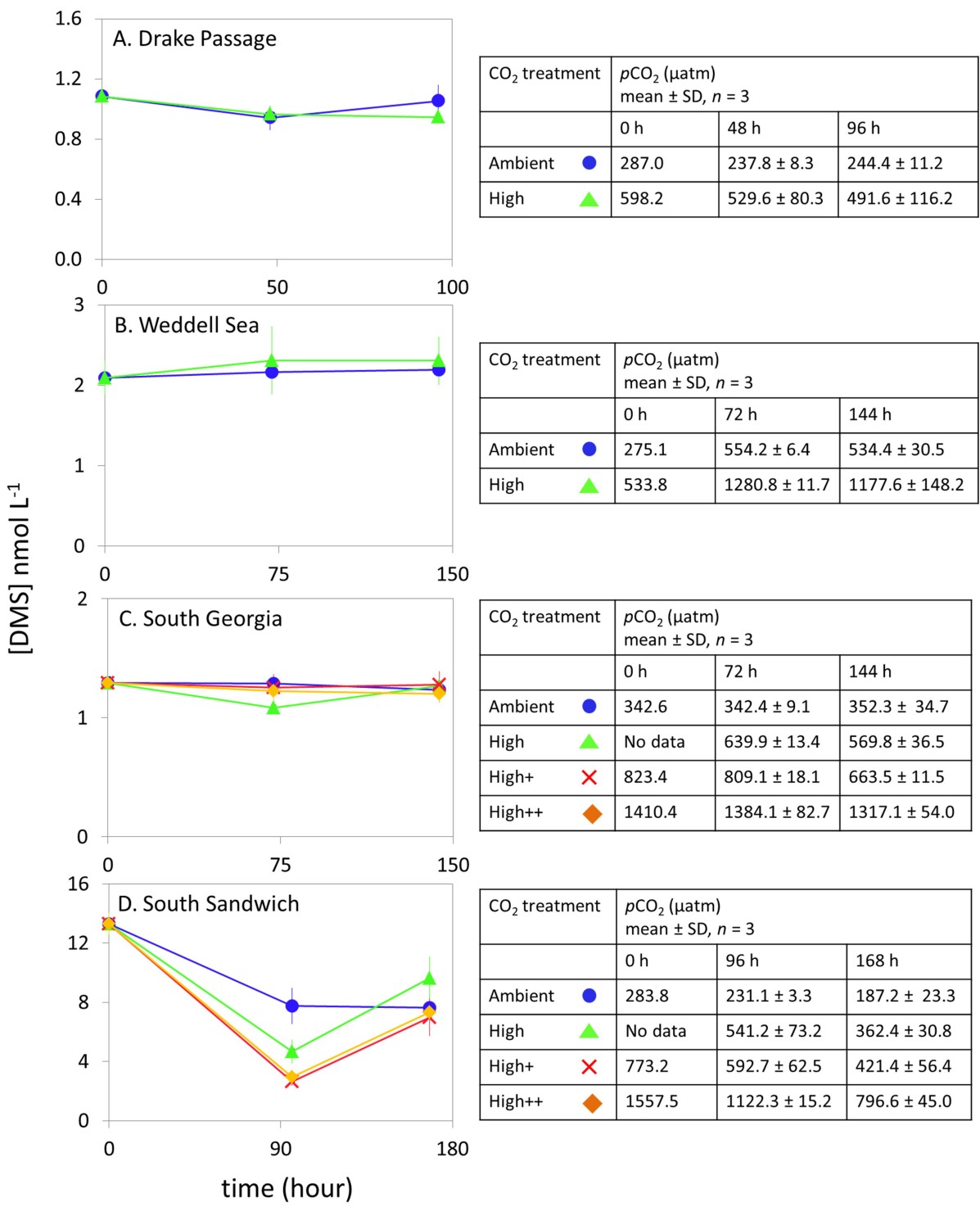

Figure 4. DMS concentrations (nmol L$^{-1}$) during experimental microcosms performed in
Southern Ocean waters. Data shown is mean of triplicate incubations, and error bars show
standard error on the mean. Tables show measurements of $p$CO$_2$ (µatm) for each treatment at
each sampling time point. Initial measurements (0 h) were from a single sample, whilst
measurements at 48 h and 96 h show mean ± SD of triplicate experimental bottles. Locations
of water collection for microcosms shown in Figure 1 C – F.

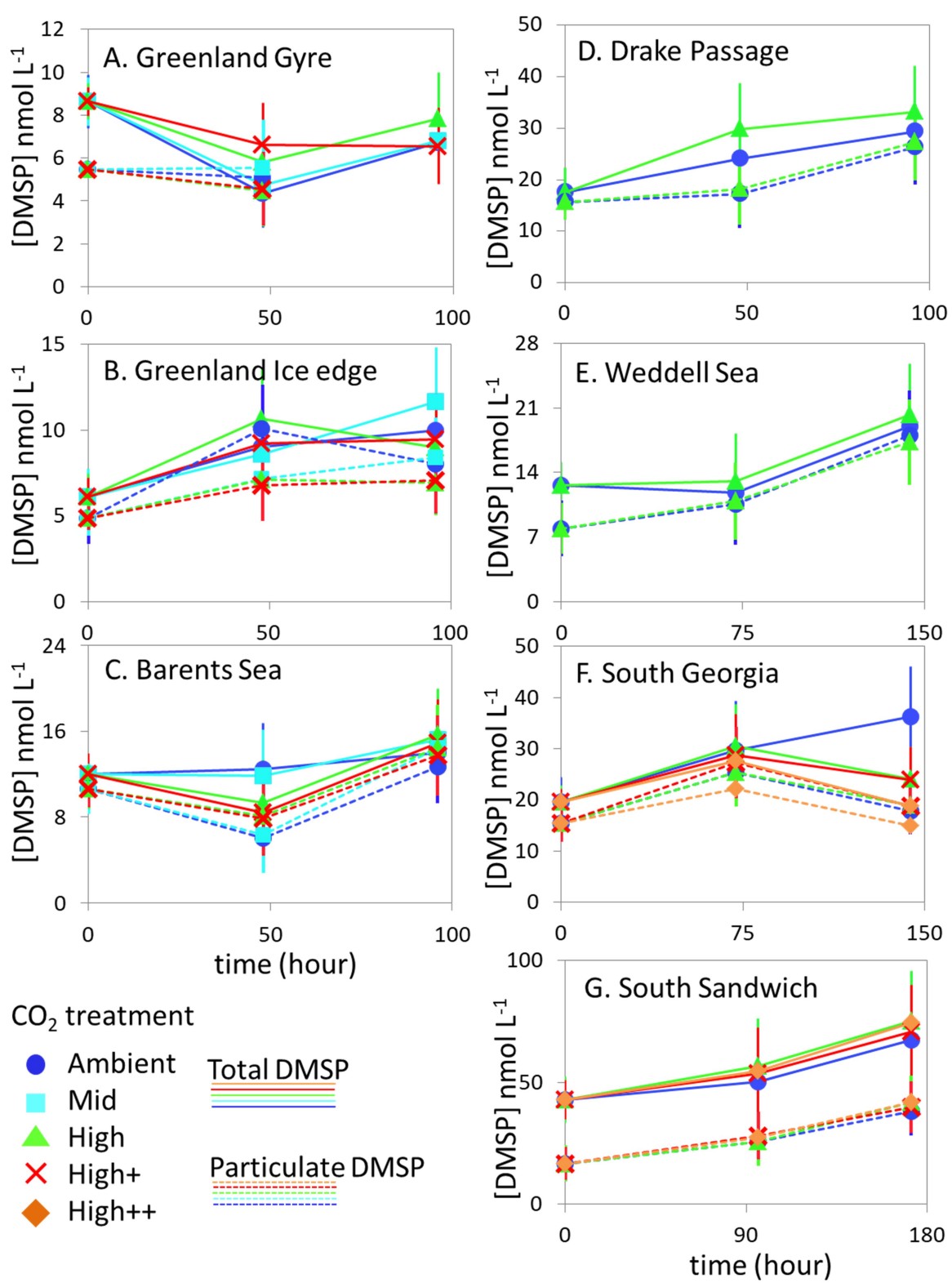


Figure 5. Total DMSP (solid lines) and particulate DMSP (dashed lines) concentrations (
nmol L$^{-1}$) during experimental microcosms performed in Arctic waters (A - C) and in
Southern Ocean waters (D – G). Data shown is mean of triplicate incubations, and error bars
show standard error on the mean. Locations of water collection for microcosms shown in
Figure 1 C – F. Particulate DMSP concentrations were used in calculations of DMSP
production rates (Figure 6).

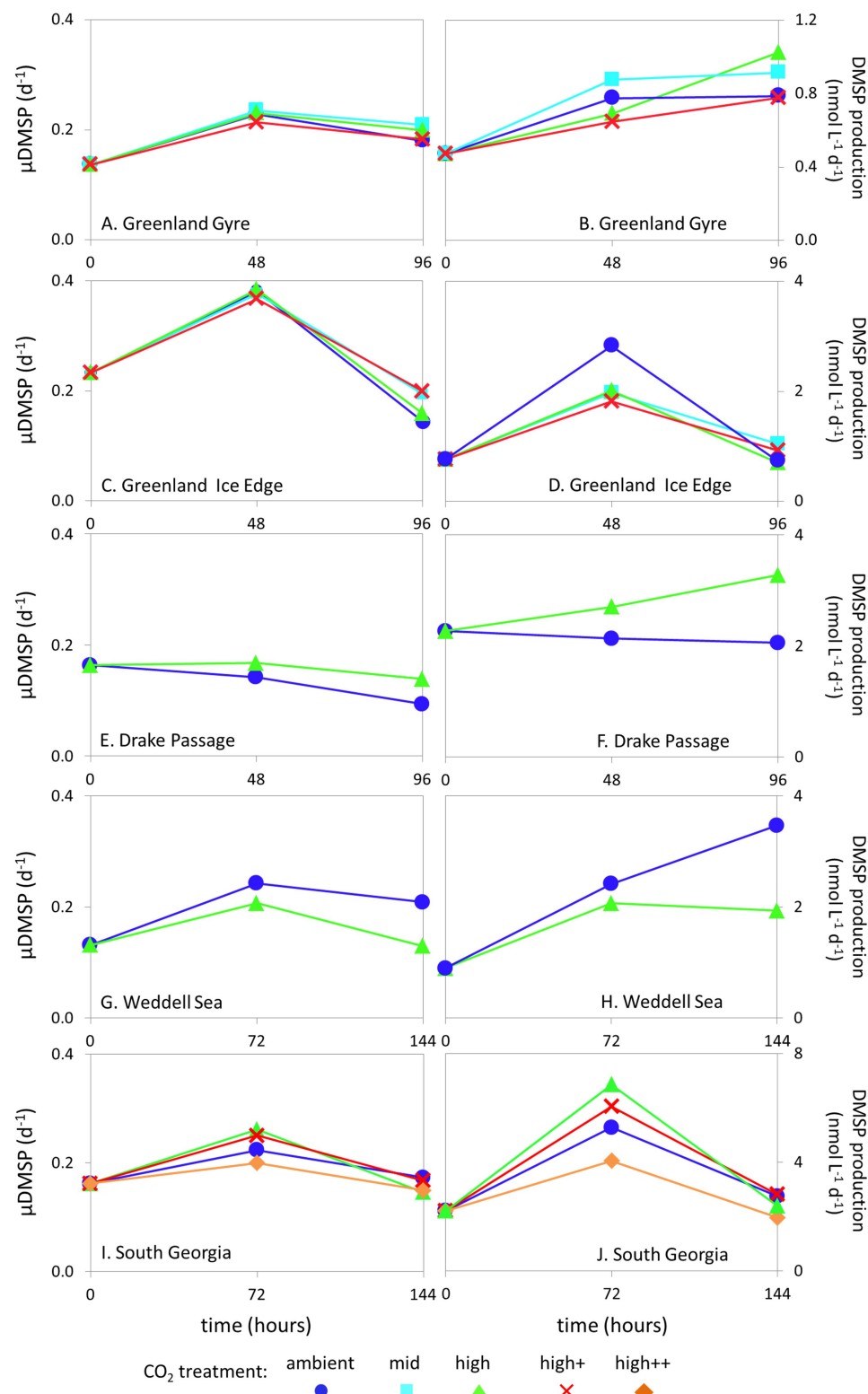

Figure 6. De novo synthesis of DMSP (μDMSP, d$^{-1}$) (left column) and DMSP production
rates (nmol L$^{-1}$ d$^{-1}$) (right column) for Arctic Ocean stations *Greenland Gyre* (A,B),
*Greenland Ice-edge* (C, D) and Southern Ocean stations *Drake Passage* (E, F), *Weddell Sea*
(G, H) and *South Georg*ia (I, J). No data is available for *Barents Sea* (Arctic Ocean) or *South*
*Sandwich* (Southern Ocean).

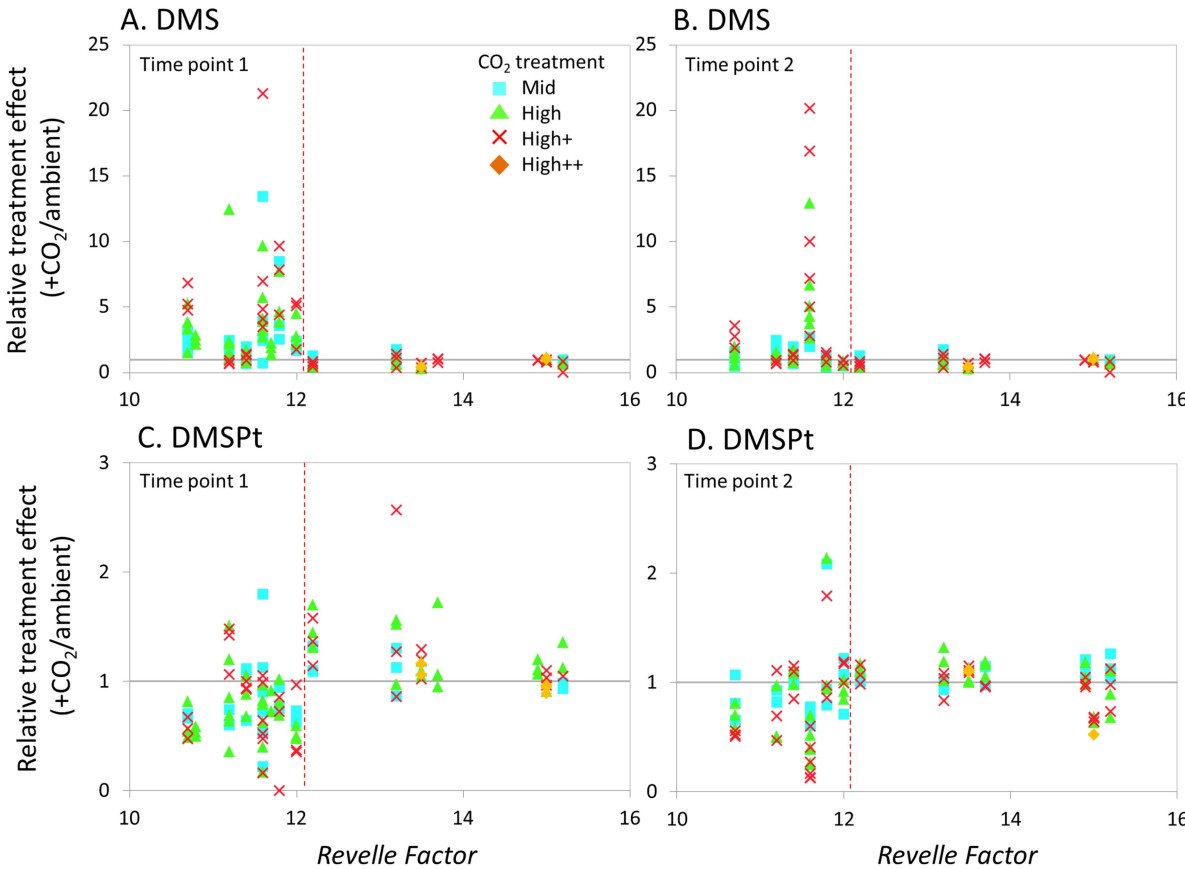


Figure 7. Relationship between Revelle Factor of the sampled water and the relative $CO_2$
treatment effect at ($[x]_{highCO2}/[x]_{ambientCO2}$) for concentrations of DMS at $T_1$ (A) and $T_2$ (B),
and for total DMSP concentrations at $T_1$ (C) and $T_2$ (D) for all microcosm experiments
performed in NW European waters, sub-Arctic and Arctic waters, and the Southern Ocean.
Grey solid line (= 1) indicates no effect of elevated $CO_2$. Revelle Factor > 12 = polar waters
(indicated by red dashed line). $T_1$ = 48 h, except for WS and SG (72 h) and SS (96 h). For
detailed analyses of the NW European shelf data, see Hopkins & Archer (2014).

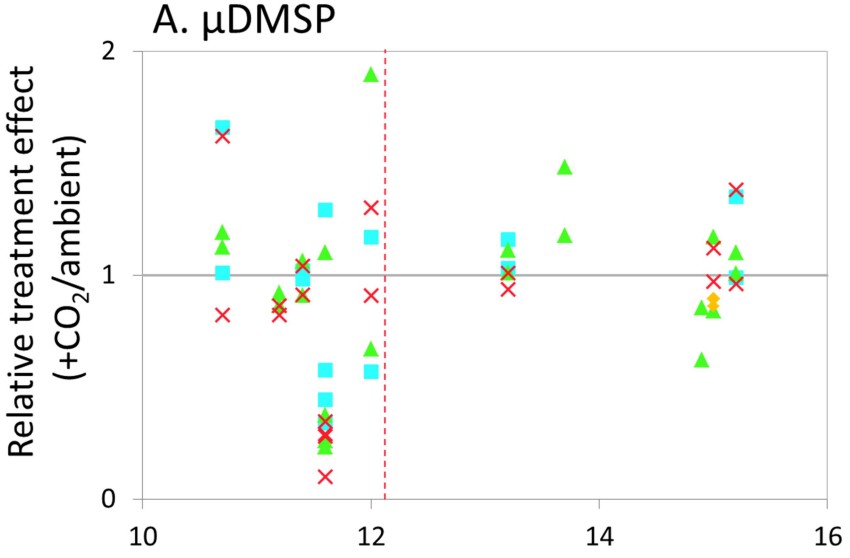

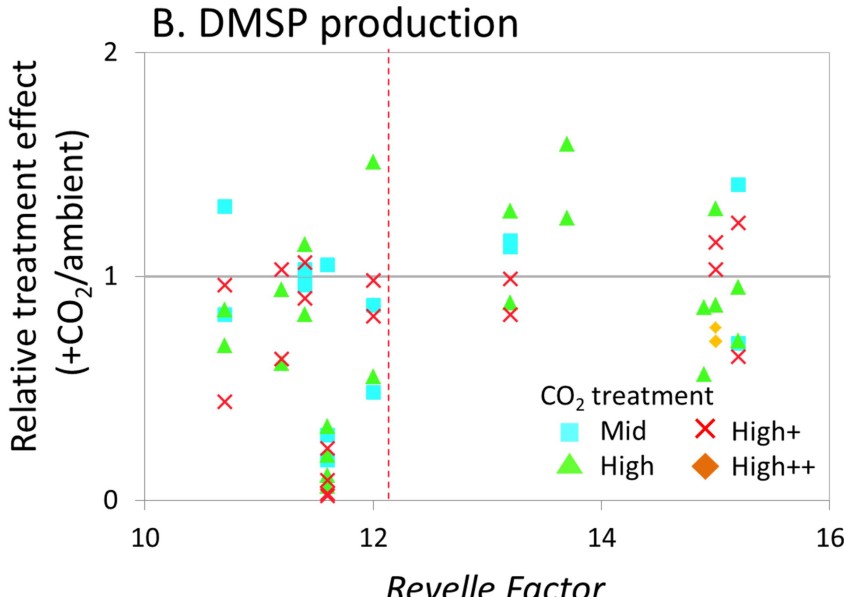

Figure 8. Relationship between the Revelle Factor of the sampled water and the relative $CO_2$ treatment effect at ($[x]_{highCO2}/[x]_{ambientCO2}$) for de novo DMSP synthesis ($\mu$DMSp, $d^{-1}$) at $T_1$ (A) and $T_2$ (B), and DMSP production rate (nmol $L^{-1}$ $d^{-1}$) at $T_1$ (C) and $T_2$ (D) for microcosm experiments performed in NW European waters, sub-Arctic and Arctic waters, and the Southern Ocean. Grey solid line (= 1) indicates no effect of elevated $CO_2$. Revelle Factor >12 = polar waters (indicated by red dashed line). $T_1$ = 48 h, T2 = 96 h, except for *Weddell Sea* and *South Georgia* (72 h, 144 h). For discussion of the NW European shelf data, see Hopkins & Archer (2014).

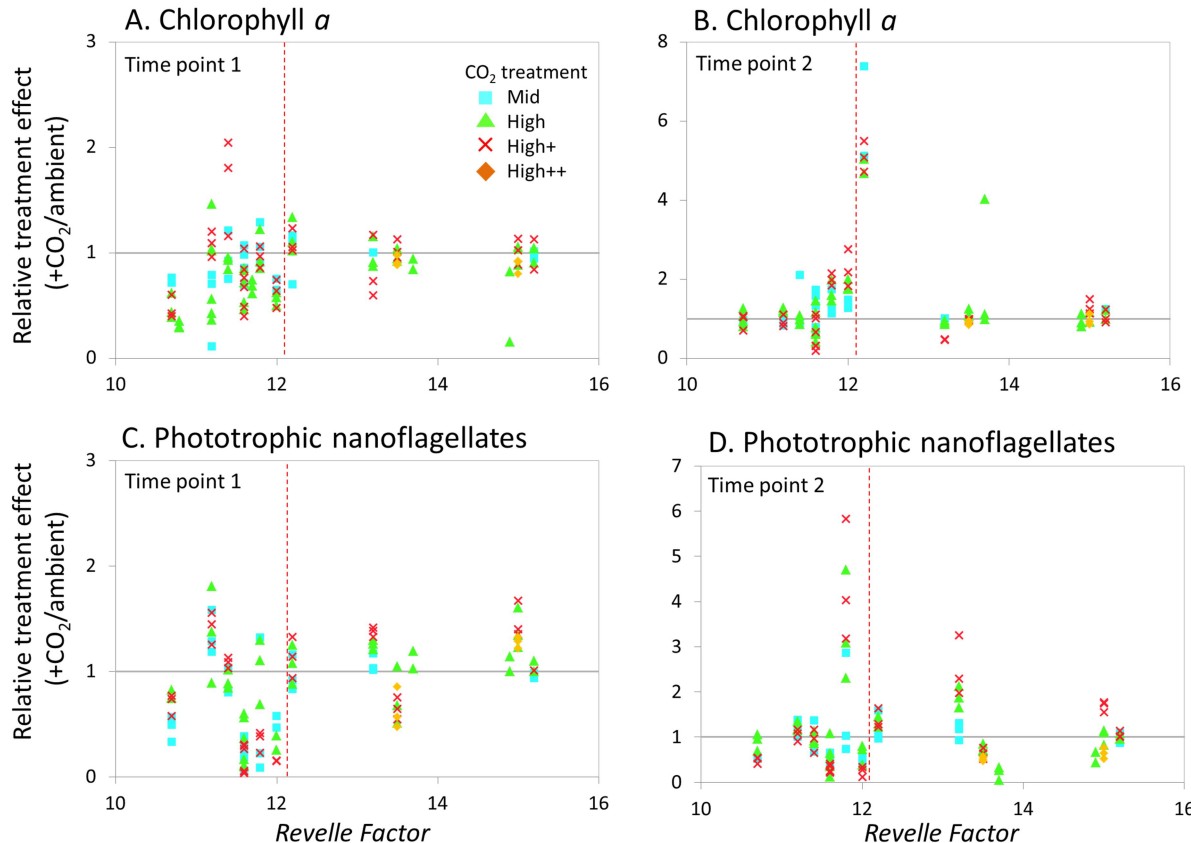

1274

Figure 9. Relationship between the Revelle Factor of the sampled water and the relative $CO_2$ treatment effect ($[x]_{highCO2}/[x]_{ambientCO2}$) for chlorophyll *a* concentrations at $T_1$ (A) and $T_2$ (B) and phototrophic nanoflagellate abundance at T1 (C) and T2 (D) for all microcosm experiments performed in NW European waters, sub-Arctic and Arctic waters, and the Southern Ocean. Grey solid line (= 1) indicates no effect of elevated $CO_2$. Revelle Factor >12 = polar waters (indicated by red dashed line). $T_1$ = 48 h, $T_2$ = 96 h, except for *Weddell Sea* and *South Georgia* (72 h, 144 h) and *South Sandwich* (96 h, 168 h).

1282

1283

1284