# Peer review of "A metanalysis of microcosm experiments shows that dimethyl"

_Biogeosciences, 2018_

## Referee Comment (RC1) · Anonymous Referee #1 · 20 Mar 2018

General comments The paper describes experimental results examining the response of DMS and DMSP concentration, synthesis & production rates to acidification in Southern Ocean & Arctic waters and compares them with previously published results from the NW European shelves. The authors report regional trends in responses, which they attribute to the variability in the carbonate system and its influence on the plasticity of the phytoplankton community and DMS/DMSP response. The analysis is somewhat limited to the carbonate system & phytoplankton size class without consideration of other factors. The paper is clear and well-written, and makes important points including regional variation in response to acidification, and also that different processes occurring at different scales are responsible for the variable responses reported over different timescales (as exemplified by comparison of microcosm versus

mesocosm responses). The paper is of publication standard if the points below relating to interpretation & analysis can be addressed.

Specific comments

Title The comparison with the NW European Shelf results is an important part of this paper and merits mention in the title

Introduction Line 122 – these microcosm experiments are not long enough to test adaptation. Results from experiments on timescales of < 1 week may give insight into plasticity and acclimation, but not "adaptive capacity"

Results Fig. 3 Error bars are relatively large at 96 hours in Arctic waters – this should be noted and discussed

Line 304-305; Fig 4c Error - "DMSP concentrations were found to DEcrease significantly in response to elevated CO2 AT 48 h for Barents Sea (Fig. 4 C)". Also note that DMSP was not significantly different at 96 hours.

Discussion 4.1 Regional differences in the response of DMS(P) to OA The interpretation of the treatment effects would benefit from statistical analysis to support the interpretation in:

Line 375-376 "De novo DMSP synthesis and DMSP production rates show a similar relationship with DIC/Alk (Fig. 7 A and B)" - is the difference between 0.91 > and < 0.91 significant? With the exclusion of one station (DIC/Alk ∼0.901) there looks to be no difference in Figure 7. Statistical confirmation required.

Line 379-380. "At T1, Chl a showed little response to elevated CO2 at polar stations, whereas a strong negative response was seen in temperate waters (Fig. 8A)" – again this description does not really match the data in the figure. The polar stations show a smaller range of treatment effect than temperate stations which show both larger positive and negative effects. Statistical confirmation required
Line 380-382. "A slight positive response in Chl a was seen at most temperate stations by T2, with generally little response at polar stations (Fig. 8 B)." Aren't the highest treatment effects at the polar stations? Statistical confirmation required

The analysis is limited to considering the carbonate system & phytoplankton size class as the factors determining regional response. Other factors will have differed between the polar waters and NW European shelf and may have influenced DMS/DMSP response to ocean acidification such as temperature, light, nutrients and phytoplankton community composition. For example, the authors mention "slower microbial metabolism at low water temperatures", so could this explain the observed difference in regional response? Datasets for these variables are most likely available, and a more comprehensive analysis that considers these would benefit the paper and interpretation. This may have already been carried out by the authors, in which case it should be noted that there are no relationships between response and these other variables.

A minor point here is that methodological differences should also be considering when assessing response. For example, different light cycles were used on different voyages.

They should also consider the degree to which the experimental manipulations alter the carbonate system relative to the ambient mean. The magnitude of change upon acid/base addition from the mean state of the carbonate system may be a more important factor than the regional range. For example, a proportionally larger shift in pH or carbonate upon acid/base addition may initiate a greater stress response and so DMS/DMSP production.

Lines 431-434: "In the following section, we explore the causes of this apparent resilience in terms of the environmental conditions to which the communities have presumably adapted." It should be noted that the variation in DIC/Alk reflects regional scale variation in single point measurements at each station (Line 362 "...the sampled waters"), and not the DIC/Alk variation at a particular site. Phytoplankton may experience greater or less variation at a single location on a temporal basis, which may

be a more important factor determining response. The role of temporal variation in determining response should be discussed.

Lines 451-457. The examples cited to support the authors contention that variability induces plasticity are from coastal waters and under ice, where greater variability would be expected. Would the variability be equally as large at the open ocean stations in this study?

Line 462-463 The authors mention the mean state here. Although the inclusion of the Tynan et al (2016) data is useful, this regional variation gives no indication of the local spatial & temporal variation that phytoplankton would experience at each station. The argument would be stronger if the responses were compared with mean local values for the carbonate system (from Tynan et al) for each station, which will to some extent, integrate temporal & spatial variability, rather than using just the values for the water sampled for the experiment (which I assume is what was done).

Technical corrections Line 55 chlorophyll-a maxima IN SURFACE WATERS Line 87 Sentence is a bit clunky Line 130-133 Shorten sentence Line 145 – Clarify that the Hopkins & Archer (2014) is from the NW European Shelf Line 256 – What does E1-E4/E5 refer to? Line 315 "Initial DMSP concentrations were higher AT THE SOUTH-ERN OCEAN STATIONS than for Arctic stations..." Line 317 "Net increases in DMSP occurred throughout, EXCEPT AT SOUTH GEORGIA..." Line 320 "the final time point at South Georgia (144 h) when a significantly LOWER DMSP with increasing $CO_2$ was observed" Line 350. As the results from the 4 unpublished NW European microcosm experiments are not presented in this paper, they should be identified as unpublished in Table 1 Line 365; Table 1 legend should identify that the polar stations are the two JR voyages excluding Station NS & IB.

---

## Referee Comment (RC2) · Anonymous Referee #2 · 12 Apr 2018

General comments:

Hopkins et al. present a large dataset on DMS(P) production by phytoplankton in short-term OA experiments from the Arctic, the Southern Ocean and the North Atlantic. This is an interesting and important dataset. I especially acknowledge the importance to publish 'negative results', i.e. absence of significant effects of experimental treatments, which is often neglected in OA research but should receive a lot more attention.

I find the hypothesis that then environmental history of organisms will determine their sensitivity to environmental change very convincing. Currently, the data (or its presentation) is not really suited to convincingly convey this message though. This does not mean that the hypothesis should not be mentioned, but it should be clearly marked as a hypothesis rather than a finding.

[Figure]

Furthermore, I would argue that the significant OA effects observed in the two cited coastal communities really question the validity of this hypothesis, as the degree of carbonate chemistry variability is much more pronounced in coastal vs. open ocean compared to temperate vs. polar. Therefore, your conclusions need to be more specific to the current study, and not towards polar systems in general.

One of my general methodological concerns that need to be addressed in the discussion is the fact that especially in short-term experiments, 50% variation in the experiment duration can have a huge impact on the outcome, especially if the phytoplankton initially show a lag phase as often observed in such experiments with natural communities. It makes a huge difference if OA effects are compared after 48h or 4d or 7d. While after 2 days, physiology most likely is not fully acclimated to the treatment conditions yet, 4d or 7d duration most likely show acclimated responses but potentially also reflect shifts in the composition of the communities. Also the differences in the number of hours at T1 and T2 should be accounted for by always referring to the number of hours rather than the time point throughout the manuscript.

It should also be included into the discussion that the significant impacts that Hussherr et al (2017) observed were measured over a much larger pCO2 range (up to 3000 $\mu$atm).

One major problem with this dataset is that the experimental carbonate chemistry was not well controlled. For example, at the 1000$\mu$atm pCO2 level, T2 pCO2 levels vary between approx. 400 and 1000$\mu$atm (Table S2). Therefore, the data should be represented using the real carbonate chemistry instead of the assigned values. I understand that this implies replotting and reanalysing most of the data, but currently the levels that are tested against each other are actually not separated when it comes to carbonate chemistry.

In conclusion, I get the impression that the authors really try to tell a story that does not fit their data. I think that the hypothesis (more variable carbonate chemistry causes

organisms to be less sensitive) presented here makes a lot of sense, but for various reasons the data set is not suited to prove or disprove it.

Specific comments:

Titel and throughout: To my knowledge, the term "resilience" refers to the ability of a system to return to the initial state after disturbance. Therefore, I do not think that the experimental setup and the response pattern (or its absence) in your study allows for statements on resilience. I suggest to use "insensitivity", "resistance" or something along these lines instead.

L22-27: As you refer to the other studies conducted in the Arctic, you also need to include their results in your statement, or be more specific that you only refer to the presented dataset and not the polar evidence in general.
L24-31: In the discussion, you do not refer to "geographical" or "regional" differences but compare temperate vs. polar systems. I would try to be more consistent here.

Introduction: The introduction is quite long, especially DMS(P) biogeochemistry is described in a lot of detail, even though most of this is not referred to in the discussion. I would suggest to shorten it. If your discussion does not focus at all on biogeochemistry, do you really need all this detail here?

L92-95: This is correct, but one shouldn't forget that it is the coastal areas that are the most productive and therefore important ones. In my opinion you do not even have to somehow restrict the importance of these two previous studies, your study is a valuable contribution even though two other ones exist.

L118: Here and in a few other instances you refer to your incubations as being "identical", but in the methods you state that the day length was adapted to the respective in situ conditions. Therefore, I would not use the term "identical".

L119-120: I think the differences in nutrients and incubation temperatures play a big role in understanding the results, so they need to be shown in one of the tables. Refer-

ring to a paper under review is not sufficient for such important information. Generally, the authors should provide all relevant information (at least in the supplement) if the other manuscript is not publically available yet.

L122-125, L130: While I do agree that differences in environmental variability most likely have an impact on the adaptive capacity of communities, you cannot estimate this adaptive capacity in short-term incubation experiments that run for several days only.

L229-231: I am wondering if it wouldn't make more sense to normalize DMSP concentrations to biomass? This is especially the case if you want to test for "stress-induced algal processes" (L135-136) rather than biomass-dependent effects.

L252-259: I do not think that you can infer growth rates from the Chla measurements, given that there was probably strong photoacclimatory processes happening in response to the change in light fields (naturally varying to constantly high). You do not really need these rates for your story, so I suggest to omit this parameter all together, i.e. also from results and discussion.

L278: The results from the Atlantic experiments are used a lot in the discussion, they should therefore also be included in the results (and methods), especially but not exclusively the previously unpublished ones.

L284-287: Methods are missing for the nanoflagellate and bacteria abundances data,

L291: Methods for irradiance measurements are missing

L314: This is important information that really helps your line of argument, I would therefore put stronger emphasis on this in the discussion.

L328-335: This comparison of standing stocks is highly dependent on the time of sampling. You therefore need to include information about and discussion on the timing of sampling relative to bloom phenology. I.e. if the Arctic and Southern Ocean samples were taken in (macro and/or micro) nutrient depleted waters after a bloom, can you re-
ally make such general statements on polar vs. temperate waters? Was the temperate sampling also conducted in similar phases of biomass dynamics? If not, you have a problematic bias towards low productivity in the polar samples that needs to be taken into account.

L340-342: This in a strong indication for the importance of other drivers (nutrients, species composition, . . .). You need to show these and check whether there are significant effects here.

L360ff: I really like this way of presenting the data. You should, however, also show the same plot with pCO2 instead of TA/DIC for comparison because I do not agree with you that this ratio gives a full overview of the in situ carbonate chemistry.

L372 and throughout the entire manuscript: Report the time points in days or hours instead of T1, T2 etc. because this is not consistently the same time point as well as for better readability and consistency throughout the text.

L377-282: This strongly suggests that, due to temperature-driven differences in metabolic rates and their effect son how fast the communities can acclimate to changed conditions, the experiments emerge out of measurement noise at different times.

Discussion: A discussion of stress vs. acclimated response is missing

L399: Everything until here reads more like results than like a discussion section. Please consider rearranging.

L410-412: The authors seem to imply that CO2 sensitivity is only occurring in form of negative effects, even though there are many studies that show beneficial effects of increased substrate availability for photosynthesis, which is particularly true for picoeukaryotes (e.g. Schulz et al. 2017). Please take this aspect into account.

L436-439: I do not agree that your data really shows this: Figure 9 indicates the Arctic Ocean carbonate chemistry to be actually more similar to the Atlantic than to the Southern Ocean.

L444-448: Such a comparison only makes sense if the same geographical and temporal ranges, and phases of biomass cycle (pre-bloom/ bloom/post-bloom, before/after winter convection etc.) were covered in the different study areas. Please clarify if this was the case.

L451-455: In the Southern Ocean, several studies have shown strong OA-effects on species composition (e.g. Tortell et al. 2008, Feng et al. 2010, Hoppe et al. 2013, Trimborn et al. 2017).

L455-457: Similarly, you are missing previous work done in the Arctic (Coello-Camba et al. 2014, Holding et al. 2015, Thoisen et al. 2015, Hoppe et al. 2017a,b) that need to be considered.

L460: n=3 is not "highly" replicated

L469: Why are you comparing your data in detail with Archer et al. (2013) but not Hussherr et al. (2017)?

L475: I would rather refer to the most common not the maximum duration.

L482-488: Is this difference really due to different sensitivities, or differences in biological rates, that lead to the fact that small physiological changes are detectable at different time points?

L515-521: You first imply that the short duration of the experiments would render changes in species composition rather unlikely, but then you report one case where you indeed observed changes. I would say that this indicates that the timescales in general would have allowed for changes in composition also in the other experiments.

L543-550: I agree that it is an interesting finding that coastal DMS production seems to be more sensitive to OA than that from the open ocean. This finding does, however, really hint against the proposed mechanisms of insensitivity, because coastal systems are a lot more variable in carbonate chemistry compared to the open ocean (e.g. Thoisen et al. 2015). Thus, the interpretation of and conclusions from the dataset

have to be reassessed.

Figures 3, 4, 5, 7, 8, S3: Given the lack of control in carbonate chemistry in many experiments (Table S2), this representation is misleading. The data needs to be presented accounted for the real carbonate chemistry in the incubations.

Technical corrections:

L11: I suggest replacing "we increase" by "to increase"

L12: I suggest referring explicitly to climate change instead of environmental change. Otherwise, the step to OA is kind of abrupt.

L28: Do you really mean "region may vary in response to OA" or rather "region may vary in their response to OA"?

L190: replace "made" by "taken"

L207: omit "all" as in the caption of figure 5 you state that these data are not available for for two of the stations.

L237-238: According to the Journal style, it would be $A_T$ and $C_T$ for total alkalinity and total dissolved inorganic carbon, respectively

L372: Omit "identical" as irradiances and temperatures were not the same

L497-500: Something does not see correct in this sentence, please rephrase

L532: Insert "low and" between "periods of" and "stable productivity"

L539: "is insensitive to OA during multiple short term microcosm" instead of "is resilient to OA during multiple, highly replicated short term microcosm"

L542: add additional references mentioned above

L559: Replace "results from our study indicate" by a more honest "we hypothesise" or something similar.

Table 1: Add macro nutrient (at least NO3) levels and incubation temperatures (will be more variable than in situ). Also "Comment" should read "Reference". Shouldn't "Sample depth" read "Sampling depth"?

All Figures: Please indicate number of replicates and type of error estimate in the caption

Figure 2: Replace "$\mu$E m-2 s-1" by "$\mu$mol photons m-2 s-1" or "$\mu$mol quanta m-2 s-1" in figure and caption. Also, the panels are so close together that the top and bottom axis descriptions get messy, please move them apart a bit.

[Figure]

---

## Author Comment (AC1) · 21 Jun 2018

**Final author comments on "Dimethylsulfide (DMS) production in polar ocean may be resilient to ocean acidification" by F.E. Hopkins et al., manuscript number bg-2018-55**

We thank both anonymous reviewers for their detailed, constructive, and positive reviews of our manuscript – we greatly appreciate the care and detail that has gone into its assessment. Below we respond to their comments point-by-point. It is worth noting that the related paper by Richier et al. (2018) has now been accepted for publication in Global Change Biology (doi: 10.1111/gcb.14324), providing further substantiation of many of the discussion points in this manuscript. The reviewers comments are shown in italics, with our responses shown in bold. Line numbers in our response refer to the revised version.

1. **Response to Anonymous Referee #1**

**General comments**

*1.1 The paper describes experimental results examining the response of DMS and DMSP concentration, synthesis & production rates to acidification in Southern Ocean & Arctic waters and compares them with previously published results from the NW European shelves. The authors report regional trends in responses, which they attribute to the variability in the carbonate system and its influence on the plasticity of the phytoplankton community and DMS/DMSP response. The analysis is somewhat limited to the carbonate system & phytoplankton size class without consideration of other factors. The paper is clear and well-written, and makes important points including regional variation in response to acidification, and also that different processes occurring at different scales are responsible for the variable responses reported over different timescales (as exemplified by comparison of microcosm versus mesocosm responses). The paper is of publication standard if the points below relating to interpretation & analysis can be addressed.*

**We thank the reviewer for their positive view of our paper, and we are confident that we have addressed their concerns.**

**Specific comments**

*1.2 Title The comparison with the NW European Shelf results is an important part of this paper and merits mention in the title*

**The title has been changed to:**

**"Dimethylsulfide (DMS) production in polar oceans may be insensitive to ocean acidification: a meta-analysis of 18 microcosm experiments from temperate to polar waters".**

**'resilient' has been replaced with 'insensitive' at the suggestion of reviewer #2, see point 2.8 below.**

*1.3 Introduction Line 122–these microcosm experiments are not long enough to test adaptation. Results from experiments on timescales of < 1 week may give insight into plasticity and acclimation, but not "adaptive capacity"*

We acknowledge this point while further noting that a key point we emphasise is that we interpret the variable sensitivity to short term experimental conditions observed as potentially corresponding to pre-existing adaptation to prevailing conditions across sampled populations/systems. Text has been altered accordingly in three places (line numbers correspond to revised version):

L124: "The focus is then on the effect of short-term $CO_2$ exposure on physiological processes, as well as the extent of the variability in acclimation between communities".

L126: "The capacity of organisms to acclimate to changing environmental conditions contributes to the level of resilience of key ecosystem functions, such as DMS production".

L132: "…our approach can provide insight into the physiological response of a variety of polar surface ocean communities, as well as their potential level of sensitivity to future OA when compared between environments that differ in carbonate chemistry…".

*1.4 Results Fig. 3 Error bars are relatively large at 96 hours in Arctic waters – this should be noted and discussed*

We feel that this observation does not add any additional insights to the results or discussion so we have simply added into the results text:

L331: "Increased variability between triplicate incubations became apparent in all three Arctic experiments by 96 h but no significant effects of elevated $CO_2$ on DMS concentrations were observed".

*1.5 Line 304-305; Fig 4c Error - "DMSP concentrations were found to DEcrease significantly in response to elevated CO2 AT 48 h for Barents Sea (Fig. 4 C)". Also note that DMSP was not significantly different at 96 hours.*

Text has been altered accordingly.

*1.6 Discussion 4.1 Regional differences in the response of DMS(P) to OA The interpretation of the treatment effects would benefit from statistical analysis to support the interpretation in:*

*Line 375-376 "De novo DMSP synthesis and DMSP production rates show a similar relationship with DIC/Alk (Fig. 7 A and B)" - is the difference between 0.91 > and < 0.91 significant? With the exclusion of one station (DIC/Alk ~0.901) there looks to be no difference in Figure 7. Statistical confirmation required.*

Statistics have been confirmed and the text now reads:

"*De novo* DMSP synthesis and DMSP production rates show a similar relationship with $C_T/A_T$ (Fig. 7 A and B), with a significant suppression of DMSP production rates in temperate waters compared to polar waters (Fig. 7B, Kruskal-Wallis One Way ANOVA *H* = 8.711, *df* = 1, *p* = 0.003). Although a similar trend was seen for *de novo* DMSP synthesis, the difference between temperate and polar waters was not statistically significant (Fig. 7A)".

*Line 379-380. "At T1, Chl a showed little response to elevated CO2 at polar stations, whereas a strong negative response was seen in temperate waters (Fig. 8A)" – again this description does not really match the data in the figure. The polar stations show a smaller range of treatment effect than temperate stations which show both larger positive and negative effects. Statistical confirmation required*

*Line 380-382. "A slight positive response in Chl a was seen at most temperate stations by T2, with generally little response at polar stations (Fig. 8 B)." Aren't the highest treatment effects at the polar stations? Statistical confirmation required*

**Statistics have been confirmed and the text now reads:**

**"At $T_1$, a statistically significant difference in response was seen between temperate and polar waters for Chl *a* (Kruskal-Wallis One Way ANOVA *H* = 20.577, *df* = 1, *p*<0.001), with minimal response to elevated $CO_2$ at polar stations, and in general a strong negative response was seen in temperate waters (Fig. 8A). By $T_2$, no significant difference in response of Chl *a* between temperate and polar waters was detectable (Fig. 8B), although a slight positive response in Chl *a* was seen at some temperate stations, and polar stations showed a minimal response, with the exception of *Barents Sea* which saw strongly enhanced Chl *a* at $T_2$ (Fig. 8 B)".**

*1.7 The analysis is limited to considering the carbonate system & phytoplankton size class as the factors determining regional response. Other factors will have differed between the polar waters and NW European shelf and may have influenced DMS/DMSP response to ocean acidification such as temperature, light, nutrients and phytoplankton community composition. For example, the authors mention "slower microbial metabolism at low water temperatures", so could this explain the observed difference in regional response? Datasets for these variables are most likely available, and a more comprehensive analysis that considers these would benefit the paper and interpretation. This may have already been carried out by the authors, in which case it should be noted that there are no relationships between response and these other variables.*

**Richier et al. (2018) provide a detailed overview of the role of other potential environmental drivers for the differences in response between temperate and polar waters. This paper has now been accepted for publication in Global Change Biology, so we will refer the reader to it for more detailed analysis of this issue. To address the reviewers concerns, we have also added an appropriate amount of discussion relating to this to the current manuscript.**

**Firstly, to address the reviewers comment on "slower microbial metabolism at low temperature waters": we failed to observe strong responses to high $CO_2$ in experiments performed in Arctic waters (cruise JR271), therefore incubation times during the Southern Ocean experiments (cruise JR274) were increased from 4 to 6 to 8 days, whilst including a higher $CO_2$ treatment (2000 μatm). However, the magnitudes of the responses were found to be independent of overall experimental duration. As Richier et al. (2018) explain, net growth rates may be expected to be 2- to 3 –fold higher in the warmest compared to the coldest waters (Eppley, 1972), and indeed maximum Chl a-normalised photosynthesis rates were indeed 2- to 3-fold higher in polar waters – but the response to experimentally-induced OA in polar waters remained insignificant despite the length of incubation being up to twice the duration of temperate experiments, and up to 4x longer, than the 48 h time point where strong responses were typically observed in temperate waters.**

**We have added the following to the methods section at L210:**

**"The magnitude of response was not related to incubation times, and expected differences in net growth rates (2- to 3-fold higher in temperate compared to polar waters (Eppley 1972)) did not account for the differences in response magnitude despite the increased incubation time in polar waters (see Richier et al. (2018) for detailed discussion)".**

**Secondly, to address the reviewers concerns regarding "other factors [that] will have differed between polar waters and NW European shelf…" that may have influenced the response, we have added the following to the Discussion:**

**L440: "Across all experiments, the response of net total community Chl *a* and net growth rates of small phytoplankton (<10 μm) scaled with $pCO_2$ treatment, and strongly correlated with in situ carbonate chemistry, whilst no relationships were found with any of the other wide range of initial physical, chemical or biological variables (Richier et al. 2018). Overall, the observed differences in regional response to carbonate chemistry manipulation could not be attributed to any other measured factor that varied systematically between temperate and polar waters. These include ambient nutrient concentrations, which varied considerably but where direct manipulation had no influence on the response, and initial community structure, which was not a significant predictor of the response (Richier et al. 2018)".**

*1.8 A minor point here is that methodological differences should also be considering when assessing response. For example, different light cycles were used on different voyages.*

**Different light cycles were used on different voyages to simulate the *in situ* light conditions/light:dark cycles for the time of year of sampling. We don't expect this to have affected the response to OA, and it would have been inappropriate to have used the same light cycle on all cruises.**

*1.9 They should also consider the degree to which the experimental manipulations alter the carbonate system relative to the ambient mean. The magnitude of change upon acid/base addition from the mean state of the carbonate system may be a more important factor than the regional range. For example, a proportionally larger shift in pH or carbonate upon acid/base addition may initiate a greater stress response and so DMS/DMSP production.*

**Ambient $pCO_2$ and pH for each sampling station are shown in Table 1. A large range of ambient carbonate chemistry was observed across all waters, with no consistent relationship with location (e.g. $pCO_2$: NW Euro shelf 287 – 400 uatm, Arctic 289 – 304 uatm, Southern Ocean 272 – 510 uatm). Experimental manipulation of carbonate chemistry was accurately calculated and implemented using the CO₂SYS programme (see Richier et al. 2018 for methods, and Richier et al. 2014 Figure 3 for a comparison of target vs actual $pCO_2$ following experimental manipulation for NW European shelf experiments). Our experiments provided no apparent evidence of a relationship between the proportional shift in pH/carbonate chemistry and the magnitude of the response as a function of the initial state of the carbonate system, rather, as emphasised, the presence or absence of any observable response broadly correlated with the initial state of the carbonate system (see also Richier et al. 2018), with subsequent response strength then scaling with magnitude of manipulation in those experiments where any response could be observed.**

*1.10    Lines 431-434: "In the following section, we explore the causes of this apparent resilience in terms of the environmental conditions to which the communities have presumably adapted." It should be noted that the variation in DIC/Alk reflects regional scale variation in single point measurements at each station (Line 362 "...the sampled waters"), and not the DIC/Alk variation at a particular site. Phytoplankton may experience greater or less variation at a single location on a temporal basis, which may be a more important factor determining response. The role of temporal variation in determining response should be discussed.*

**We agree with the reviewer that variation of carbonate chemistry on a temporal basis may be important in determining the response to experimental OA, although, as discussed in Richier et al. (2018), we also note that it is the range of variability experienced at the cellular level over generation timescales (i.e. days) which is likely to be the most important. However, given the lack of temporal data for each sampling station we acknowledge that we cannot make definitive statements in this regard. We have acknowledged in the text that the amount of variation in carbonate chemistry experienced by plankton communities at a single location on a temporal basis should also be considered:**

**L495 onwards: "Although it might be expected that carbonate system variability on the level 'experienced' by the cells, i.e. ~daily cellular level variability, might be the most important factor driving sensitivity (Flynn et al. 2012; Richier et al. 2018), our data represent only a snapshot (4 – 6 weeks) of a year, and thus do not contain information on the range in variability over seasonal cycles. For comparison with Arctic stations, Hagens and Middelburg (2016) report a seasonal pH variability of up to 0.25 units from a single site in the open ocean surface waters in the Iceland Sea, whilst Kapsenberg et al. (2015) report an annual variability of 0.3 – 0.4 units in the McMurdo Sound, Antarctica. This implies that both polar open ocean and coastal/sea ice locations experience equally large variations in carbonate chemistry over seasonal cycles. In open ocean waters this is driven by enhanced drawdown of DIC and $CO_2$ during the productive spring and summer months, countered by lower productivity and strong mixing in the winter (Hagens and Middelburg 2016). In coastal and sea-ice affected regions, seasonal pH variability may be enhanced further by tidal exchanges, and by dilution of $T_C/T_A$ caused by sea-ice melt (Kapsenberg et al. 2015)."**

*1.11    Lines 451-457. The examples cited to support the authors contention that variability induces plasticity are from coastal waters and under ice, where greater variability would be expected. Would the variability be equally as large at the open ocean stations in this study?*

**The carbonate chemistry/pH variability may be as large in the open waters of the polar oceans, as in coastal/sea-ice waters. The open waters of the polar oceans experience high levels of productivity during the spring and summer, and given that these waters are less well-buffered with respect to $CO_2$ uptake, this will lead to a greater range of both short-term cellular scale variability (Flynn et al. 2012; Richier et al. 2018) and seasonal carbonate chemistry characteristics in these waters (Sabine et al. 2004, Orr et al. 2005). Hagen and Middleburg (2016) assess the factors that control seasonal pH variability in surface waters, using a station in the open waters of the Iceland Sea as one of their examples. At this site, pH is shown to vary by up to 0.25 pH units over a seasonal cycle. This is due to a strong drawdown of DIC and $pCO_2$ during the productive spring months, and a rise in DIC and $pCO_2$ in winter as a result of reduced productivity and strong**

mixing. This is of a similar magnitude to seasonal pH variability at a coastal, sea-ice dominated site in the Antarctic (McMurdo Sound, Kapsenberg et al. 2015) where pH variability of 0.3 – 0.4 units is observed – and this range is among the greatest observed in the ocean. Therefore, it is reasonable to assume that the seasonal variability at open ocean polar stations may be of a similar order to the variability observed in coastal/sea-ice stations.

The text has been edited and extended to take into account the information detailed above as for point 1.10.

*Line 462-463 The authors mention the mean state here. Although the inclusion of the Tynan et al (2016) data is useful, this regional variation gives no indication of the local spatial & temporal variation that phytoplankton would experience at each station. The argument would be stronger if the responses were compared with mean local values for the carbonate system (from Tynan et al) for each station, which will to some extent, integrate temporal & spatial variability, rather than using just the values for the water sampled for the experiment (which I assume is what was done).*

The DIC/Alk values used for comparison of the response magnitude are representative of the "mean local values" for each station, and are also "the water sampled for the experiment". We are unclear what the reviewer would like to see here. Table 1 shows the $pCO_2$ and pH for each sampling station and the DIC/Alk was derived from the other measurements of the carbonate system made from these same samples. We do not believe these values integrate temporal and spatial variability as they are discrete measurements made on the water sampled for the incubations. Furthermore community composition is also transient, such that the phytoplankton that were present in the sampled water would not necessarily have been present all the time/everywhere. As emphasised above, our overall hypothesis is that cellular level variability is likely to be the most significant driver of local adaption of communities and hence sensitivity to manipulative forcing (Flynn et al. 2012; Richier et al. 2018).

**Technical corrections**

*1.12    Line 55 chlorophyll-a maxima IN SURFACE WATERS*

Text changed accordingly, now reads "…and have been linked to chlorophyll *a* maxima in surface waters and the presence of aerosols formed from DMS oxidation products such as methanesulfonate (MSA)".

*1.13    Line 87 Sentence is a bit clunky*

Sentence has been re-worded and now reads: "OA is occurring at a rate not seen on Earth for 300 Ma, and so the potential effects on marine organisms, communities and ecosystems could be wide-ranging and severe".

*1.14    Line 130-133 Shorten sentence*

The sentence has been shortened and now reads: "Nevertheless, our approach can provide insight into the physiological response and level of acclimation to future OA of a variety of polar surface ocean communities adapted to different in situ carbonate chemistry environments (Stillman and Paganini 2015), alongside the implications this may have for DMS production".

*1.15    Line 145 – Clarify that the Hopkins & Archer (2014) is from the NW European Shelf*

**Text changed accordingly, now reads: "Data are combined with the results from an earlier study on board the RRS Discovery (D366) described in Hopkins & Archer (2014) performed in the temperate waters of the NW European shelf".**

*1.16    Line 256 – What does E1E4/E5 refer to?*

**We assume the reviewer is referring to Line 265. E1-E4/E5 describes the experiment identifiers for the polar cruises - but incorrectly - so thank you for pointing this out. The text now refers the reader to Table 1 for station identifiers.**

*1.17    Line 315 "Initial DMSP concentrations were higher AT THE SOUTHERN OCEAN STATIONS than for Arctic stations…"*

**Text changed accordingly, now reads: "Initial DMSP concentrations were higher at the Southern Ocean stations than for Arctic stations,…"**

*1.18    Line 317 "Net increases in DMSP occurred throughout, EXCEPT AT SOUTH GEORGIA…"*

**Text changed accordingly, now reads: "Net increases in DMSP occurred throughout, except at South Georgia, and were on the order of between <10 nmol L$^{-1}$ - >30 nmol L$^{-1}$ over the course of the incubations".**

*1.19    Line 320 "the final time point at South Georgia (144 h) when a significantly LOWER DMSP with increasing CO2 was observed"*

**Text changed accordingly, now reads: "Concentrations were not generally $p$CO$_2$-treatment dependent with the exception of the final time point at *South Georgia* (144 h) when a significantly lower DMSP with increasing CO$_2$ was observed…"**

*1.20    Line 350. As the results from the 4 unpublished NW European microcosm experiments are not presented in this paper, they should be identified as unpublished in Table 1*

**As stated in the text that the reviewer refers to, the data from the 4 previously unpublished NW European microcosm experiments are included in the meta-analysis in this paper (Figures 6, 7, 8), so it is reasonable to identify them as "*This study*" in Table 1.**

*Line 365; Table 1 legend should identify that the polar stations are the two JR voyages excluding Station NS & IB.*

**Text added to Table 1 legend: "All polar stations were sampled for JR271 and JR274, with the exception of NS and IB".**

1.    **Response to Anonymous Referee #2**

**General comments**

*2.1 Hopkins et al. Present a large dataset on DMS(P) production by phytoplankton in short term OA experiments from the Arctic, the Southern Ocean and the North Atlantic. This is an interesting*

*and important dataset. I especially acknowledge the importance to publish 'negative results', i.e. absence of significant effects of experimental treatments, which is often neglected in OA research but should receive a lot more attention.*

**We thank the reviewer for their positive view of our work, and also agree that it is important to publish 'negative results' to give the full picture on the effects of ocean acidification.**

*2.2 I find the hypothesis that then environmental history of organisms will determine their sensitivity to environmental change very convincing. Currently, the data (or its presentation) is not really suited to convincingly convey this message though. This does not mean that the hypothesis should not be mentioned, but it should be clearly marked as a hypothesis rather than a finding.*

**We have changed "suggest" to "hypothesise" on L24.**

**We have changed "Our findings support the notion that,.." to "This supports our hypothesis that…" on L524.**

**We have changed "However, results from our study indicate that the DMS response to OA…", to "However, we hypothesise that the DMS response to OA…" on L657.**

*2.3 Furthermore, I would argue that the significant OA effects observed in the two cited coastal communities really question the validity of this hypothesis, as the degree of carbonate chemistry variability is much more pronounced in coastal vs. open ocean compared to temperate vs. polar. Therefore, your conclusions need to be more specific to the current study, and not towards polar systems in general.*

**We would argue that pH variability over seasonal time scales in the open ocean polar waters is comparable to coastal/sea ice dominated areas. Additionally, as noted elsewhere in our responses, we hypothesise that cellular scale variations are likely to be the most relevant (Flynn et al. 2012; Richier et al. 2018).**

**See response to reviewer #1, point 1.13.**

*2.4 One of my general methodological concerns that need to be addressed in the discussion is the fact that especially in short-term experiments, 50% variation in the experiment duration can have a huge impact on the outcome, especially if the phytoplankton initially show a lag phase as often observed in such experiments with natural communities. It makes a huge difference if OA effects are compared after 48h or 4d or 7d. While after 2 days, physiology most likely is not fully acclimated to the treatment conditions yet, 4d or 7d duration most likely show acclimated responses but potentially also reflect shifts in the composition of the communities. Also the differences in the number of hours at T1 and T2 should be accounted for by always referring to the number of hours rather than the time point throughout the manuscript.*

**For the NW European shelf cruise and the Arctic Cruise, all experiments were 96 h in duration, with samples taken at 0 h, 48 h and 96 h. As we failed to observe strong responses within experiments performed in the Arctic, incubation times were increased for a subset of experiments on the Southern Ocean cruise. This was to investigate whether the lack of strong response in Artic waters was related to slower microbial metabolism in the low temperature waters. To address**

this, three experiments of increasingly longer duration from 96 h to 144 h to 168 h were performed (Weddell Sea, South Georgia, South Sandwich), and with the inclusion of a higher target pCO2 of 2000 μatm. However, the magnitude of the response in both biological and DMS-related variables were found to be independent of the experimental duration. See also Richier et al. (2018). This is evidenced by the effect of $CO_2$ treatment on net growth rates. Net growth rates may be expected to be 2- to 3 –fold higher in the warmest compared to the coldest waters (Eppley, 1972), and indeed maximum Chl a-normalised photosynthesis rates were 2- to 3-fold higher in polar waters – but the response to experimentally-induced OA in polar waters remained insignificant despite the length of incubation being up to twice the duration of temperate experiments, and up to 4x longer, than the 48 h time point where strong responses were typically observed in temperate waters.

The following has been added at L210:

"The magnitude of response was not related to incubation times, and expected differences in net growth rates (2- to 3-fold higher in temperate compared to polar waters (Eppley 1972)) did not account for the differences in response magnitude despite the increased incubation time in polar waters (see Richier et al. (2018) for detailed discussion)".

*2.5 It should also be included into the discussion that the significant impacts that Hussherr et al (2017) observed were measured over a much larger pCO2 range (up to 3000 μatm).*

We have added to the text at L606:

"It should be noted that this response was seen over a range of $pCO_2$ from 500 - 3000 μatm, far beyond the levels used in the present study".

We do not feel this warrants any further discussion, as using a gradient of $CO_2$ treatment levels is an accepted and useful technique in ocean acidification experiments. It allows the use of regression statistics for assessment of possible $CO_2$ effects, and increases the chances of detecting any threshold level for $CO_2$/pH sensitive processes (Riebesell et al., 2013, Biogeosciences, 10, 1835–1847).

*2.6 One major problem with this dataset is that the experimental carbonate chemistry was not well controlled. For example, at the 1000μatm pCO2 level, T2 pCO2 levels vary between approx. 400 and 1000μatm (Table S2). Therefore, the data should be represented using the real carbonate chemistry instead of the assigned values. I understand that this implies replotting and reanalysing most of the data, but currently the levels that are tested against each other are actually not separated when it comes to carbonate chemistry.*

Although we acknowledge that our approach of allowing the carbonate system to vary as a result of biological activity necessitated that some drift occurred following initial imposed conditions, we would argue that our plotting/presentation of the data remains appropriate – and would not agree that this is a "major problem". In Figures 3, 4, 5, 6, 7 and 8 the legend clearly states the $CO_2$ values shown are "nominal", meaning they are representative of the initial target $CO_2$ treatments used for each experiment. This is simpler and clearer than showing a complicated range of values for each individual plot and experiment. However, we do concede that we could have presented

the actual treatment levels somewhere in the original submission, and we have now done so, with the addition of a table to the supplementary information (see below).

For the NW European shelf cruise, a comprehensive comparison of the accuracy and precision of the carbonate chemistry manipulation method, as well as an analysis of 'actual' vs 'target' $pCO_2$ values and the variability in the former over the experimental duration has already been presented in Figure 3, Richier et al. (2014), demonstrating that the achieved $pCO_2$ levels were well matched to target values at $T_0$ for E01 – E05, whilst acknowledging that differences in $pCO_2$ between target and initial values were more pronounced in the higher-$pCO_2$ treatments, a reflection of the lower buffer capacity of the carbonate system at higher $pCO_2$.

 We have added the following to the methods section (L181 onwards):

"Full details of the carbonate chemistry manipulations can be found in Richier et al. (2014) and Richier et al. (2018). Broadly, achieved $pCO_2$ levels were well-matched to target values at $T_0$, although differences in $pCO_2$ between target and initial values were greater in the higher $pCO_2$ treatments, due to lowered carbonate system buffer capacity at higher $pCO_2$. For all 18 experiments, actual attained $pCO_2$ values were on average around 89% ± 12% (± 1 SD) of target values. The attained $pCO_2$ values are presented in Table S1 on the Supplementary Information. For simplicity, experimental data is presented against its target ('nominal') $pCO_2$ treatment throughout the paper".

And we have added this table to the Supplementary Information:

Table S1. Summary of $pCO_2$ (µatm) and $pH_T$ (total scale) measured immediately following carbonate chemistry manipulation of experimental bioassays (Time point 0, $T_0$).

| Cruise ID | Expt ID | $pCO_2$ (µatm) at $T_0$ | | | | | $pH_T$ at $T_0$ | | | | |
|---|---|---|---|---|---|---|---|---|---|---|---|
| | | ambient | 550 (nominal) | 750 (nominal) | 1000 (nominal) | 2000 (nominal) | ambient | 550 (nominal) | 750 (nominal) | 1000 (nominal) | 2000 (nominal) |
| D366 | E01 | 342.3 | 564.1 | 746.4 | 969.6 | | 8.1 | 7.9 | 7.8 | 7.7 | |
| | E02 | n.d. | 533.4 | n.d. | 862.7 | | n.d. | 7.9 | n.d. | 7.8 | |
| | E02b | n.d. | n.d. | n.d. | n.d. | | n.d. | n.d. | n.d. | n.d. | |
| | E03 | 345.4 | 531.2 | 673.9 | 877.8 | | 8.1 | 7.9 | 7.9 | 7.8 | |
| | E04 | 395.4 | 533.4 | 691.4 | 936.6 | | 8.1 | 7.9 | 7.8 | 7.7 | |
| | E04b | n.d. | n.d. | n.d. | n.d. | | n.d. | n.d. | n.d. | n.d. | |
| | E05 | 374.7 | 528.9 | 730.5 | 917.5 | | 8.1 | 7.9 | 7.8 | 7.7 | |
| | E05b | n.d. | n.d. | n.d. | n.d. | | n.d. | n.d. | n.d. | n.d. | |
| | E06 | n.d. | n.d. | n.d. | n.d. | | n.d. | n.d. | n.d. | n.d. | |
| JR271 | NS | 286.5 | 524.7 | n.d. | 620.1 | | 8.2 | 7.9 | n.d. | 7.9 | |
| | IB | 280.4 | 434.3 | 583.3 | 673.1 | | 8.2 | 8.0 | 7.9 | 7.9 | |
| | GG | 326.8 | 565.2 | 741.8 | 1012.2 | | 8.1 | 7.9 | 7.8 | 7.7 | |
| | GI | 312.2 | 583.9 | 789.3 | 948.2 | | 8.1 | 7.9 | 7.7 | 7.7 | |
| | BS | 310.6 | 535.1 | 649.1 | 683.6 | | 8.1 | 7.9 | 7.9 | 7.8 | |
| JR274 | DP | 287.0 | | 598.2 | | | 8.2 | | 7.9 | | |
| | WS | 275.1 | | 533.8 | | | 8.2 | | 7.9 | | |
| | SG | 342.6 | | n.d. | 823.4 | 1410.4 | 8.1 | | n.d. | 7.7 | 7.5 |
| | SS | 283.8 | | n.d. | 773.2 | 1557.5 | 8.2 | | n.d. | 7.8 | 7.5 |

For figure 6, 7 and 8, the data is plotted against the carbonate chemistry ($C_T/A_T$) of the sampled waters (i.e. measurements made before carbonate chemistry manipulation), not from the incubations, so again the data does not require replotting.

*2.7 In conclusion, I get the impression that the authors really try to tell a story that does not fit their data. I think that the hypothesis (more variable carbonate chemistry causes organisms to be less*

*sensitive) presented here makes a lot of sense, but for various reasons the data set is not suited to prove or disprove it.*

**We hope that, given the changes we have now made to the manuscript and the clarification we have provided on all the reviewers points, alongside the related analysis within Richier et al. (2018), that the reviewer will now be satisfied that the dataset appropriately addresses the hypothesis posed.**

**Specific comments**

*2.8 Title and throughout: To my knowledge, the term "resilience" refers to the ability of a system to return to the initial state after disturbance. Therefore, I do not think that the experimental setup and the response pattern (or its absence) in your study allows for statements on resilience. I suggest to use "insensitivity", "resistance" or something along these lines instead.*

**L481: "resilience" replaced with "insensitivity"**

**L507: "resilience" replaced with "the ability to resist"**

**L589: now reads: "The results of our microcosm experiments suggest insensitivity of *de novo* DMSP production and net DMS production in the microbial communities of the polar open oceans to short term changes in carbonate chemistry".**

*2.9 L22-27: As you refer to the other studies conducted in the Arctic, you also need to include their results in your statement, or be more specific that you only refer to the presented dataset and not the polar evidence in general.*

**We have altered the text at L23-28 to take the reviewers comment into account, and emphasise we are specifically referring to the presented dataset:**

**"Based on our findings, we hypothesise that the differences in DMS response between temperate and polar waters reflect the natural variability in carbonate chemistry to which the respective communities of each region may already be adapted. This implies that future temperate oceans could be more sensitive to OA resulting in a change in DMS emissions to the atmosphere, whilst perhaps surprisingly DMS emissions from the polar oceans may remain relatively unchanged".**

*2.10    L24-31: In the discussion, you do not refer to "geographical" or "regional" differences but compare temperate vs. polar systems. I would try to be more consistent here.*

**L23 – 25 now reads: "Based on our findings, we hypothesise that the differences in DMS response between temperate and polar waters reflect the natural variability in carbonate chemistry to which the respective communities of each region may already be adapted".**

**At L32 we use "geographically distinct regions" and "regionally distinct" in reference to the temperate vs polar waters, which we believe is appropriate.**

*2.11    Introduction: The introduction is quite long, especially DMS(P) biogeochemistry is described in a lot of detail, even though most of this is not referred to in the discussion. I would suggest to*

*shorten it. If your discussion does not focus at all on biogeochemistry, do you really need all this detail here?*

**We disagree with the reviewer. It is important to set the scene, to convince the reader of the importance of DMS, and justify why we are interested in the response of DMS to OA, both in general biogeochemical terms, and specifically with regard to the polar regions of this study.**

*2.12    L92-95: This is correct, but one shouldn't forget that it is the coastal areas that are the most productive and therefore important ones. In my opinion you do not even have to somehow restrict the importance of these two previous studies, your study is a valuable contribution even though two other ones exist.*

**We have re-worded the sentence at L96-97 at the reviewer's recommendation**:

**"However, these two single studies provide limited information on the wider response of the open Arctic or Southern Oceans".**

*2.13    L118: Here and in a few other instances you refer to your incubations as being "identical", but in the methods you state that the day length was adapted to the respective in situ conditions. Therefore, I would not use the term "identical".*

**L20: "identical" has been replaced with "similar"**

**L120: "identical" changed to "near-identical"**

**L411: "identical" has been deleted**

*2.14    L119-120: I think the differences in nutrients and incubation temperatures play a big role in understanding the results, so they need to be shown in one of the tables. Referring to a paper under review is not sufficient for such important information. Generally, the authors should provide all relevant information (at least in the supplement) if the other manuscript is not publically available yet.*

**As indicated above, the paper of Richier et al. (2018) has now been accepted for publication.**

**In situ temperatures at the time of sample collection are already shown in Table 1. Incubation temperatures were maintained (±1°C) at the in situ value.**

**Methods text has been adjusted so now reads:**

**L190: "Bottles were incubated inside a custom-designed temperature- and light-controlled shipping container, set to match (±<1°C) the *in situ* water temperature at the time of water collection (shown in Table 1) (see Richier et al. 2018)".**

**The nutrients and incubation temperatures did not play a role in understanding the results. We refer to Richier et al. (2018) for more detailed discussion of this, and have added the following to the current manuscript:**

**L440: "Across all experiments, the response of net total community Chl *a* and net growth rates of small phytoplankton (<10 μm) scaled with pCO$_2$ treatment, and strongly correlated with in situ carbonate chemistry, whilst no relationships were found with any of the other wide range of initial physical, chemical or biological variables (Richier et al. 2018). Overall, the observed differences in regional response to carbonate chemistry manipulation could not be attributed to any other measured factor that varied between temperate and polar waters. These include ambient nutrient concentrations, which varied considerably but had no influence on the response, and initial community structure, which was not a significant predictor of the response (Richier et al. 2018)".**

*2.15     L122-125, L130: While I do agree that differences in environmental variability most likely have an impact on the adaptive capacity of communities, you cannot estimate this adaptive capacity in short-term incubation experiments that run for several days only.*

**See response 1.3 to reviewer #1. Text has been altered accordingly.**

*2.16     L229-231: I am wondering if it wouldn't make more sense to normalize DMSP concentrations to biomass? This is especially the case if you want to test for "stress-induced algal processes" (L135-136) rather than biomass-dependent effects.*

**We feel this is not necessary, as we present specific rates of DMSP synthesis. In vivo DMSP synthesis is closely associated with photosynthesis within the cell, so determination of the rate of this process gives an indication of the effects of stress-induced algal processes on DMSP production. This is a much more useful parameter than biomass-normalised DMSP standing stocks, as the DMSP pool is the net results of various and varying processes (see Stefels et al. 2009), with variable contributions to DMSP production by different groups of phytoplankton.**

*2.17     L252-259: I do not think that you can infer growth rates from the Chla measurements, given that there was probably strong photoacclimatory processes happening in response to the change in light fields (naturally varying to constantly high). You do not really need these rates for your story, so I suggest to omit this parameter all together, i.e. also from results and discussion.*

**At the reviewer's suggestion we have removed relative growth rates from the paper.**

*2.18     L278: The results from the Atlantic experiments are used a lot in the discussion, they should therefore also be included in the results (and methods), especially but not exclusively the previously unpublished ones.*

**We have now described the methods and results from the 6 previously unpublished experiments in temperate waters.**

**Some minor adjustments have been made to the methods text to account for the temperate experiments, but the reader is generally referred to the related studies for the full details (Hopkins and Archer 2014, Richier et al. 2014, Richier et al. 2018):**

**L151: "Additionally, four previously unpublished experiments from D366 are also included (E02b, E04b, E05b, E06) as well as two temperate experiments from JR271 (NS and IB) (see Table 1)".**

**L197: "For Southern Ocean and all temperate water stations, an 18:6 light: dark cycle was used".**

L202: "Experiments were generally run for ≥4 days (15 out of 18 experiments), with initial sampling proceeded by two further time points. For three temperate experiments (E02b, E04b, E05b, see Table 1) a shorter 2 day incubation was performed, with a single sampling point at the end. For E06 (see Table 1) high time frequency sampling was performed (0, 1, 4, 14, 24, 48, 72, 96 h) although only the data at 48 h and 96 h is considered in this analysis".

Figure 2 now includes depth profile data from all 18 sampling stations, and the results text now includes full description of the data for all 18 stations:

L301: "At temperate sampling stations, sea surface temperatures ranged from 10.7°C for *Iceland Basin*, to 15.3°C for *Bay of Biscay*, with surface salinity in the range 34.1 – 35.2, with the exception of station E05b which had a relatively low salinity of 30.5 (Figure 2 and Table 1)".

L312: "Chl *a* concentrations in temperate waters ranged from 0.3 µg L$^{-1}$ for two North Sea stations (*E05* and *North Sea*) up to 3.5 µg L$^{-1}$ for *Irish Sea* (Figure 2 and Table 1). Chl a was also variable in polar waters, exceeding 4 µg L$^{-1}$ at *South Sandwich* and 2 µg L$^{-1}$ at *Greenland Ice-edge*, whilst the remaining stations ranged from 0.2 µg L$^{-1}$ (*Weddell Sea*) to 1.5 µg L$^{-1}$ (Figure 2)".

L318: "In temperate waters, maximum DMS concentrations were generally seen in near surface measurements, ranging from 1.0 nM for *E04* to 21.1 nM for *E06*, with rapidly decreasing concentrations with depth (Figure 2 G). DMSP also generally peaked in the near surface waters, ranging from 12.0 nM for E04 to 72.5 nM for *E06*, but the maximum overall DMSP concentration of 89.8 nM was observed at ~20 m for *E05b* (Figure 2 H). Surface DMS concentrations in polar waters were generally lower than temperate waters, ranging from 1 – 3 nmol L$^{-1}$, with the exception of *South Sandwich* where concentrations of ~12 nmol L$^{-1}$ were observed (Figure 2 G)".

The DMS and DMSP results from the 6 previously unpublished temperate microcosm experiments are now shown in the Supplementary Information, in Table S4 (E02b, E04b, E05b, E06 from D366) and Figure S2 (*North Sea* and *Iceland Basin* from JR271), and described in brief in the results section:

L355: "Results from the previously unpublished experiments from temperate waters are in strong agreement with the five experiments presented in Hopkins and Archer (2014), with consistently decreased DMS concentrations and enhanced DMSP under elevated $CO_2$. The data is presented in the Supplementary Information, Table S4 and Figure S2, and included in the meta-analysis in section 4.1 of this paper".

Table S4. DMS and DMSPt response (mean $\pm$ SD, $n = 3$) to high $CO_2$ treatments during previously unpublished small-scale experiments from the NW European shelf cruise D366. For details of stations, see Table 1 in the main paper.

| | 0 h ambient | 48 h ambient | 48 h 550 µatm | 48 h 750 µatm |
|---|---|---|---|---|
| **DMS (nM)** | | | | |
| *E02b* | 2.4 ± 0.3 | 2.1 ± 0.6 | | 2.7 ± 0.6 |
| *E04b* | | 6.4 ± 1.4 | | 14.7 ± 8.1 |
| *E05b* | | 3.3 ± 0.1 | | 4.5 ± 0.6 |
| *E06* | 18.7 ± 0.5 | 18.1 | 24.2 | 25.2 |
| **DMSPt (nM)** | | | | |
| *E02b* | | 49.5 ± 2.0 | | 26.4 ± 2.9 |
| *E04b* | | 68.2 ± 10.3 | | 36.8 ± 7.5 |
| *E05b* | | 48.7 ± 11.2 | | 37.4 ± 4.8 |
| *E06* | 76.7 ± 5.7 | 114.6 | 98.43 | 108.5 |

[Figure]

Figure S2. DMS, total DMSP and particulate DMSP concentrations (nmol L$^{-1}$) during experimental microcosms performed in temperate waters at stations *North Sea* and *Iceland Basin* from cruise JR271. Data shown is mean of triplicate incubations, and error bars show standard error on the mean. Locations of water collection for microcosms are given in Table 1.

*2.19 L284-287: Methods are missing for the nanoflagellate and bacteria abundances data,*

**The following has been added to the methods section (L278):**

**"2.8 Community composition**

**Composition of small phytoplankton and bacteria community composition was assessed by flow cytometry. For details of methodology, see Richier et al. (2014)".**

*2.20    L291: Methods for irradiance measurements are missing*

**Text has been added to the methods at L162:**

**"At each station, pre-dawn vertical profiles of temperature, salinity, oxygen, fluorescence, turbidity and irradiance were used to choose and characterise the depth of experimental water collection".**

*2.21    L314: This is important information that really helps your line of argument, I would therefore put stronger emphasis on this in the discussion.*

**It is slightly unclear but does the reviewer refer to the statement: "Significant differences ceased to be detectable by the end of the incubations (168 h)"?**

**The section referred to describes the DMS response within the *South Sandwich* experiment which showed a significant $CO_2$ treatment effect after 96 h of incubation, which then ceased to be detectable by the end of the experiment (172 h). This single time point measurement out of the 7 polar experiments is an exception to the general overall trend of no DMS response – it was also accompanied by almost identical mean DMSP concentrations under all $CO_2$ treatments (Figure 4 G). Therefore it is difficult to gauge the significance of this result. Once this data is combined into the meta-analysis, it is clear this DMS response at *South Sandwich* is negligible compared to the magnitude of responses we saw in temperate waters (Figure 6 A).**

*2.22    L328-335: This comparison of standing stocks is highly dependent on the time of sampling. You therefore need to include information about and discussion on the timing of sampling relative to bloom phenology. I.e. if the Arctic and Southern Ocean samples were taken in (macro and/or micro) nutrient depleted waters after a bloom, can you really make such general statements on polar vs. temperate waters? Was the temperate sampling also conducted in similar phases of biomass dynamics? If not, you have a problematic bias towards low productivity in the polar samples that needs to be taken into account.*

**Coherent responses to OA occurred regardless of initial conditions, in terms of both the general biological response, and the DMS/P response. See Richier et al. (2018) for an assessment of the observed responses in comparison to a range of initial environmental. Importantly, differences in net phytoplankton growth rates as a function of $pCO_2$ treatment showed no correlation with any of the other wide range of initial physical, chemical and biological variables tested, including nutrient concentrations. Initial community structure was not an important factor in determining responses to pCO2 treatment (Richier et al. 2018).  Thus although it is likely that we sampled waters that were at different stages of bloom phenology, this did not appear have an influence on our findings. Indeed, we note that a wide range of initial chlorophyll standing stocks was sampled on both high latitude cruises (Table 1). Overall, the most important factor influencing the biological, and DMS/P response to elevated $pCO_2$, was the carbonate chemistry characteristics of the sampled waters. Thus, our findings suggest that both the organism and ecosystem level response to OA is related to variability in the mean state of the carbonate chemistry system, alongside the associated natural range of variability in carbonate chemistry experienced by organisms (Flynn et al. 2012; Richier et al. 2018). Both these factors are likely linked to regional variability in the buffering capacity of ocean waters.**

The lower rates of DMSP synthesis in polar waters compared to temperate waters is not necessarily due to lower levels of productivity but rather 'slower metabolic processes' as we state in this section. We compare our results to 'non-bloom conditions' from the Archer et al. (2013) paper because although higher rates were observed during this study ($10 - 15$ nmol $L^{-1}$ $d^{-1}$), they occurred following artificial addition of inorganic nutrients to the mesocosms, which is not comparable to open ocean rates measured during the OA microcosm experiments.

We addressed the issue of 'slower metabolic rates' in colder polar waters in response 1.7 to reviewer #1.

*2.23    L340-342: This in a strong indication for the importance of other drivers (nutrients, species composition, ...). You need to show these and check whether there are significant effects here.*

Whilst we agree it would be interesting to attempt to unravel what is driving any differences in DMSP production in polar waters, we feel that it is outside of the scope of this paper.

*2.24    L360ff: I really like this way of presenting the data. You should, however, also show the same plot with pCO2 instead of TA/DIC for comparison because I do not agree with you that this ratio gives a full overview of the in situ carbonate chemistry.*

We feel it would not be a useful exercise to replot all the data against $pCO_2$, based on single discrete measurements. The relevance of this value is unclear, as it can be so variable in space and time.

We use DIC/Alk as it is the simplest way of representing the buffer capacity of the sampled waters. We could also have plotted against the Revelle factor of the sampled waters, and the relationship would have looked almost identical, as the Revelle factor is indeed a function of DIC/Alk, and quantified the ocean's sensitivity to an increase in $CO_2$. Therefore we believe that DIC/Alk is the simplest and more appropriate way of visualising our data in terms of its geographical location.

*2.25    L372 and throughout the entire manuscript: Report the time points in days or hours instead of T1, T2 etc. because this is not consistently the same time point as well as for better readability and consistency throughout the text.*

Reporting the time points in hours throughout the manuscript would make the results text clunky and confusing to read, as there is some variability. Therefore, we will keep the $T_1/T_2$ notation, but refer to the times broadly and refer the reader to Table 1 which outlines all the specific sampling times.

Added at L201: "($T_1$, $T_2$, see Table 1 for specific times for each experiment)"..,

L410 now reads: "…was minimal at all sampling points…"

L413 now reads: "…particularly at $T_1$ ($48 - 96$ h)…"

Figure 7 legend text has been altered: "$T_1$ = 48 h, T2 = 96 h, except for *Weddell Sea* and *South Georgia* (72 h, 144 h)".

**Figure 8 legend text has been altered: "$T_1$ = 48 h, $T_2$ = 96 h, except for *Weddell Sea* and *South Georgia* (72 h, 144 h) and *South Sandwich* (96 h, 168 h)".**

*2.26    L377-282: This strongly suggests that, due to temperature-driven differences in metabolic rates and their effects on how fast the communities can acclimate to changed conditions, the experiments emerge out of measurement noise at different times.*

**If the differences were driven by temperature, rates would be expected to be higher in warmer temperate waters and lower in cold polar waters – but this is not the case. Also see response 1.6 to reviewer 1.**

**To make our point more clearly, the text now reads (L413):**

**"*De novo* DMSP synthesis and DMSP production rates show a similar relationship with $C_T/A_T$ (Fig. 7 A and B), with a significant suppression of DMSP production rates in temperate waters compared to polar waters (Fig. 7B, Kruskal-Wallis One Way ANOVA $H$ = 8.711, $df$ = 1, $p$ = 0.003). Although a similar trend was seen for *de novo* DMSP synthesis, the difference between temperate and polar waters was not statistically significant (Fig. 7A)".**

**Therefore, this suppression of rates in temperate waters is likely related to the relative decreases in net growth (Chl a accumulations, phytoplankton cell counts, community biomass) seen in temperate waters (see also Richier et al. 2018).**

*2.27    Discussion: A discussion of stress vs. acclimated response is missing*

**The following text has been added to the discussion at L631:**

**"Our results imply that the phytoplankton communities of the temperate microcosms initially responded to the rapid increase in $pCO_2$ via a stress-induced response, resulting in large and significant increases in DMS concentrations occurring over the shortest timescales (2 days), with a lessening of the treatment effect with an increase in incubation time (Hopkins and Archer 2014). Within non-nutrient amended treatments such a reduction in response with time may also have been driven by nutrient exhaustion, which could have lead the system to a similar state across all $CO_2$ treatments, although we note that carbonate chemistry manipulation induced responses were also similar within nutrient amended treatments (Richier et al. 2014, 2018). The dominance of short response timescales in well-buffered temperate waters may also indicate rapid acclimation of the phytoplankton populations following the initial stress response, which forced the small-sized phytoplankton beyond their range of acclimative tolerance and lead to increased DMS (Richier et al. 2018, Hopkins and Archer 2014).**

**This supports the hypothesis that populations from higher latitude, less well-buffered waters, already possess a certain degree of physiological tolerance to variations in carbonate chemistry environment. Although initial community size structure was not a significant predictor of the response to high $CO_2$, it is possible that a combination of both community composition and the natural range in variability in carbonate chemistry – as a function of buffer capacity – may influence the DMS/P response to OA over a range of timescales (Richier et al. 2018)".**

*2.28    L399: Everything until here reads more like results than like a discussion section. Please consider rearranging.*

**The section the reviewer refers to describes the meta-analysis of all the data from the 3 cruises – we believe this can be considered suitable content for the discussion and have left it unchanged. It should be viewed as a synthesis rather than a simple description of results.**

*2.29    L410-412: The authors seem to imply that CO2 sensitivity is only occurring in form of negative effects, even though there are many studies that show beneficial effects of increased substrate availability for photosynthesis, which is particularly true for picoeukaryotes (e.g. Schulz et al. 2017). Please take this aspect into account.*

**The section the reviewer refers to does not imply this. Rather, here we provide an explanation for our observations.**

*2.30    L436-439: I do not agree that your data really shows this: Figure 9 indicates the Arctic Ocean carbonate chemistry to be actually more similar to the Atlantic than to the Southern Ocean.*

**We disagree. The data show that the variability in carbonate chemistry in the polar oceans is much larger than in temperate waters – as described in the text the reviewer refers to.**

*2.31    L444-448: Such a comparison only makes sense if the same geographical and temporal ranges, and phases of biomass cycle (pre-bloom/ bloom/post-bloom, before/after winter convection etc.) were covered in the different study areas. Please clarify if this was the case.*

**See response 1.12 to reviewer #1 with regard to accounting for the possible variability in pH over seasonal scales.**

*2.32    L451-455: In the Southern Ocean, several studies have shown strong OA-effects on species composition (e.g. Tortell et al. 2008, Feng et al. 2010, Hoppe et al. 2013, Trimborn et al. 2017). L455-457: Similarly, you are missing previous work done in the Arctic (Coello-Camba et al. 2014, Holding et al. 2015, Thoisen et al. 2015, Hoppe et al. 2017a,b) that need to be considered.*

**We have re-worded the suggested line so that it no longer implies that this is all the available data.**

**L512:"A number of previous studies in polar waters have reported similar findings". However, we believe it would over-complicate this part of the discussion, and disrupt the flow, if we were to bring in the other suggested references at this point in the paper.**

**L519: We have added in two recently published study which provides further substantiation for our hypothesis, "Subarctic phytoplankton populations demonstrated a high level of resilience to OA in short term experiments, suggesting a high level of physiological plasticity that was attributed to the prevailing strong gradients in $pCO_2$ levels experienced in the sample region (Hoppe et al. 2017). Furthermore, a recent study describing ten $CO_2$ manipulation experiments in Arctic waters found that primary production was largely insensitive to OA over a large range of light and temperature levels (Hoppe et al. 2018). This supports our hypothesis that, relative to temperate communities, polar microbial communities may have a high capacity to compensate for**

environmental variability (Hoppe et al. 2018), and are thus already adapted to, and are able to tolerate, large variations in carbonate chemistry".

Therefore, we have made reference to these studies later in the paper at the end of section 4.4, where we already provide evidence that the DMS response is likely to be variable over temporal and spatial scales:

L611: "Furthermore, a number of other studies from both the Arctic e.g. (Coello-Camba et al. 2014; Holding et al. 2015; Thoisen et al. 2015) and the Southern Ocean e.g. (Tortell et al. 2008; Hoppe et al. 2013; Trimborn et al. 2017) suggest that polar phytoplankton communities can demonstrate sensitivity to OA, in contrast to our findings. This emphasises the need to gain a more detailed understanding of both the spatial and seasonal variability in the polar phytoplankton community and associated DMS response to changing ocean acidity".

*2.33    L460: n=3 is not "highly" replicated*

"Highly" has been omitted.

*2.34    L469: Why are you comparing your data in detail with Archer et al. (2013) but not Hussherr et al. (2017)?*

We use this section to highlight the differences between experimental approaches, as it is useful for the reader to understand why we might see such different results between microcosm experiments and mesocosm experiments. We have altered the text to make this point come across more clearly to the reader:

"Experimental data clearly provide useful information on the potential future DMS response to OA, but these data become most powerful when incorporated in Earth System Models (ESM) to facilitate predictions of future climate. To date, two modelling studies have used ESM to assess the potential climate feedback resulting from the DMS sensitivity to OA (Six et al. 2013; Schwinger et al. 2017), and both have used results from mesocosm experiments. However, the DMS responses to OA within our short term microcosm experiments contrast with the results of most previous mesocosm experiments, and of particular relevance to this study, an earlier Arctic mesocosm experiment (Archer et al. 2013).  Whilst no response in DMS concentrations to OA was generally seen in the microcosm experiments discussed here, a significant decrease in DMS with increasing levels of $CO_2$ in the earlier mesocosm study was seen. Therefore, it is useful to consider how the differences in experimental design between microcosms and mesocosms may result in contrasting DMS responses to OA".

 As Hussherr et al. also used a microcosm approach, we include a short comparison at the end of this section (L594 onwards) to emphasise that discrepancies can also occur even when using similar experimental techniques.

*2.35    L475: I would rather refer to the most common not the maximum duration.*

"maximum of…" has been omitted.

*2.36 L482-488: Is this difference really due to different sensitivities, or differences in biological rates, that lead to the fact that small physiological changes are detectable at different time points?*

**It is unlikely that biological rates will vary significantly between mesocosms and microcosms, as each experimental system should be a reasonable representation of the natural system which was sampled. Although there is not total certainty that it is due to differences in sensitivities, this is a hypothesis we put forward to help explain the differences in response to OA that we observe between these experimental approaches.**

*2.37 L515-521: You first imply that the short duration of the experiments would render changes in species composition rather unlikely, but then you report one case where you indeed observed changes. I would say that this indicates that the timescales in general would have allowed for changes in composition also in the other experiments.*

**Text has been altered and now reads (L589): "We did not generally see any broad-scale $CO_2$-effects on community structure in polar waters. This can be demonstrated by a lack of significant differences in the mean ratio of >10 μm Chl *a* to total Chl *a* (*>10 μm : total*) between $CO_2$ treatments, implying there were no broad changes in community composition (Table 2). *South Sandwich* was an exception to this, where large and significant increases in the mean ratio of *>10 μm : total* were observed at 750 μatm and 2000 μatm $CO_2$ relative to ambient $CO_2$ (ANOVA, *F* = 207.144, *p*<0.001, *df* = 3), demonstrated at even the short timescale of the microcosm experiments, it is possible for some changes to community composition to occur".**

*2.38 L543-550: I agree that it is an interesting finding that coastal DMS production seems to be more sensitive to OA than that from the open ocean. This finding does, however, really hint against the proposed mechanisms of insensitivity, because coastal systems are a lot more variable in carbonate chemistry compared to the open ocean (e.g. Thoisen et al. 2015). Thus, the interpretation of and conclusions from the dataset have to be reassessed.*

**Given the reviewers comments on this issue, we believe that the comparison between 'coastal' and 'open ocean' waters complicates this part of our discussion, so we have removed mention of this comparison. We instead discuss the possibility that there is likely to be regional variability in the response of DMS to OA. The key point is that the DMS response to OA in polar regions is complex and likely to be influenced by a number of temporal and spatial factors. The main users of our data are climate modellers, and we wish to emphasise that when trying to model the future flux of DMS, it is important to take this variability into account. The section now reads (L623):**

**"Our findings contrast with two previous studies performed in Arctic waters (Archer et al. 2013, Hussherr et al. 2017) which showed significant decreases in DMS in response to OA. These discrepancies may be driven by differences in the sensitivity of microbial communities to changing carbonate chemistry between different areas, or by variability in the response to OA depending on the time of year, nutrient availability, and ambient levels of growth and productivity. This serves to highlight the complex spatial and temporal variability in DMS response to OA which warrants further investigation to improve model predictions".**

*2.39    Figures 3, 4, 5, 7, 8, S3: Given the lack of control in carbonate chemistry in many experiments (Table S2), this representation is misleading. The data needs to be presented accounted for the real carbonate chemistry in the incubations.*

**We have addressed this above in point 2.6.**

**Technical Corrections**

*2.40    L11: I suggest replacing "we increase" by "to increase"*

**Text changed accordingly.**

*2.41    L12: I suggest referring explicitly to climate change instead of environmental change. Otherwise, the step to OA is kind of abrupt.*

**Done.**

*2.42    L28: Do you really mean "region may vary in response to OA" or rather "region may vary in their response to OA"?*

**Text changed accordingly, now reads: "By demonstrating that DMS emissions from geographically distinct regions may vary in their response to OA,…"**

*2.43    L190: replace "made" by "taken"*

**Done.**

*2.44    L207: omit "all" as in the caption of figure 5 you state that these data are not available for two of the stations.*

**Text changed accordingly, now reads: "*De novo* DMSP synthesis and gross production rates were determined for all microcosm experiments, except *Barents Sea* and *South Sandwich*,…".**

*2.45    L237-238: According to the Journal style, it would be A_T and C_T for total alkalinity and total dissolved inorganic carbon, respectively*

**We have changed $T_A$ to $A_T$ and DIC to $C_T$ throughout.**

*2.46    L372: Omit "identical" as irradiances and temperatures were not the same*

**Done.**

*2.47    L497-500: Something does not see correct in this sentence, please rephrase*

**This sentence has been rephrased (L571): "Moreover, the coastal Arctic mesocosms were enriched with nutrients after 10 days, affording relief from nutrient limitation and allowing differences between $p$CO$_2$ treatments to be exposed, including a strong DMS(P) response".**

*2.48    L532: Insert "low and" between "periods of" and "stable productivity"*

**Done.**

*2.49    L539: "is insensitive to OA during multiple short term microcosm" instead of "is resilient to OA during multiple, highly replicated short term microcosm"*

**Done.**

*2.50    L542: add additional references mentioned above*

**We have instead removed reference to Davidson et al. (2016) as this was incorrectly cited here, and only refer to the results from our own study (which was the intention).**

*2.51    L559: Replace "results from our study indicate" by a more honest "we hypothesise" or something similar.*

**Done.**

*2.52    Table 1: Add macro nutrient (at least NO3) levels and incubation temperatures (will be more variable than in situ). Also "Comment" should read "Reference". Shouldn't "Sample depth" read "Sampling depth"?*

**The temperate of the incubation container was maintained at the in situ sampling temperature (±<1°C) (see methods in Richier et al. 2014, Biogeosciences, doi:10.5194/bg-11-4733-2014). Methods text (L191) has been altered to confirm this:**

**"Bottles were incubated inside a custom-designed temperature- and light-controlled shipping container, set to match the *in situ* water temperature at the time of water collection (±<1°C, see Richier et al. 2018)".**

**Nitrate concentrations have been added to the Table as suggested. And other suggested changes have been made.**

*2.53    All Figures: Please indicate number of replicates and type of error estimate in the caption*

**Now included in figure captions for Figure 3 and Figure 4:**

**"Data shown is mean of triplicate incubations, and error bars show standard error on the mean".**

*2.54    Figure 2: Replace "µE m-2 s-1" by "µmol photons m-2 s-1" or "µmol quanta m-2 s-1" in figure and caption. Also, the panels are so close together that the top and bottom axis descriptions get messy, please move them apart a bit.*

**Done.**

**References**

Archer, S. D., S. A. Kimmance, J. A. Stephens, F. E. Hopkins, R. G. J. Bellerby, K. G. Schulz, J. Piontek and A. Engel (2013). "Contrasting responses of DMS and DMSP to ocean acidification in Arctic waters." Biogeosciences **10**(3): 1893-1908.

Coello-Camba, A., S. Agustí, J. Holding, J. M. Arrieta and C. M. Duarte (2014). "Interactive effect of temperature and CO2 increase in Arctic phytoplankton." Frontiers in Marine Science **1**: 49.

Eppley, R. W. (1972). "Temperature and phytoplankton growth in the sea." Fish. bull **70**(4): 1063-1085.

Flynn, K. J., J. C. Blackford, M. E. Baird, J. A. Raven, D. R. Clark, J. Beardall, C. Brownlee, H. Fabian and G. L. Wheeler (2012). "Changes in pH at the exterior surface of plankton with ocean acidification." Nature Climate Change **2**(7): 510-513.

Hagens, M. and J. J. Middelburg (2016). "Attributing seasonal pH variability in surface ocean waters to governing factors." Geophysical Research Letters **43**(24): 12,528-512,537.

Holding, J. M., C. M. Duarte, M. Sanz-Martin, E. Mesa, J. M. Arrieta, M. Chierici, I. E. Hendriks, L. S. Garcia-Corral, A. Regaudie-de-Gioux, A. Delgado, M. Reigstad, P. Wassmann and S. Agusti (2015). "Temperature dependence of CO2-enhanced primary production in the European Arctic Ocean." Nature Clim. Change **advance online publication**.

Hoppe, C. J., N. Schuback, D. M. Semeniuk, M. T. Maldonado and B. Rost (2017). "Functional Redundancy Facilitates Resilience of Subarctic Phytoplankton Assemblages toward Ocean Acidification and High Irradiance." Frontiers in Marine Science **4**: 229.

Hoppe, C. J. M., C. S. Hassler, C. D. Payne, P. D. Tortell, B. Rost and S. Trimborn (2013). "Iron Limitation Modulates Ocean Acidification Effects on Southern Ocean Phytoplankton Communities." PLOS ONE **8**(11): e79890.

Hoppe, C. J. M., K. K. E. Wolf, N. Schuback, P. D. Tortell and B. Rost (2018). "Compensation of ocean acidification effects in Arctic phytoplankton assemblages." Nature Climate Change **8**(6): 529-533.

Kapsenberg, L., A. L. Kelley, E. C. Shaw, T. R. Martz and G. E. Hofmann (2015). "Near-shore Antarctic pH variability has implications for the design of ocean acidification experiments." Scientific Reports **5**: 9638.

Richier, S., E. P. Achterberg, M. P. Humphreys, A. J. Poulton, D. J. Suggett, T. Tyrrell and C. M. Moore (2018). "Geographical $CO_2$ sensitivity of phytoplankton correlates with ocean buffer capacity." Global Change Biology.

Schwinger, J., J. Tjiputra, N. Goris, K. D. Six, A. Kirkevåg, Ø. Seland, C. Heinze and T. Ilyina (2017). "Amplification of global warming through pH-dependence of DMS-production simulated with a fully coupled Earth system model (under review in Biogeosciences, doi: 10.5194/bg-2017-33)." Biogeosciences.

Six, K. D., S. Kloster, T. Ilyina, S. D. Archer, K. Zhang and E. Maier-Reimer (2013). "Global warming amplified by reduced sulphur fluxes as a result of ocean acidification." Nature Climate Change **3**(11): 975.

Stillman, J. H. and A. W. Paganini (2015). "Biochemical adaptation to ocean acidification." Journal of Experimental Biology **218**(12): 1946-1955.

Thoisen, C., K. Riisgaard, N. Lundholm, T. G. Nielsen and P. J. Hansen (2015). "Effect of acidification on an Arctic phytoplankton community from Disko Bay, West Greenland." Marine Ecology Progress Series **520**: 21-34.

Tortell, P. D., C. D. Payne, Y. Li, S. Trimborn, B. Rost, W. O. Smith, C. Riesselman, R. B. Dunbar, P. Sedwick and G. R. DiTullio (2008). "$CO_2$ sensitivity of Southern Ocean phytoplankton." Geophysical Research Letters **35**.

Trimborn, S., T. Brenneis, C. J. M. Hoppe, L. M. Laglera, L. Norman, J. Santos-Echeandía, C. Völkner, D. Wolf-Gladrow and C. S. Hassler (2017). "Iron sources alter the response of Southern Ocean phytoplankton to ocean acidification." Marine Ecology Progress Series **578**: 35-50.

---

## Author Comment (AC2) · 21 Jun 2018

We thank both anonymous reviewers for their detailed, constructive, and positive reviews of our manuscript – we greatly appreciate the care and detail that has gone into its assessment. Our response to both sets of reviews has been uploaded as a single supplement file.

Please also note the supplement to this comment:
https://www.biogeosciences-discuss.net/bg-2018-55/bg-2018-55-AC2-supplement.pdf

---

## Author Comment (AC3) · 28 Feb 2019

The review process for this paper has been suspended whilst I am on maternity leave and I will be addressing the reviewers' comments and resubmitting the paper when I return to work later in the year. Apologies for the delay.

---

## Referee Report (RR1)

General Comments

The paper describes a series of high pCO2 incubation experiments performed on three cruises in temperate, Arctic and Southern Ocean waters during the summer period in each location. Studied in the incubations are the changes in DMS and DMSP over the short term (hours to days), and the changes in production rates of these compounds. The manuscript is clear and well written, and highlights the regional differences in the response to elevated pCO2, and attributes the differences to the variability in the carbonate system. Given that two of the environments studied were in polar regions, discussion of the effect of sea ice on the carbonate chemistry and the existing response of the phytoplankton community to extreme environments was lacking within the manuscript, particularly a mention of the pH changes experienced by cells while living within the sea ice.

A number of technical and specific comments arose on reading of the manuscript, and I recommend publication if these can be addressed.

Check $p$CO$_2$ is in italics, as there were instances where it was not.

Specific comments

L28. 'This implies that…' The previous sentence did not suggest this implication, so needs rewording. It seems like a sentence is missing here, which actually describes your findings, and is followed by the implication.

L71 Please highlight that the 3.4 Tg S is from the whole southern ocean, as calculated by JT16.

L88. Is there a concurrent predicted decrease in the southern polar region pH as well?

L96. During the introduction there is very little mention of DMSP, other than it is the precursor for DMS. Given that DMSP is one of your measured parameters in the experiments this requires further elaboration in the introduction, in particular the changes in DMSP as identified from the existing mesocosm experiments. This is important given that DMSP showed increases during the Arctic mesocosm.

L100-117. This section on mesocosms is interesting, but slightly irrelevant to the paper as a whole given that your studies are microcosm incubations. As the introduction is already very long, this section could be shortened to one or two sentences describing previous OA responses.

L114 High cost is also a significant factor in why mesocosm experiments are so limited!

L120. Please clarify that the temperate experiments shown here are in addition to those in Hopkins and Archer 2014; ie this manuscript includes four previously unpublished temperate experiments, as well as those previously published.

L136, remove additional comma before reference.

L142 What are the main differences in the environment between temperate and polar environments? You have not mentioned the distinct seasonal cycle of sea ice formation and retreat, different at both poles, which will likely also alter the carbonate chemistry. Does the acclimation of polar phytoplankton to the physiological stress of survival through the polar winter give them an added advantage when it comes to acclimating to OA? Is DMSP produced by polar phytoplankton (i.e. osmoregulation during periods of extreme salinity shift) for a different reason than temperate phytoplankton?

L190 $p$CO2

L222. In the southern ocean, was this acidification fixing method used for the DMSP samples, given previous issues highlighted by del Valle et al 2011 in samples containing Phaeocystis antarctica?

L253 spaces missing between mass 63 etc.

L281 composition used twice in the sentence.

L317. Please state the station where 1.5 µg L-1 was identified, as you have for the minimum.

L318.'reflected in'. Please reword, firstly as reflected implies light reflectance (given the topic of irradiance), and secondly it reads oddly with two instances of 'in'.

L320 – 328. Use of nM when the following paragraph uses nmol L$^{-1}$

L327. Could not understand the significance of the superscript 1, is it a typo?

L480. Should P be lower case?

L485. In this section I would like to see more discussion of sea ice and the extreme environment is imposes on the cells, which could account for some of their resilience to change. Many polar phytoplankton survive the extreme cold of the polar winter by living within the sea ice itself, in an extremely changeable habitat, and these cells seed the surface waters in summer time on melting of the sea ice. Cells are regularly exposed to hypersaline and highly nutrient variable environments, at temperatures below freezing, and in highly elevated pH environments (Thomas and Dieckmann 2002, Rysgaard et al 2012). Although your experiments were not associated directly with the polar sea ice and occurred during summer, the influence of the ice on the phytoplankton population will be dependent on the seed populations, and allow for greater tolerance of the incubation perturbation than in temperate communities. In the seasonal cycle of the Antarctic, the behaviour of the summer phytoplankton community development is dependent on the conditions experienced the previous winter (Venables et al 2013).

L523. P in italics

L561. Hopkins reference is 2010b, but only 1 Hopkins 2010 ref is present.

L609 italic p

L626, commas missing from references.

---

## Author Response (AR2)

**Final author comments on "Polar dimethylsulfide (DMS) production insensitive to ocean acidification during shipboard microcosm experiments: a meta-analysis of 18 experiments from temperate to polar waters." by F.E. Hopkins et al., manuscript number bg-2018-55.**

Firstly, I would like to apologise for the delay in addressing the reviews to this manuscript. I have been on maternity leave for 12 months.

Secondly, I would particularly like to thank Reviewer #1 for providing a second review of our manuscript. We appreciate their time, effort and insight that have enabled us to improve the paper in many ways. Many thanks also to Reviewer #2 for their helpful, supportive and positive review of our manuscript.

The reviewers comments are shown *in italics*, with our responses shown **in bold**. For reference, our responses to the first round of reviews are shown *in grey italics*. Line numbers in our response refer to the revised version. The marked-up version of the manuscript comes after the response (starting at pg 20), and line numbers refer to this version.

1. **Response to Anonymous Referee #1**

*1.1: After reading manuscript and responses of the authors, I am disappointed with the revision. While the authors implemented many comments, most of my substantial criticism has either been rejected (e.g. 2.6, 2.16, 2.23, 2.24, 2.25, 2.30) or not accounted for appropriately, i.e. by just deleting part of the manuscript instead of discussing potential inconsistencies (e.g. 2.38). It seems to me that any criticism that questions the general interpretation of the dataset has not been included. Under these circumstances, I am not willing to provide a point by point reply to the response, but just highlight the most important points.*

**We regret that we disappointed the reviewer. After careful consideration of their comments, we have made some major changes to the manuscript which we will go through point-by-point below. We hope that we have more than adequately addressed their concerns.**

*1.2: One major problem with this dataset is that the experimental carbonate chemistry was not well controlled. For example, at the 1000µatm pCO2 level, T2 pCO2 levels vary between approx. 400 and 1000µatm (Table S2). Therefore, the data should be represented using the real carbonate chemistry instead of the assigned values. I understand that this implies replotting and reanalysing most of the data, but currently the levels that are tested against each other are actually not separated when it comes to carbonate chemistry.*

**We agree with the reviewer and have now rectified this issue, please see 1.9 below in which we fully address this concern.**

**In the next section (1.3 – 1.8), we have revisited the comments from the first review that the reviewer felt were not adequately addressed in our first response (2.6, 2.16, 2.23, 2.24, 2.25, 2.30, 2.38) and made changes to our manuscript.**

*1.3: 2.16 L229-231: I am wondering if it wouldn't make more sense to normalize DMSP concentrations to biomass? This is especially the case if you want to test for "stress-induced algal processes" (L135-136) rather than biomass-dependent effects.*

*OUR PREVIOUS RESPONSE: We feel this is not necessary, as we present specific rates of DMSP synthesis. In vivo DMSP synthesis is closely associated with photosynthesis within the cell, so*

*determination of the rate of this process gives an indication of the effects of stress-induced algal processes on DMSP production. This is a much more useful parameter than biomass-normalised DMSP standing stocks, as the DMSP pool is the net results of various and varying processes (see Stefels et al. 2009), with variable contributions to DMSP production by different groups of phytoplankton.*

We believe that our previous response was a fair answer to a valid question, rather than a rejection of their criticism. We reiterate our point below. However, we have altered the text to create further clarity:

The reviewer refers to this line: "Gross DMSP production rates during the incubations (nmol $L^{-1}$ $h^{-1}$) were calculated from $\mu$DMSP and the initial particulate DMSP (DMSPp) concentration of the incubations (shown in Figure 4)".

The line describes the accepted method for calculating gross DMSP production rates as previously published by, for example, Stefels et al. 2009, Archer et al. 2013 and Hopkins & Archer 2014, so we feel our previous response is fair. As we previously described, the ability to measure specific rates of DMSP synthesis using mass spectrometric techniques has greatly improved our ability to assess how the DMSP pool is regulated in response to physiological stress. This is a much more useful measurement than net changes in standing stocks of DMSP, whether normalised to biomass or not. For clarity, I have reworded the line and added references:

"In vivo DMSP gross production rates during the incubations (nmol $L^{-1}$ $h^{-1}$) were calculated from $\mu$DMSP and the initial particulate DMSP (DMSPp) concentration of the incubations (Hopkins & Archer 2014, Stefels et al. 2009). These rates provide important information on how the physiological status of DMSP-producing cells may be affected by OA within the bioassays".

*1.4: 2.23 L340-342: This in a strong indication for the importance of other drivers (nutrients, species composition, ...). You need to show these and check whether there are significant effects here.*

*OUR PREVIOUS RESPONSE: Whilst we agree it would be interesting to attempt to unravel what is driving any differences in DMSP production in polar waters, we feel that it is outside of the scope of this paper.*

We apologise for our previous rather vague response. However, this was not intended as a rejection of criticism, rather we were not certain exactly what the reviewer was asking for. The reviewer refers to this line:

"Nevertheless, no consistent and significant effects of high $CO_2$ were observed for rates of *de novo* DMSP synthesis or DMSP production in polar waters".

Therefore, we assume they are referring to the lack of $CO_2$ response generally seen in $\mu$DMSP and DMSP production rates across all polar experiments. The data show little difference between any of the treatments, which suggests "other drivers (nutrients, species composition…)" do not play any importance either.

Alternatively, they may be referring to the apparent, but opposing, $CO_2$ response seen in these parameters at *Drake Passage* (time point 2, 96 h) and *Weddell Sea* (time point 2, 144 h). Here, it may be possible to unravel some more details of the observed response.

For the benefit of the reviewer, we have plotted the auxiliary measurements for *Drake Passage* (Figure R1) and *Weddell Sea* (Figure R2) below. The response we saw in DMSP production rates at *Drake Passage* (significantly higher DMSP production rates, Figure 6 F) corresponds to significantly higher nitrate concentrations for the *high+* treatment, relative to *ambient* (ANOVA *t* = 7.913, *p* = 0.001) (Figure R1 E). The remaining auxiliary measurements show no clear trends.

For *Weddell Sea*, the auxiliary data is quite noisy over the triplicate bottles, no trends are apparent (Figure R2), and no auxiliary measurements correspond to the significantly lowered DMSP production rates at time point 2, 144 h.

For both experiments, it is possible that there were changes in the composition of large phytoplankton (diatoms, dinoflagellates) that may have resulted in differences in DMSP production rates. However, this size fraction of phytoplankton was not quantified during these experiments.

We have edited the section of the manuscript (from L397) in question and added additional discussion (underlined) as follows:

"No consistent effects of high $CO_2$ were observed for either DMSP synthesis or production in polar waters, similar to findings for DMSP standing stocks. However, some notable but contrasting differences between $CO_2$ treatments were observed. There was a 36% and 37% increase in µDMSP and DMSP production respectively at 750 µatm for the *Drake Passage* after 96 h (Figure 5 E, F), and a 38% and 44% decrease in both at 750 µatm after 144 h for *Weddell Sea* (Figure 5 G, H). For *Drake Passage*, the difference between treatments at 96 h coincided with significantly higher nitrate concentrations in the High $CO_2$ treatment (Nitrate/nitrite at 96 h: Ambient = 18.9 ± 0.2 µmol L$^{-1}$, +$CO_2$ = 20.2 ± 0.1 µmol L$^{-1}$, ANOVA *t* = 7.913, *p* = 0.001). However, it is uncertain whether the difference in nutrient availability between treatments (approximately 5 %) within this bioassay experiment would be significant enough to strongly influence the rate of DMSP production.

The differences in DMSP production rates did not correspond to any other measured parameter. It is possible that changes in phytoplankton community composition may have led to differences in DMSP production rates for *Drake Passage* and *Weddell Sea*, but no quantification of large cells (diatoms, dinoflagellates) was undertaken for these experiments".

[Figure]

Figure R1. Auxiliary measurements from bioassay experiment *Drake Passage.*

[Figure]

Figure R2. Auxiliary measurements from bioassay experiment *Weddell Sea.*

*1.5: 2.24 L360ff: I really like this way of presenting the data. You should, however, also show the same plot with pCO2 instead of TA/DIC for comparison because I do not agree with you that this ratio gives a full overview of the in situ carbonate chemistry.*

*OUR PREVIOUS RESPONSE: We feel it would not be a useful exercise to replot all the data against pCO₂, based on single discrete measurements. The relevance of this value is unclear, as it can be so variable in space and time. We use DIC/Alk as it is the simplest way of representing the buffer capacity of the sampled waters. We could also have plotted against the Revelle factor of the sampled waters, and the relationship would have looked almost identical, as the Revelle factor is indeed a function of DIC/Alk, and quantified the ocean's sensitivity to an increase in CO₂. Therefore we believe that DIC/Alk is the simplest and more appropriate way of visualising our data in terms of its geographical location.*

**We agree this is a good way of visualising the data (Figures 7, 8, 9, revised numbering). However, we did not intend for this parameter ($C_T/A_T$) to give a full overview of the carbonate chemistry. Furthermore, the same plot with $pCO_2$ would not demonstrate the same phenomenon as this parameter varies so much in space and time in the surface ocean and does not correspond to**

latitude. $C_T/A_T$ was used because, similarly to the Revelle Factor ($R$), its value increases towards high latitudes (see Figure 3 in Sabine et al. (2004) Science, 305, 367 – 371). Therefore given the reviewer's concerns with regard to the use of $C_T/A_T$, we have decided to re-plot the figures using $R$ for the sampled waters instead. $R$ provides a measure of the total amount of $CO_2$ that can be dissolved in the mixed surface layer. Regions with high $R$ are less-well buffered with respect to changing $CO_2$ input – such that an increase in atmospheric $CO_2$ would result in a greater increase in sea water $pCO_2$, compared to waters with low $R$. Cold polar waters have high Revelle Factors, primarily a result of the increased solubility of $CO_2$ in cold waters. Thus, these plots are a neat way of showing how the response of DMS and other parameters to OA varies between temperate and polar waters, and how the $CO_2$ sensitivity of different communities may relate to the in situ carbonate chemistry. This lowered buffering capacity in polar waters also results in a greater variability in carbonate chemistry experienced by surface ocean communities over a seasonal cycle. This is further impacted by other polar phenomena such as the seasonal build-up and retreat of sea ice. We discuss further details of this in our response to 1.10 below.

We have altered relevant text appropriately (From L433 onwards):

"The relative treatment effects ($[x]_{highCO2}/[x]_{ambientCO2}$) for DMS and DMSP (Figure 7), DMSP synthesis and production (Figure 8), and Chl $a$ and phototrophic nanoflagellate abundance (Figure 9) are plotted against the Revelle Factor of the sampled waters. The Revelle Factor (R), calculated here with CO2Sys using measurements of carbonate chemistry parameters ($R = (\Delta pCO_2/\Delta TCO_2)/(pCO_2/TCO_2)$, Lewis and Wallace, 1998), describes how the partial pressure of $CO_2$ in seawater ($PCO_2$) changes for a given change in DIC (Revelle and Suess 1957; Sabine et al. 2004). Its magnitude varies latitudinally, with lower values (9 – 12) from the tropics to temperate waters, and the highest values in cold high latitude waters (13 – 15). Thus polar waters can be considered poorly buffered with respect to changes in DIC. An equivalent change in $PCO_2$ in temperate waters would result in smaller changes in pH or saturation states than would be observed in polar waters (Egleston et al. 2010). Furthermore, the seasonal sea ice cycle strongly influences carbonate chemistry, such that sea ice regions exhibit wide fluctuations in carbonate chemistry (Revelle and Suess, 1957; Sabine et al., 2004). Sampling stations with a $R$ above ~12 represent the seven polar stations (right of red dashed line Fig. 6 and 7). The surface waters of the polar oceans have naturally higher levels of DIC and a reduced buffering capacity, driven by higher $CO_2$ solubility in colder waters (Sabine et al. 2004). Thus, the relationship between experimental response and $R$ is a simple way of demonstrating the differences in response to OA between temperate and polar waters and provides some insight into how the $CO_2$ sensitivity of different surface ocean communities may relate to the *in situ* carbonate chemistry".

*1.6: 2.25 L372 and throughout the entire manuscript: Report the time points in days or hours instead of T1, T2 etc. because this is not consistently the same time point as well as for better readability and consistency throughout the text.*

*OUR PREVIOUS RESPONSE: Reporting the time points in hours throughout the manuscript would make the results text clunky and confusing to read, as there is some variability. Therefore, we will keep the $T_1/T_2$ notation, but refer to the times broadly and refer the reader to Table 1 which outlines all the specific sampling times.*
*Added at L201: "($T_1$, $T_2$, see Table 1 for specific times for each experiment)".,*
*L410 now reads: "…was minimal at all sampling points…"*
*L413 now reads: "…particularly at $T_1$ (48 – 96 h)…"*
*Figure 7 legend text has been altered: "$T_1$ = 48 h, $T_2$ = 96 h, except for Weddell Sea and South Georgia (72 h, 144 h)".*
*Figure 8 legend text has been altered: "$T_1$ = 48 h, $T_2$ = 96 h, except for Weddell Sea and South Georgia (72 h, 144 h) and South Sandwich (96 h, 168 h)".*

**We now agree and have now removed all use of T0, T1 and T2 notation and simply refer to exact times throughout the manuscript.**

*1.7: 2.30 L436-439: I do not agree that your data really shows this: Figure 9 indicates the Arctic Ocean carbonate chemistry to be actually more similar to the Atlantic than to the Southern Ocean.*

*OUR PREVIOUS RESPONSE: We disagree. The data show that the variability in carbonate chemistry in the polar oceans is much larger than in temperate waters – as described in the text the reviewer refers to.*

**Please see our detailed response to 1.10 below which we hope now thoroughly addresses these concerns.**

*1.8: 2.38 L543-550: I agree that it is an interesting finding that coastal DMS production seems to be more sensitive to OA than that from the open ocean. This finding does, however, really hint against the proposed mechanisms of insensitivity, because coastal systems are a lot more variable in carbonate chemistry compared to the open ocean (e.g. Thoisen et al. 2015). Thus, the interpretation of and conclusions from the dataset have to be reassessed.*

*OUR PREVIOUS RESPONSE: Given the reviewers comments on this issue, we believe that the comparison between 'coastal' and 'open ocean' waters complicates this part of our discussion, so we have removed mention of this comparison. We instead discuss the possibility that there is likely to be regional variability in the response of DMS to OA. The key point is that the DMS response to OA in polar regions is complex and likely to be influenced by a number of temporal and spatial factors. The main users of our data are climate modellers, and we wish to emphasise that when trying to model the future flux of DMS, it is important to take this variability into account. The section now reads (L623):*
*"Our findings contrast with two previous studies performed in Arctic waters (Archer et al. 2013, Hussherr et al. 2017) which showed significant decreases in DMS in response to OA. These discrepancies may be driven by differences in the sensitivity of microbial communities to changing carbonate chemistry between different areas, or by variability in the response to OA depending on the time of year, nutrient availability, and ambient levels of growth and productivity. This serves to highlight the complex spatial and temporal variability in DMS response to OA which warrants further investigation to improve model predictions".*

**Please see our detailed response to 1.11 below which we hope now thoroughly addresses these concerns.**

*1.9: Most importantly, I still not agree with using the nominal carbonate chemistry levels instead of the actual data. As already described in the first round of reviews, control and OA treatments diverge strongly in some of the experiments. As can be seen in table SI1, measured pCO2 levels varied strongly from the nominal values at all time points, and this discrepancy seems to be larger in the polar compared to the temperate experiments. Already at t0, actual values at 100 and 1400 µatm in both polar areas were in most cases several hundred µatm lower than assigned, i.e. about 20-40% lower. This trend is even more pronounced at T2, where measured pCO2 values are as low as 362 µatm in the 750 µatm, 421 µatm at the 1000 µatm, and 767 µatm in the 1400 µatm treatment. These pCO2 levels in the OA treatments significantly overlapped with control conditions. Thus, the presented carbonate chemistry data clearly does not meet the quality requirements of our scientific community, and therefore the nominal levels cannot be used for data analysis.*

We now accept that using the nominal $pCO_2$ values had failed to provide a clear understanding of the data to the reader. All figures are now shown with the actual measured $pCO_2$ values, rather than 'nominal' values. Instead of referring to 'nominal' values, we have divided the treatments into Mid $CO_2$, High $CO_2$, High+ $CO_2$ and High++ $CO_2$, with the actual data shown in Figures 3 and 4 for each experimental time point. We hope this addresses the reviewer's concerns.

To explain this to the reader, the text in section 2.2 now reads:

"The carbonate chemistry within the experimental bottles was manipulated by addition of equimolar HCl and $NaHCO_3^-$ (1 mol $L^{-1}$) to achieve a range of $CO_2$ treatments: Mid $CO_2$ (Target: 550 μatm), High $CO_2$ (Target: 750 μatm), High+ $CO_2$ (Target: 1000 μatm) and High++ $CO_2$ (Target: 2000 μatm) (Gattuso et al., 2010). Three treatment levels were used during the sub-Arctic/Arctic microcosms (Mid, High, High+). For Southern Ocean experiments, two experiments (*Drake Passage* and *Weddell Sea*) underwent combined $CO_2$ and Fe additions (ambient, Fe (2 nM), High $CO_2$, Fe (2 nM) & High $CO_2$ (only high $CO_2$ treatments will be examined here; no response to Fe was detected in DMS or DMSP concentrations). Three $CO_2$ treatments (High, High+, High++) were tested in the last two experiments (*South Georgia* and *South Sandwich*)".

For all 18 experiments the actual attained $pCO_2$ values at t0 were on average 89% ± 12% (±1 SD) of target values. By T1 and T2 biological activity within the bottles resulted in a drawdown of DIC and a concomitant decrease in $pCO_2$ of around 25% by T1 and >50% by T2 (see Richier et al. 2018) – an unavoidable phenomenon within this experimental design. We hope that by clearly presenting all of the $pCO_2$ data for all experiments and all time points in Figures 3 and 4, this will now be clear to the reader.

*1.10: Similarly, I still disagree with the statement that "a narrow range of values for all carbonate parameters was observed in the NW European shelf waters relative to the less well-buffered Arctic and Southern Ocean waters"(new L 490-492). As can be clearly seen in Figure 9, CT/AT ratios (which the authors use as a proxy for buffer capacity) were actually higher in the SO compared to the NW European shelf waters which had the lowest median. The "much larger" variability in polar waters (response letter 2.30) is in fact only observed for the Arctic. These facts clearly contradict the authors conclusion that "that populations from higher latitude, less well-buffered waters, already possess a certain degree of acclimative tolerance to variations in carbonate chemistry environment" (new L 646-648).*

Having now revisited Figure 9, we agree with the reviewer, and concede that these data didn't really illustrate our point. The figure summarised the underway data along each cruise track so really only provided a snapshot of the variability within each region, whilst masking any detail. By grouping all the data together in this way, it also masked the regional differences within each cruise, making it impossible to appreciate the carbonate chemistry characteristics within each region. Any detail or evidence to illustrate our point was lost in the noise. Therefore, we have removed this figure from the paper.

Instead, we have supported our hypothesis with information from the paper by Tynan et al. (2016) which was intended to provide a carbonate chemistry context for all studies derived related to the polar cruises, and Rerolle et al. (2014) which provides underway pH measurements from the NW European shelf cruise. We discuss our findings in the context of the processes that influence carbonate chemistry variability in polar waters (strong biological drawdown, the influence of sea ice, organic matter respiration, water mass effects), and compare this to the variability observed in temperate waters.

**The section in the manuscript now reads (from L581):**

[revised manuscript text omitted]

*1.11: Lastly, I still find the question of coastal/open ocean vs. polar/temperate differences not resolved. The authors wrote in the first version of the manuscript: "We provide further evidence that, in contrast to temperate communities (Hopkins and Archer 2014), polar communities we sampled were relatively insensitive to variations in carbonate chemistry (Davidson et al. 2016; Richier et al. under review), manifested here as a minimal effect on net DMS production. Our findings contrast with two previous studies performed in coastal Arctic waters (Archer et al. 2013, Hussherr et al. 2017) which showed significant decreases in DMS in response to OA." After my comment that this contradicts their hypothesis of generally higher sensitivity of polar compared to temperate communities, the authors just omitted the description of the two cited studies as being "coastal" (response letter 2.38), instead of adapting their discussion to the fact these two coastal studies with higher natural carbonate chemistry variability contradict the hypothesis brought forward in this study. I the response letter, the authors furthermore argue that "The carbonate chemistry/pH variability may be as large in the open waters of the polar oceans as in coastal sea ice waters" (1.11) and they cite two studies to support this statement. It is a well-known and widely accepted fact, however, that coastal sites experience larger environmental variability that oceanic ones. For example, the study by Thoisen et al. (2015) that I referred to already in the first review, showed report a seasonal pH gradient as large as 0.8 units (7.5 -8.3) for a costal Arctic site. In conclusion, I am still convinced that the CO2 sensitivity observed in two previous coastal studies contradicts the hypothesis of this study. The attempt to just ignore instead of discuss this apparent contradiction is, in my view, not an acceptable scientific practise.*

**We absolutely agree that the results of the two previous studies (Archer et al. 2013, Hussherr et al. 2017) contradict the results of this study as we have made clear in the paper:**
**"Our findings contrast with two previous studies performed in Arctic waters (Archer et al. 2013, Hussherr et al. 2017) which showed significant decreases in DMS in response to OA. These**

discrepancies may be driven by differences in the sensitivity of microbial communities to changing carbonate between different areas, or by variability in the response to OA depending on the time of year, nutrient availability, and ambient levels of growth and productivity. This serves to highlight the complex spatial and temporal variability in DMS response to OA which warrants further investigation to improve model predictions".

However, in the revised version, and given the reviewer's previous comments, we removed reference to 'coastal' as the Hussherr et al. (2017) study is not classified as coastal – this was our oversight and further reason that this argument did not stand. So we only discuss the discrepancies with the Hussherr et al. study in terms of differences in time of year and levels of productivity. This serves to highlight the complexities in the response of surface ocean communities to OA, and the difficulties encountered when attempting to compare results from experiments with vastly different designs.

However, we do accept that the discussion of the contradiction in findings could be handled more clearly in the paper. The reason we omitted some discussion following the first review was because we agreed with the reviewer that the coastal vs open ocean polar argument wasn't well handled and was creating confusion.

It seems that the issues arise in two sections of the paper: 4.3 Adaptation to a variable carbonate chemistry environment, and 4.4 Comparison to an Arctic mesocosm experiment.

We have re-worked the discussion in 4.3 (from L581). We no longer categorise results into coastal vs open ocean, as this does not do an adequate job of explaining our data. We have taken more consideration of the general polar environment, in particular the impact of sea ice and strong biological drawdown in carbonate chemistry. Two of our sampling stations were in sea ice zone, sampled in the period following recent ice melt. These two stations (Greenland ice edge and Weddell Sea) both showed the least response in DMS to OA out of all our experiments. Thus, we now argue that ice edge communities can withstand experimental OA, possibly due to the strong influence that sea ice formation and melt has on in situ carbonate chemistry and hence, acclimative tolerance of the associated communities. Our polar experiments in general agree with many previous studies that polar microbial/phytoplankton communities are able to resist experimentally-induced OA. We present the changes to section 4.3 in response 1.10 above.

We recognise that the results of Archer et al. (2013) and Hussherr et al. (2017) contradict our findings, so we have now re-visited section 4.4 with the reviewer's comments in mind and to make our point clearer. The reasons are not fully resolved but could be due to a number of factors:

Drawing comparisons between mesocosms and microcosms is challenging given the great differences in experimental design. Both address a different set of hypotheses, as already outlined in the discussion. Both have strengths and weaknesses. The EPOCA mesocosm experiment was located in a sheltered fjord on the west coast of Svalbard, heavily influenced by glacial meltwater, and during a year that had not experienced any winter sea ice. This environment is so different to the sampling stations for the microcosm experiments that it is perhaps unsurprising that the general response to OA was so different. In particular, the fjord was characterised by low in situ $CO_2$ concentrations of 185 µatm, whereas polar in situ $CO_2$ concentration in this study ranged from 273 µatm at *South Sandwich* to 510 µatm at *Weddell Sea*. The second half of the mesocosm experiment was nutrient-enriched, and it was following this perturbation that the greatest response in DMS and DMSP to OA was observed. The OA response became most evident when the nutrient-induced bloom led to the formation of a 'winners vs losers' dynamic – something not attained within the microcosms, which received no nutrient addition. Before the nutrient addition to the mesocosms, the response to OA in DMS and DMSP was detectable but minimal – and here I argue that perhaps this demonstrates some resilience within the fjord communities that could result from adaptation to a coastally-influenced variability in carbonate chemistry. This resilience was nullified by the nutrient addition, as perhaps this allowed a shift in community composition to species with less tolerance to high $CO_2$.

The section beginning "The short duration of the microcosm experiments (4 – 7 d) allows…" may be creating confusion, although we are satisfied that our discussion is valid so have not made any changes here. In this section we compare the short term microcosm experiments (4 – 7 d) to the first 5 – 10 d of mesocosm experiments. This seems valid because one of the biggest discrepancies between microcosms and mesocosms is the duration of the experiments. We argue that for most mesocosms, little difference between treatments in terms of the DMS/P response is seen during this initial phase – so this could imply that the communities do possess some short-term tolerance to the induced OA. We then postulate that this may be because mesocosms are generally performed in coastal waters, wherein the communities may have naturally experienced a variable carbonate chemistry environment. It's only once a bloom-dynamic develops do we see strong responses in DMS/P to OA in mesocosm experiments, and this is where the data becomes less comparable to the microcosm experiments. Thus we go on to discuss the affect that nutrient addition has on the growth dynamics within the mesocosm, and how this could drive the observed response to CO2 addition.

*1.12: Please also note that in my view, papers should be understandable and convincing by themselves. So while I did read Richier et al. (2018), I am providing my views on the current manuscript only.*

**We also agree that a paper should be able to stand alone, but some cross-referencing to a paper describing the same experiments from the same research cruises should be acceptable to the reader. This is useful to avoid over-complicating an interesting story about the response of DMS to OA and distracting from the main thrust of this manuscript.**

**2. Response to Anonymous Referee #2**

*2.1 The paper describes a series of high pCO2 incubation experiments performed on three cruises in temperate, Arctic and Southern Ocean waters during the summer period in each location. Studied in the incubations are the changes in DMS and DMSP over the short term (hours to days), and the changes in production rates of these compounds. The manuscript is clear and well written, and highlights the regional differences in the response to elevated pCO2, and attributes the differences to the variability in the carbonate system.*

**We appreciated these positive and supportive comments on our manuscript.**

*2.2 Given that two of the environments studied were in polar regions, discussion of the effect of sea ice on the carbonate chemistry and the existing response of the phytoplankton community to extreme environments was lacking within the manuscript, particularly a mention of the pH changes experienced by cells while living within the sea ice.*

**We agree and have now added the following section to the manuscript, which significantly improves our discussion and interpretation of the dataset (from L581):**

[revised manuscript text omitted]

*2.3 A number of technical and specific comments arose on reading of the manuscript, and I recommend publication if these can be addressed.*

**Many thanks – we hope we have addressed all your comments appropriately.**

*2.4 Check pCO2 is in italics, as there were instances where it was not.*

**All checked and changed throughout.**

*2.5 L28. 'This implies that…' The previous sentence did not suggest this implication, so needs rewording. It seems like a sentence is missing here, which actually describes your findings, and is followed by the implication.*

**Sentence now reads: "If so, future temperate oceans could be more sensitive to OA resulting in a change in DMS emissions to the atmosphere, whilst perhaps surprisingly DMS emissions from the polar oceans may remain relatively unchanged".**

*2.6  L71 Please highlight that the 3.4 Tg S is from the whole southern ocean, as calculated by JT16.*

**Sentence now reads: "Around 3.4 Tg of sulfur is released from the Southern Ocean to the atmosphere between December and February, a flux that represents ~15 % of global annual emissions of DMS (Jarníková and Tortell 2016)".**

*2.7 L88. Is there a concurrent predicted decrease in the southern polar region pH as well?*

**Sentence now reads: "The greatest declines in pH are likely in the Arctic Ocean with a predicted fall of 0.45 units by 2100 (Steinacher et al. 2009), with a fall of ~0.3 units predicted for the Southern Ocean (McNeil and Matear 2008; Hauri et al. 2016)".**

*2.8 L96. During the introduction there is very little mention of DMSP, other than it is the precursor for DMS. Given that DMSP is one of your measured parameters in the experiments this requires further elaboration in the introduction, in particular the changes in DMSP as identified from the existing mesocosm experiments. This is important given that DMSP showed increases during the Arctic mesocosm.*

**We intentionally focused the introduction on DMS, as our main take home message from the paper is the DMS story which will be of interest to earth-system modellers. However, we agree that integrating some DMSP background may be useful – however, we have kept it brief to avoid lengthening what the reviewer considers an introduction which is 'already very long'.**

**Text now reads:**

**"Despite the imminent threat to polar ecosystems and the importance of DMS emissions to atmospheric processes, our knowledge of the response of polar DMS production to OA is limited to a single mesocosm experiment performed in a coastal fjord in Svalbard (Archer et al. 2013; Riebesell et al. 2013) and one shipboard microcosm experiment with seawater collected from Baffin Bay (Hussherr et al. 2017). Both studies reported significant reductions in DMS concentrations with increasing levels of $p$CO$_2$ during seasonal phytoplankton blooms. Hussherr et al. (2017) also saw reductions in total DMSP whilst Archer et al. (2013) observed a significant increase in this compound, driven by CO$_2$-induced increases in growth and abundance of dinoflagellates. However, these two single studies provide limited information on the wider response of the open Arctic or Southern Oceans".**

*2.9 L100-117. This section on mesocosms is interesting, but slightly irrelevant to the paper as a whole given that your studies are microcosm incubations. As the introduction is already very long, this section could be shortened to one or two sentences describing previous OA responses.*

**I have reduced the text down slightly but feel that the information that remains is useful for the later discussion comparing the results of microcosms with mesocosms.**

**Text now reads (from L104):**

**"Mesocosm experiments have been a critical tool for assessing OA effects on surface ocean communities (Engel et al. 2005; Kim et al. 2006; Engel et al. 2008; Schulz et al. 2008; Hopkins et al. 2010; Kim et al. 2010; Schulz et al. 2013; Webb et al. 2015; Crawfurd et al. 2016; Webb et al. 2016). The response of DMS to OA has been examined several times, predominantly at the same site in Norwegian coastal waters (Vogt et al. 2008; Hopkins et al. 2010; Avgoustidi et al. 2012; Webb et al. 2015), twice in Korean coastal waters (Kim et al. 2010; Park et al. 2014), and a single study in the coastal Arctic waters of Svalbard (Archer et al. 2013). Mesocosm enclosures, ranging in volume from ~11,000 – 50,000 L, allow the response of surface ocean communities to a range of CO$_2$ treatments to be monitored under near-natural light and temperature conditions over time scales (weeks - months) that allow a 'winners vs loser' dynamic to develop. The response of DMS cycling to elevated CO$_2$ is generally driven by changes to the microbial community structure (Engel et al. 2008; Hopkins et al. 2010; Archer et al. 2013; Brussaard et al. 2013). The size, construction and associated costs of mesocosms has limited their deployment to coastal/sheltered waters, resulting in minimal geographical coverage, and leaving large gaps in our understanding of the response of open ocean phytoplankton communities to OA."**

*2.10 L114 High cost is also a significant factor in why mesocosm experiments are so limited!*

**Agreed! Sentence now reads:** *"The size, construction and associated costs of the mesocosms has limited their deployment to coastal/sheltered waters, resulting in minimal geographical coverage, and leaving large gaps in our understanding of the response of open ocean phytoplankton communities to OA".*

*2.11 L120. Please clarify that the temperate experiments shown here are in addition to those in Hopkins and Archer 2014; ie this manuscript includes four previously unpublished temperate experiments, as well as those previously published.*

**Sentence now reads: "We build on the previous temperate NW European shelf studies of Hopkins & Archer (2014) by presenting data from four previously unpublished experiments from the NW European shelf cruise, and by extending our experimental approach to the Arctic and Southern Oceans".**

*2.12 L136, remove additional comma before reference.*

**Done.**

*2.13 L142 What are the main differences in the environment between temperate and polar environments? You have not mentioned the distinct seasonal cycle of sea ice formation and retreat, different at both poles, which will likely also alter the carbonate chemistry. Does the acclimation of polar phytoplankton to the physiological stress of survival through the polar winter give them an added advantage when it comes to acclimating to OA? Is DMSP produced by polar phytoplankton (i.e. osmoregulation during periods of extreme salinity shift) for a different reason than temperate phytoplankton?*

**We hope that our response to 2.2 above addresses most of these comments.**

**We have added some additional discussion with regard to the role of DMSP production by polar communities. We have kept it brief, as the data is not that conclusive, but agree with the reviewer that it is useful to include some discussion of this kind:**

**This section now reads (from L458):**

**"In contrast, at temperate stations, DMSP concentrations displayed a clear negative treatment effect, whilst at polar stations a positive effect was evident under high $CO_2$, and particularly at the first time point (48 – 96 h) (Fig. 7 C and D). *De novo* DMSP synthesis and DMSP production rates show a less consistent response in either environment (Fig. 8 A and B), although a significant suppression of DMSP production rates in temperate waters compared to polar waters was seen (Fig. 8B, Kruskal-Wallis One Way ANOVA *H* = 8.711, *df* = 1, *p* = 0.003). A similar but not significant response was seen for *de novo* DMSP synthesis (Fig. 8A).**

**This data suggests that DMSP concentrations in polar waters may be upregulated in response to OA compared to temperate waters. Given the potential photoprotective and antioxidant role that DMSP plays, and which may be particularly relevant in the highly variable polar sea-ice environment (e.g. irradiance, carbonate chemistry), these changes may reflect a physiological protective response to the experimental OA (Sunda et al. 2002; Galindo et al. 2016). An increase in DMSP concentrations could have either resulted from a physiological up-regulation of DMSP synthesis or a reduction in bacterial DMSP consumption processes. However, DMSP synthesis rates did not provide any conclusive evidence of upregulation in polar waters. Instead, we observed a suppression of rates in temperate waters which may reflect the adverse effects of rapid OA on DMSP producers (Richier et al. 2014, Hopkins and Archer 2014). In contrast, the lesser**

response seen in polar waters may reflect a higher acclimative tolerance to rapid changes in carbonate chemistry amongst polar communities. Further experiments with polar communities would help to further unravel the potential importance of such mechanisms, and whether they facilitated the ability of polar phytoplankton communities to resist the high $CO_2$ treatments".

We have also simplified the text following on from this section to improve the clarity of the discussion (L516 onwards):

"The responses to OA observed for DMS and DMSP production are likely to be reflected in the dynamics of the DMSP-producing phytoplankton. In an assessment across all experiments, Richier et al. (2018) showed that the maximal response to OA of total Chl *a* and net growth rates of small phytoplankton (<10 μm) observed during each experiment, declined the most in relation to increased buffering capacity and temperature of the initial water. Generally, less significant relationships were found between the phytoplankton response and the other wide range of physical, chemical or biological variables that were examined (Richier et al. 2018).

In correspondence with the analyses carried out by Richier et al (2018), at 48 – 96 h (see Table 1), a statistically significant difference in response was seen between temperate and polar waters for Chl *a* (Kruskal-Wallis One Way ANOVA *H* = 20.577, *df* = 1, *p*<0.001). In general, at polar stations phytoplankton showed minimal response to elevated $CO_2$, in contrast to a strong negative response in temperate waters (Fig. 9A). By the second time point (96 – 144 h, see Table 1), no significant difference in response of Chl *a* between temperate and polar waters was apparent (Fig. 9B). As shown in Richier et al. (2014), phototrophic nanoflagellates responded to high $CO_2$ with large decreases in abundance in temperate waters and increases in abundance in polar waters (Fig. 9 C and D), with some exceptions: *North Sea* and *South Sandwich* gave the opposite response. The responses had lessened by the second time point (96 – 168 h, see Table 1).

In contrast, bacterial abundance did not show the same regional differences in response to high $CO_2$ (see Hopkins and Archer (2014) for temperate waters, and Figure S1, supplementary information, for polar waters). Bacterial abundance in temperate waters gave variable and inconsistent responses to high $CO_2$. For all Arctic stations, *Drake Passage* and *Weddell Sea*, no response to high $CO_2$ was observed. For *South Georgia* and *South Sandwich*, bacterial abundance increased at 1000 and 2000 μatm, with significant increases for *South Georgia* after 144 h of incubation (ANOVA *F* = 137.936, *p*<0.001). Additionally, at Arctic stations *Greenland Gyre* and *Greenland Ice-edge*, no overall effect of increased $CO_2$ on rates of DOC release, total carbon fixation or POC : DOC was observed (Poulton et al. 2016).

Overall, the observed differences in the regional response of DMSP and DMS to carbonate chemistry manipulation could not be attributed to any other measured factor that varied systematically between temperate and polar waters. These include ambient nutrient concentrations, which varied considerably but where direct manipulation had no influence on the response, and initial community structure, which was not a significant predictor of the phytoplankton response (Richier et al. 2018)".

*2.14 L190 pCO2*

**Done.**

*2.15 L222. In the southern ocean, was this acidification fixing method used for the DMSP samples, given previous issues highlighted by del Valle et al 2011 in samples containing Phaeocystis antarctica?*

We are grateful to the reviewer for pointing this out. Upon revisiting our methods section, we have altered the text accordingly to take account of this oversight. Text now reads (from L231):

"Methods for the determination of seawater concentrations of DMS and DMSP are identical to those described in Hopkins & Archer (2014) and will therefore be described in brief here. Seawater DMS concentrations were determined by cryogenic purge and trap, with gas chromatography and pulsed flame photometric detection (GC-PFPD) (Archer et al., 2013). DMSP concentrations were measured as DMS following alkaline hydrolysis. Samples for total DMSP concentrations from temperate waters were fixed by addition of 35 µl of 50 % $H_2SO_4$ to 7 mL of seawater (Kiene and Slezak 2006), and analysed following hydrolysis within 2 months of collection (Archer et al. 2013). Samples of DMSP that were collected in polar waters were hydrolysed within 1 h of sample collection and analysed 6 – 12 h later. The $H_2SO_4$ fixation method was not used for samples from polar waters given the likely occurrence of *Phaeocystis sp*. which can result in the overestimation of DMSP concentrations (del Valle et al. 2009).   Similarly, concentrations of DMSPp were determined at each time point by gravity filtering 7 ml of sample onto a 25 mm GF/F filter and preserving the filter in 7 ml of 35 mM $H_2SO_4$ in MQ-water (temperate samples) or immediately hydrolysing (polar samples) and analysing by GC-PFPD. DMS calibrations were performed using alkaline cold-hydrolysis (1 M NaOH) of DMSP sequentially diluted three times in MilliQ water to give working standards in the range 0.03 – 3.3 ng S $mL^{-1}$. Five point calibrations were performed every 2 – 4 days throughout the cruise".

*2.16 L253 spaces missing between mass 63 etc.*

**Sorted.**

*2.17 L281 composition used twice in the sentence.*

Sentence now reads: "Small phytoplankton community composition was assessed by flow cytometry. For details of methodology, see Richier et al. (2014)".

*2.18 L317. Please state the station where 1.5 µg L-1 was identified, as you have for the minimum.*

Sentence now reads: "Chl a was also variable in polar waters, exceeding 4 µg $L^{-1}$ at *South Sandwich* and 2 µg $L^{-1}$ at *Greenland Ice-edge*, whilst the remaining stations ranged from 0.2 µg $L^{-1}$ (*Weddell Sea*) to 1.5 µg $L^{-1}$ (*Barents Sea*) (Figure 2)".

*2.19 L318.'reflected in'. Please reword, firstly as reflected implies light reflectance (given the topic of irradiance), and secondly it reads oddly with two instances of 'in'.*

Sentence now reads: "The high Chl *a* concentrations at *South Sandwich* correspond to low in-water irradiance levels at this station (Fig. 2 C)".

*2.20 L320 – 328. Use of nM when the following paragraph uses nmol L-1*

**All changed to nmol $L^{-1}$.**

*2.21 L327. Could not understand the significance of the superscript 1, is it a typo?*

**Yes typo. Deleted.**

*2.22 L480. Should P be lower case?*

**Changed to lower case *p*.**

*2.23 L485. In this section I would like to see more discussion of sea ice and the extreme environment is imposes on the cells, which could account for some of their resilience to change. Many polar phytoplankton survive the extreme cold of the polar winter by living within the sea ice itself, in an extremely changeable habitat, and these cells seed the surface waters in summer time on melting of the sea ice. Cells are regularly exposed to hypersaline and highly nutrient variable environments, at temperatures below freezing, and in highly elevated pH environments (Thomas and Dieckmann 2002, Rysgaard et al 2012). Although your experiments were not associated directly with the polar sea ice and occurred during summer, the influence of the ice on the phytoplankton population will be dependent on the seed populations, and allow for greater tolerance of the incubation perturbation than in temperate communities. In the seasonal cycle of the Antarctic, the behaviour of the summer phytoplankton community development is dependent on the conditions experienced the previous winter (Venables et al 2013).*

**We have addressed this comment in 2.2 above.**

*2.24 L523. P in italics*

**Done.**

*2.25 L561. Hopkins reference is 2010b, but only 1 Hopkins 2010 ref is present.*

**The b has been deleted.**

*2.26 L609 italic p*

**Done.**

*2.27 L626, commas missing from references.*

**Commas inserted.**

**References: see manuscript.**

[revised manuscript text omitted]

A. Greenland Gyre

| $CO_2$ treatment | | $pCO_2$ (µatm) mean ± SD, $n = 3$ | | |
|---|---|---|---|---|
| | | 0 h | 48 h | 96 h |
| Ambient | ● | 326.8 | No data | 251.5 ± 5.1 |
| Mid | ■ | 565.2 | 493.8 ± 49.0 | 416.7 ±18.5 |
| High | ▲ | 741.8 | 607.6 ± 53.9 | 526.1 ± 18.1 |
| High+ | ✕ | 1012.2 | 824.1 ± 99.5 | 668.6 ± 33.6 |

B. Greenland Ice edge

| $CO_2$ treatment | | $pCO_2$ (µatm) mean ± SD, $n = 3$ | | |
|---|---|---|---|---|
| | | 0 h | 48 h | 96 h |
| Ambient | ● | 312.2 | 281.1 ± 1.8 | 201.1 ± 3.4 |
| Mid | ■ | 583.9 | 444.2 ± 9.4 | 301.3 ± 8.4 |
| High | ▲ | 789.3 | 576.3 ± 35.2 | 412.6 ± 78.6 |
| High+ | ✕ | 948.2 | 803.6 ± 10.5 | 483.8 ± 38.1 |

C. Barents Sea

| $CO_2$ treatment | | $pCO_2$ (µatm) mean ± SD, $n = 3$ | | |
|---|---|---|---|---|
| | | 0 h | 48 h | 96 h |
| Ambient | ● | 310.6 | 291.8 ± 10.6 | 289.9 ± 1.5 |
| Mid | ■ | 535.1 | 482.8 ± 4.9 | 483.5 ± 18.6 |
| High | ▲ | 649.1 | 653.7 ± 3.6 | 547.0 ± 28.4 |
| High+ | ✕ | 683.6 | 763.0 ± 12.4 | 673.5 ± 65.7 |

Figure 3. DMS concentrations (nmol L$^{-1}$) during experimental microcosms performed in Arctic waters . Data shown is mean of triplicate incubations, and error bars show standard error on the mean. Tables show measurements of $pCO_2$ (µatm) for each treatment at each sampling time point. Initial measurements (0 h) were from a single sample, whilst measurements at 48 h and 96 h show mean ± SD of triplicate experimental bottles. Locations of water collection for microcosms shown in Figure 1 C – F.

[Figure]

Figure 4. DMS concentrations (nmol L$^{-1}$) during experimental microcosms performed in
Southern Ocean waters. Data shown is mean of triplicate incubations, and error bars show
standard error on the mean. Tables show measurements of $p$CO$_2$ (μatm) for each treatment at
each sampling time point. Initial measurements (0 h) were from a single sample, whilst
measurements at 48 h and 96 h show mean ± SD of triplicate experimental bottles. Locations
of water collection for microcosms shown in Figure 1 C – F.

[Figure]

Figure 45. Total DMSP (solid lines) and particulate DMSP (dashed lines) concentrations (
nmol L$^{-1}$) during experimental microcosms performed in Arctic waters (A - C) and in
Southern Ocean waters (D – G). Data shown is mean of triplicate incubations, and error bars
show standard error on the mean. Locations of water collection for microcosms shown in
Figure 1 C – F. Particulate DMSP concentrations were used in calculations of DMSP
production rates (Figure 6).

[Figure]

Figure 56. De novo synthesis of DMSP (μDMSP, $d^{-1}$) (left column) and DMSP production
rates (nmol $L^{-1}$ $d^{-1}$) (right column) for Arctic Ocean stations *Greenland Gyre* (A,B),
*Greenland Ice-edge* (C, D) and Southern Ocean stations *Drake Passage* (E, F), *Weddell Sea*
(G, H) and *South Georg*ia (I, J). No data is available for *Barents Sea* (Arctic Ocean) or *South*
*Sandwich* (Southern Ocean).

[Figure]

Figure 67. Relationship between Revelle Factor of the sampled water and the relative $CO_2$ treatment effect at ($[x]_{highCO2}/[x]_{ambientCO2}$) for concentrations of DMS at $T_1$ (A) and $T_2$ (B), and for total DMSP concentrations at $T_1$ (C) and $T_2$ (D) for all microcosm experiments performed in NW European waters, sub-Arctic and Arctic waters, and the Southern Ocean. Grey solid line (= 1) indicates no effect of elevated $CO_2$. Revelle Factor > 12 = polar waters (indicated by red dashed line). $T_1$ = 48 h, except for WS and SG (72 h) and SS (96 h). For detailed analyses of the NW European shelf data, see Hopkins & Archer (2014).

[Figure]

[Figure]

Figure 8. Relationship between the Revelle Factor of the sampled water and the relative $CO_2$ treatment effect at ($[x]_{highCO2}/[x]_{ambientCO2}$) for de novo DMSP synthesis ($\mu$DMSp, d$^{-1}$) at $T_1$ (A) and $T_2$ (B), and DMSP production rate (nmol L$^{-1}$ d$^{-1}$) at $T_1$ (C) and $T_2$ (D) for microcosm experiments performed in NW European waters, sub-Arctic and Arctic waters, and the Southern Ocean. Grey solid line (= 1) indicates no effect of elevated $CO_2$. Revelle Factor >12 = polar waters (indicated by red dashed line). $T_1$ = 48 h, T2 = 96 h, except for *Weddell Sea* and *South Georgia* (72 h, 144 h). For discussion of the NW European shelf data, see Hopkins & Archer (2014).

[Figure]

Figure 89. Relationship between the Revelle Factor of the sampled water and the relative $CO_2$ treatment effect ($[x]_{highCO2}/[x]_{ambientCO2}$) for chlorophyll *a* concentrations at $T_1$ (A) and $T_2$ (B) and phototrophic nanoflagellate abundance at T1 (C) and T2 (D) for all microcosm experiments performed in NW European waters, sub-Arctic and Arctic waters, and the Southern Ocean. Grey solid line (= 1) indicates no effect of elevated $CO_2$. Revelle Factor >12 = polar waters (indicated by red dashed line). $T_1$ = 48 h, $T_2$ = 96 h, except for *Weddell Sea* and *South Georgia* (72 h, 144 h) and *South Sandwich* (96 h, 168 h).

---

## Author Response (AR3)

**Final author comments on "Polar dimethylsulfide (DMS) production insensitive to ocean acidification during shipboard microcosm experiments: a meta-analysis of 18 experiments from temperate to polar waters." by F.E. Hopkins et al., manuscript number bg-2018-55.**

*We are grateful to the reviewer for their positive view of our manuscript and their thorough assessment, which will bring great improvements. The reviewers comments are shown in italics, with our responses shown* **in bold**. *Line numbers in our response refer to the revised version. The marked-up version of the manuscript comes after the response (starting at pg 10), and line numbers refer to this version.*

1. **Response to Anonymous Referee #1**

*1.1 The paper describes the results of microcosm experiments examining the response of DMS and DMSP concentration, synthesis & production rates to acidification in Southern Ocean & Arctic waters, and compares them with previously published results from the NW European shelves. The primary results are the absence of an effect of high $CO_2$ on DMS/DMSP in polar waters, and the contrasting significant effect of high $CO_2$ in decreasing DMSP concentration/DMSP production & increasing DMS production in temperate waters. The authors relate the difference in regional DMS response to variability of the carbonate system of each region; other factors (phytoplankton community, nutrients etc) are rejected due to the absence of significant relationships. The paper makes some interesting points regarding regional variation in response to acidification, which should be considered in models.*

**We thank the reviewer for their supportive comments on our paper. Our primary target audience includes climate modellers, and we agree that the DMS data we present should be considered in models where information on future DMS fluxes from polar regions is currently minimal.**

*1.2 I have two "medium" concerns with paper, the first being that the DMSP response to High $CO_2$ is a little overstated; a significant DMSP response to High $CO_2$ is limited to the first 48 hours in the polar experiments, after which there is no significant difference from the control. My other concern is that the interpretation, and proof of their hypothesis, rests on differences in regional variability of pH & the carbonate system, yet the data presented to support this are somewhat limited. I appreciate there are limited pH datasets available, but the data discussed are primarily large-scale spatial variability, and not the temporal variability that the phytoplankton experience.*

**We thank the reviewer for their considered comments and appreciate the improvements that they will make to our manuscript. We have endeavoured to address all their comments and concerns in our point-by-point response below.**

*Specific comments*
*1.3 Title doesn't scan as written ("IS insensitive") and is a little confusing with reference to the polar results, & then then the metanalysis of polar AND temperate results. A better title might be:*
*"DMS sensitivity to ocean acidification as determined in a meta-analysis of temperate to polar microcosm experiments"*
*Or, to highlight the polar results:*
*"A metanalysis of microcosm experiments shows that DMS production in polar waters is insensitive to ocean acidification"*

We tend to agree with the reviewer with regards to the confusing title – the result of multiple reviews! So we thank this reviewer for pointing this out and making some nice suggestions for the title. We have decided to go with:

**"A metanalysis of microcosm experiments shows that DMS production in polar waters is insensitive to ocean acidification"**

*1.4 Abstract Line 26 "resulting in an INCREASE in DMS emissions to the atmosphere….."*

**Text altered accordingly.**

*1.5 Discussion Line 443-445; The text implies there is a significant positive effect on DMSP of high CO2 at the 2nd time point, but this does seem apparent in Fig. 7D*

**The reviewer refers to:**

**"In contrast, at temperate stations, DMSP concentrations displayed a clear negative treatment effect, whilst at polar stations a positive effect was evident under high CO$_2$ and particularly at the first time point (48 – 96 h) (Fig. 7 C and D)".**

**We don't use the term 'significant' in this part of the text. We feel it is fair to say "a positive effect was evident", as the majority of data points fall above 1, indicating a positive treatment effect of high CO$_2$.**

*1.6 Line 450 "This data suggests that DMSP concentrations in polar waters may be upregulated…" again this is only for the first 48 hours; Fig 7D doesn't show a significant mean effect at 96 hours for polar waters. This could be a "shock" response to the dramatic alteration of pH and the carbonate system, before the phytoplankton community acclimate, which should be mentioned. Also, perhaps more accurate to say the results reflect a downregulation of DMSP in temperate waters (relative to polar waters), as the mean DMSP effect is significant for both experimental time periods temperate waters in Figs 7C&D (unlike in polar waters where there is only a significant difference in the first 24 hours in Fig 7C).*

**This paragraph has been re-written to take the reviewer's comments into consideration. Now reads (from L475):**

**"Our data imply that DMSP concentrations in temperate waters were downregulated in response to OA, attributed to the adverse effects of rapid OA on the growth of DMSP producers which led to reductions in the abundance of these types of phytoplankton (Richier et al. 2014, Hopkins and Archer 2014). By comparison, a more muted, but generally positive, DMSP response was seen in polar waters at the first time point, whilst these treatment effects were more or less undetectable by the second time point. There is some evidence that the enhanced DMSP concentrations in polar waters were accompanied by increased DMSP production rates (Figure 8), although data is not available for all experiments. However, these changes may reflect a short term 'shock' physiological protective response to the experimental OA, similar to that seen in response to other short term stressors such as high irradiance that result in an increase in DMSP concentrations (Sunda et al., 2002;Galindo et al., 2016). The lack of treatment effect in DMSP concentrations by the second time point may be indicative that the community had, to some extent, acclimated to the change, allowing DMSP production/concentrations to return to baseline levels. This may reflect a higher degree of**

tolerance to rapid changes in carbonate chemistry amongst polar communities - species which are already adapted to highly variable irradiance/carbonate chemistry regimes (Thomas and Dieckmann, 2002; Rysgaard et al., 2012; Thoisen et al., 2015). Further experiments with polar communities would help to unravel the potential importance of such mechanisms and whether they facilitated the ability of polar phytoplankton communities to resist the high $CO_2$ treatments".

*1.7 Line 459 "rapid OA on DMSP producers…" more detail required here. Is this an algal physiological response or change in phytoplankton community composition?*

**We have rewritten this section so this sentence is no longer included in its original form. See response to 1.6 above.**

**This new sentence, starting at L475 replaces the previous:**

**"Our data imply that DMSP concentrations in temperate waters were downregulated in response to OA, attributed to the adverse effects of rapid OA on the growth of DMSP producers which led to reductions in the abundance of these types of phytoplankton (Richier et al. 2014, Hopkins and Archer 2014)".**

*1.8 Line 460 This paragraph is a little contradictory; starts by saying DMSP production is upregulated in polar relative to temperate, and finishes with "the lesser response seen in polar waters"*

**Again, we have rewritten this section to make our point more clearly (See response to 1.6 above), so it is no longer included in its original form. The underlined sentence below addresses the reviewers specific comment:**

**"Our data imply that DMSP concentrations in temperate waters were downregulated in response to OA, attributed to the adverse effects of rapid OA on the growth of DMSP producers which led to reductions in the abundance of these types of phytoplankton (Richier et al. 2014, Hopkins and Archer 2014). By comparison, a more muted, but generally positive, DMSP response was seen in polar waters at the first time point, whilst these treatment effects were more or less undetectable by the second time point".**

*1.9 Line 466. "In an assessment across..."; this sentence should be rewritten for clarity*

**We have rewritten this sentence and broken the information down into two sentences. It now reads (starts L513):**

**"In an assessment across all experiments, Richier et al. (2018) showed that the magnitude of biological responses to short term $CO_2$ changes reflected the buffer capacity of the sampled waters. A consistent suppression of net growth rates in small phytoplankton (<10 μm) and total Chl *a* concentrations was observed under high $CO_2$ within experiments performed in temperate waters with higher buffer capacity".**

*1.10 Line 533-540. "The polar waters sampled during our study were characterised by pronounced gradients in carbonate chemistry over small spatial scales". Where is this shown? And what are the "small spatial scales"? The pH range of the whole voyage ("along each cruise track") is mentioned but theres no information provided on the length of these transects which*

*makes its difficult to compare the pH gradient between regions, or assess how significant the pH spatial gradients are. Phytoplankton won't experience the total pH range measured on a voyage, as their exposure will be limited by physical constraints such as currents and water masses. Furthermore, the sites sampled within each voyage will experience different pH variability. The information on the drivers of spatial gradients (in paragraph Line 541-554) is interesting, but theres no indication of the spatial scale associated with this to relate spatial pH gradients to phytoplankton. The authors have provided some examples of the temporal pH variation at certain sites (Paragraph Line 555-564), but these do not consider all pH data available (for example, Beare et al (2013) show large variation (7.8-8.5) in the central North Sea). The authors supply supporting evidence from other experiments that polar phytoplankton are relatively insensitive to variation in pH, but to support their hypothesis that the regional variation in pH is the primary factor driving differences in DMSP response, more evidence and analysis of differences in regional pH variability are required.*

*Beare, D., McQuatters-Gollop, A., van der Hammen, T., Machiels, M., Teoh, S. J., & Hall-Spencer, J. M. (2013). Long-term trends in calcifying plankton and pH in the North Sea. PLoS One, 8(5), e61175.*

**We thank the reviewer for highlighting this section, which has been subject to the scrutiny of several reviewers and had perhaps lost it way somewhat. Unfortunately, we do not have data on the pH variability at each sampling station so necessarily we must make our point using what data we have available, and that is the underway measurements along the cruise track. We realised an important point was missing from this section: that the OA response in polar vs temperate waters relates to the Revelle Factor (buffering capacity) of the sampled waters – it is this then that contributes to the greater level of variability in carbonate chemistry seen in polar waters. Thus, the point we wish to make is that the polar communities are already adapted to a variable carbonate chemistry environment, and this is the result of lowered buffering capacity in polar waters, and is also accentuated by other processes that occur in polar waters which create fluctuations in carbonate chemistry.**

**We thank the reviewer for flagging Beare et al. (2013) who do indeed show a wide range in pH in the North Sea. However, the sampling locations (Figure 1) are biased towards the German Bight, a region heavily influenced by the outflow of regional rivers which results in strong alkalinity driven pH fluctuations (see Artioli et al. 2012, 2014). The unique riverine-influenced, shallow characteristics of this region make it challenging to compare this data to the open ocean sampling locations that we are presenting.**

**Artioli, Y., Blackford, J. C., Butenschön, M., Holt, J. T., Wakelin, S. L., Thomas, H., ... & Allen, J. I. (2012). The carbonate system in the North Sea: Sensitivity and model validation. *Journal of Marine Systems*, *102*, 1-13.**

**Artioli, Y., Blackford, J. C., Nondal, G., Bellerby, R. G. J., Wakelin, S. L., Holt, J. T., ... & Allen, J. I. (2014). Heterogeneity of impacts of high CO 2 on the North Western European Shelf. *Biogeosciences*, *11*(3), 601-612.**

**We have added some additional text to section 4.4 and we hope this satisfies the reviewer's concerns. The section now reads (starts L582):**

**"4.3 Adaptation to a variable carbonate chemistry environment**

**Given that DMS production by polar phytoplankton communities appeared to be insensitive to experimental OA compared to significant sensitivity in temperate communities, we hypothesise**

that polar communities are adapted to greater natural variability in carbonate chemistry over spatial and seasonal scales. **This greater variability is partly the result of the lower buffering capacity (Revelle Factor) of polar waters compared to lower latitude waters, and partly due to specific processes that occur in the polar regions that strongly alter DIC concentrations (e.g. sea ice formation and melt, enhanced $CO_2$ dissolution into cold polar waters, upwelling of $CO_2$ rich water). Therefore, polar plankton communities are not only subject to geophysical processes that strongly alter in situ carbonate chemistry on both spatial and seasonal scales, but such changes are accompanied by larger pH changes than would occur in more strongly buffered temperate waters. Therefore, polar surface ocean communities are perhaps more likely to experience fluctuations between high pH and low pH over relatively smaller time/space scales (Tynan et al., 2016). Thus below, we discuss our findings in the context of the spatial pH variability we observed for each cruise track, and explore some of the processes that drive this variability in polar waters. Information on the pH variability at each sampling station is not available, so we cannot be certain of the exact carbonate chemistry variability to which each of the sampled communities may have been exposed and adapted. However, we can consider the overall variability in carbonate chemistry over the spatial scales of the cruise tracks to demonstrate the characteristics of each study area.**

The polar waters sampled during our study were characterised by pronounced gradients in carbonate chemistry over relatively small spatial scales.  In underway samples taken along each cruise track (Arctic Ocean 3500 nm, Southern Ocean 4000 nm), pH varied by 0.45 units (8.00 – 8.45) in the Arctic, and 0.40 units (8.30 - 7.90) in the Southern Ocean (Tynan et al. 2016). In some cases this range in variability was seen over relatively small distances: Figure 4 in Tynan et al. (2016) shows that pH fluctuated from 8.45 and 8.0 over a distance of 50 – 100 miles in the sea-ice influenced Fram Strait. By comparison, pH varied by a total of 0.2 units (8.22 - 8.02) in underway samples from the NW European shelf sea cruise (Rerolle et al. 2014).  The observed horizontal gradients in polar waters were driven by different physical and biogeochemical processes in each ocean. In the Arctic Ocean, this variability in carbonate chemistry was partly driven by physical processes that controlled water mass composition, temperate and salinity, particularly in areas such as the Fram Strait and Greenland Sea. Along the ice-edge and into the Barents Sea, biological processes exerted a strong control, as abundant iron resulted in high chlorophyll concentrations, low DIC and elevated pH. By contrast, variations in temperature and salinity had only a small influence on carbonate chemistry in the Southern Ocean in areas with iron limitation, and larger changes were driven by a combination of calcification, advection and upwelling. Where iron was replete, e.g. near South Georgia, biological DIC drawdown had a large impact on carbonate chemistry (Tynan et al. 2016). A further set of processes was in play in sea ice influenced regions. At the Arctic ice edge, abundant iron drove strong bloom development along the ice edge, whilst sea ice retreat in the Southern Ocean was not always accompanied by iron release (Tynan et al. 2016)."

*1.11 Line 634 Section 4.4. This section should highly other benefits of mesocosms (inclusion of larger components of the foodweb and physical factors (mixing, stratification, particle export) that are excluded from microcosms, and note that their longer duration & more holistic/inclusive framework makes mesocosm results more relevant for Earth System models. This section should also consider the shock effect of sharply altering pH in microcosms*

> **We completely agree with the reviewer's comments on this and agree that it would be useful to include such information in the paper. However, the specific section that they refer to does not seem to be the best part of the paper for this. We feel that it would fit better in the introduction, in the section beginning "Mesocosm experiments have been a critical tool..." where we already touch upon such issues. Therefore, we have added the following text to this section (starts L124):**

**"The pseudo-natural conditions of mesocosm experiments offer the benefit of the inclusion of community dynamics of three or more trophic levels, providing the opportunity to investigate the influence of ecosystem dynamics on biogeochemical processes under experimental conditions (Riebesell et al., 2013b). Furthermore, physical processes such as particle export (Bach et al., 2016), which would be excluded by smaller scale experiments, can be considered within the holistic mesocosm framework, and make the results relevant for use within Earth system models (Six et al. 2013). However,** the size, construction and associated costs of mesocosms has limited their deployment to coastal/sheltered waters, resulting in minimal geographical coverage, and leaving large gaps in our understanding of the response of open ocean phytoplankton communities to OA".

With regards to the reviewer's comment on the shock effect of altering pH in microcosms: we do not include specific discussion in this regard because we did not see any evidence of a shock response in the polar experiments. However, we do allude to our previous findings (Hopkins and Archer 2014) as possibly being driven by an acute 'shock' response to sudden pH change (e.g. L157-159). We have added some additional text to the introduction to show consideration for the potential shock effect (starts L149):

"The rapid $CO_2$ changes implemented in this study, and during mesocosm studies, are far from representative of the predicted rate of change to seawater chemistry over the coming decades, **and the potential to induce a 'shock' response to the sudden alteration of carbonate chemistry should be considered, particularly when working at the smaller microcosm scale".**

*1.12 Line 722. "Our findings contrast with two previous studies....." this sentence (the reasons for differences in previous polar microcosm responses) should be expanded on in the Discussion section*

The "two previous studies..." to which this sentence refers to are Archer et al. (2013) and Hussherr et al. (2017). The whole of section 4.4 is dedicated to discussing the reasons why our findings contrast with the Arctic mesocosm experiment of Archer et al., whilst L774-786 discusses the differences with the polar microcosm experiment of Hussherr et al. The line which has been highlighted by the reviewer is simply a summary of the previous discussion. We feel no further discussion is required.

*Technical corrections/Minor comments*

*Introduction*
*1.13 Line 52-66; why does this paragraph focus on pack ice & associated climate-related changes?*

The reviewer refers to the section which starts "The biologically-rich seas surrounding the Arctic pack ice...". The intention of this paragraph was to explain the importance of the Arctic Ocean for DMS production – our motivation for investigating the effects of OA from this region. Perhaps use of the phrase 'pack-ice' in unnecessary and we can simply refer to the Arctic Ocean ice-edge and open waters. Therefore we have reworded as (L55)

"The biologically-rich ice-edge regions and open seas of the Arctic are a strong source of DMS to the Arctic atmosphere (Levasseur, 2013)".

We finish this paragraph by describing how sea ice loss may impact future DMS emission from the Arctic. Given the reviewer's comment, we feel it would be better to end this paragraph by mentioning the potential effect of OA in the Arctic on DMS emissions. Thus we have added:

"The influence that OA will have on the production and flux of DMS, and how this may further influence the Arctic radiative balance, is poorly understood and requires further experimental and modelling efforts".

*1.14 Line 113 "'winners vs loser' dynamic": this requires some explanation*

**Now reads (L116 – 121):**

"Mesocosm enclosures, ranging in volume from ~11,000 – 50,000 L, allow the response of surface ocean communities to a range of $CO_2$ treatments to be monitored under near-natural light and temperature conditions over time scales (weeks - months). This is sufficient time to allow a 'winners vs loser' dynamic to develop, whereby the succession of the phytoplankton community is altered due to the differing sensitivities of different taxonomic groups to changes in carbonate chemistry (Bach et al., 2017)".

*1.15 Line 137 "Polar" not required in this sentence as the paper presents experiments from temperate waters as well as polar*

**"polar" deleted from sentence.**

*Methods*
*1.16 Line 183. For the Drake Passage and Weddell Sea experiments suggest the Fe experiments are not mentioned (as their results aren't discussed), and the same notation is used as for the other experiments (High CO2, High CO2 +, etc)*

**We have removed mention of the Fe experiments and the section now reads:**

"For Southern Ocean experiments, two experiments (*Drake Passage* and *Weddell Sea*) considered one $CO_2$ treatments (High). Three $CO_2$ treatments (High, High+, High++) were tested in the last two experiments (*South Georgia* and *South Sandwich*)".

*1.17 Line 215-219. A 48-hour experiment seems very short; however, as pointed out growth rates are faster in temperate waters than polar incubations. The authors may want to use these differential growth rates to justify the comparison of response of shorter duration temperate experiments & longer duration polar experiments.*

**To make this point clearer to the reader, we have modified a line in this section (L233):**

"The differential growth/metabolic rates between temperate and polar waters justify the comparison of response of shorter duration temperate experiments and longer duration polar experiments".

*1.18 Line 211. IF the high frequency results are not discussed them exclude this from the Methods*

**The sentence referring to the high time frequency sampling has been deleted.**

*Results*
*1.19 Fig 2. As DMS & DMSP results (G-I) are only presented to depths of 100m, the non-DMS/P*

*parameters (A-F) should be only shown for this depth range, particularly as only the surface values for non-DMS/P variables are discussed in the Results section. Currently details of the depth profiles in A-F are not visible due to the extended vertical axis used.*

**Figure 2 has been altered so all parameters are plotted to 100m.**

*1.20 Fig 2. The depth profiles on DMS/P are not really discussed; the maxima is not always at the surface and these sub-surface maxima appear to be associated chl-a subsurface maxima*

**We have altered the text starting at L351 to take the reviewer's comments into consideration:**

**"In temperate waters, maximum DMS concentrations were generally seen in near surface measurements, ranging from 1.0 nmol L$^{-1}$ for *E04* to 21.1 nmol L$^{-1}$ for *E06*, with rapidly decreasing concentrations with depth (Figure 2 G). As an exception to this, DMS concentrations at *South Sandwich* showed a sub-surface maximum of 15 nM at 32 m, coincident with a subsurface Chl *a* maximum of 5.4 µg L$^{-1}$. DMSP generally ranged from 12 – 20 nmol L$^{-1}$, except *Barents Sea* where surface concentrations exceeded 60 nmol L$^{-1}$ (Figure 2 H). DMSP tended to peak in the near surface waters, ranging from 12.0 nmol L$^{-1}$ for E04 to 72.5 nmol L$^{-1}$ for *E06*, although in some cases a subsurface maximum in overall DMSP concentrations was seen, as observed for *E05b* (89.8 nmol L$^{-1}$ 20 m), and again coincident with a subsurface Chl a peak of >2 µg L$^{-1}$ (Figure 2 F and H). Surface DMS concentrations in polar waters were generally lower than temperate waters, ranging from 1 – 3 nmol L$^{-1}$, with the exception of *South Sandwich* where concentrations of ~12 nmol L$^{-1}$ were observed (Figure 2 G), and resulted in high DMS:DMSP of 0.6 – 0.9 in the surface layer (Figure 2 I). DMS:DMSP did not exceed 0.5 at any other sampling stations".**

*1.21 Line 344. This description is misleading; both the 48-hr and 96-hr samples were collected within the incubation period*

**Text altered accordingly and now reads (L369):**

**"Initial concentrations of 1 – 2 nmol L$^{-1}$ remained relatively constant over the first 48 h and then showed small increases of 1 - 4 nmol L$^{-1}$ over the remainder of the incubation period."**

*1.22 Line 351 Fig 5 is DMSP data, not Fig 4*

**Corrected.**

*1.23 Line 372 The unpublished temperate data are an important component of this paper and so the Suppl. Table 2 data should instead be a Figure in the main paper (possibly including comparison with the publ. data from Hopkins& Archer (2014)).*

**We don't feel it would be useful to add another figure to the main paper – Figure 7, 8 and 9 include the data from the previously unpublished experiments in a useful visual meta-analysis. However, we have added a table to the main paper showing the DMS and DMSPt data from the four small-scale experiments to make the information easily accessible to the reader (moved from the supplementary info).**

*1.24 Line 388. "effect" missing*

**Sentence now reads: "No consistent effect of high CO$_2$…"**

*Discussion*
*1.25 Line 487 "For all Arctic stations …"; Drake Passage & the Weddell sea are not in the Arctic*

**The text has been altered and so now reads (L540):**

**"For all Arctic stations, as well as Southern Ocean stations *Drake Passage* and *Weddell Sea*, no response to high $CO_2$ was observed".**

*1.26 Line 698-702. These two sentences don't seem to address the topic of Section 4.4, or this paragraph and should perhaps be moved elsewhere in the text*

**These two sentences have been removed from this section, and we have altered the text at L682-686:**

[revised manuscript text omitted]